# SDS-22 stabilizes GSP-1/-2 PP1 subunits contributing to polarity establishment in *C. elegans* embryos

Yi Li (ID), Ida Calvi (ID) & Monica Gotta (ID) ✉

## Abstract

In many cells, polarity depends on the asymmetric distribution of the conserved PAR proteins, maintained by a balanced activity between kinases and phosphatases. The *C. elegans* one-cell embryo is polarized along the anterior-posterior axis, with the atypical protein kinase C PKC-3 enriched in the anterior, and the ring finger protein PAR-2 in the posterior. PAR-2 localization is regulated by PKC-3 and the PP1 phosphatases GSP-1/-2. Here we find that depletion of the conserved PP1 interactor SDS-22 leads to a partial rescue of the polarity defects of a *pkc-3* temperature-sensitive mutant. Consistent with the rescue, SDS-22 depletion or mutation results in reduced GSP-1/-2 protein levels and activity. The decreased levels of GSP-1/-2 can be rescued by reducing proteasomal activity. Our data suggest that SDS-22 contributes to polarity by protecting the GSP-1 and GSP-2 catalytic subunits from proteasome-mediated degradation, supporting recent data in human cells showing that SDS22 is required to stabilize nascent PP1.

**Keywords** Cell Polarity; PP1 Phosphatases; SDS22; PAR Proteins; Proteasomal Degradation
**Subject Categories** Cell Adhesion, Polarity & Cytoskeleton; Development; Post-translational Modifications & Proteolysis

## Introduction

Cell polarity is required to maintain cellular and tissue architecture and function. In epithelial cells, polarity specifies the apical, basal and lateral membrane domains, enabling the selective barrier function of the epithelium (Riga et al, 2020). In migrating cells, front-to-rear polarity is important for cells to respond to environmental cues such as inflammatory factors, bacterial products, or growth factors (Llense and Etienne-Manneville, 2015). Cell polarity is crucial for asymmetric cell division, which is a prerequisite for stem cells to self-renew and to generate specialized daughter cells (Santoro et al, 2016), and for embryos to properly develop (Campanale et al, 2017). In many cells, polarity is regulated by the conserved partitioning defective (PAR) proteins, which were first identified in *C. elegans* (Goldstein and Macara, 2007; Lang and Munro, 2017).

The one-cell *C. elegans* embryo is polarized along the anterior-posterior axis. The anterior PAR proteins (the PDZ proteins PAR-3 and PAR-6, the atypical protein kinase C PKC-3 and the small GTPase CDC-42) are localized to the anterior cortex and cytoplasm of the embryo, while the posterior PAR proteins (the kinase PAR-1, the ring finger protein PAR-2, the lethal giant larvae ortholog LGL-1 and the CDC-42 GAP CHIN-1) are enriched in the posterior half of the embryo (reviewed in (Goehring, 2014; Lang and Munro, 2017)). Before polarity establishment, the anterior PAR proteins are uniformly localized on the cortex, while the posterior PAR proteins are localized in the cytoplasm. Shortly after fertilization, the centrosomal mitotic kinase Aurora A (AIR-1) removes the Rho GEF ECT-2 from the posterior pole (Kapoor and Kotak, 2019; Klinkert et al, 2019; Longhini and Glotzer, 2022; Zhao et al, 2019). This event initiates a cortical flow that segregates the anterior PAR proteins to the anterior and liberates the posterior cortex of the embryo, which is then occupied by the posterior PAR proteins (Munro et al, 2004).

Several studies have emphasized the importance of the regulation of PAR-2 phosphorylation by PKC-3 in establishing and maintaining cortical polarity (Hao et al, 2006; Motegi et al, 2011). Recently, our group has shown that PP1 phosphatases are also required for cortical polarity in one-cell embryos (Calvi et al, 2022). Before polarity establishment, PKC-3 phosphorylates PAR-2 and inhibits PAR-2 localization at the cortex (Hao et al, 2006; Motegi et al, 2011). The PP1 phosphatases GSP-1/-2 are required for PAR-2 cortical posterior accumulation and embryo polarization. Phosphomimetic mutants of PAR-2 (Hao et al, 2006), or mutations that abolish the interaction between PAR-2 and GSP-1/-2 (Calvi et al, 2022), prevent PAR-2 localization at the posterior, indicating that PP1 phosphatases are crucial for polarity establishment in *C. elegans* embryos.

Analogous to the intricate regulation of protein kinases, the activity of phosphatases towards specific substrates must be tightly regulated for proper cellular function, including cell division, heat shock response, and glycogen metabolism (Ceulemans and Bollen, 2004). The activity of PP1 is primarily regulated post-translationally through interactions with PP1 interacting proteins, which guide its localization, substrate specificity and function. Over 200 PP1 interactors have been reported in different species and they regulate both in time and space the targeting of catalytic subunits towards different substrates (Aggen et al, 2000; Verbinnen

Department of Cell Physiology and Metabolism, Faculty of Medicine, University of Geneva, Geneva 1211, Switzerland. ✉E-mail: monica.gotta@unige.ch

et al, 2017; Virshup and Shenolikar, 2009). One non-canonical regulator of PP1 is the conserved SDS22 protein. In mammalian cells SDS22 suppresses Aurora B activity to control cell division (Duan et al, 2016; Posch et al, 2010; Wurzenberger et al, 2012). In *Drosophila melanogaster*, kinetochore-localized PP1–Sds22 dephosphorylates moesin at cell poles, facilitating anaphase elongation through polar relaxation (Rodrigues et al, 2015). PP1-87B and Sds22 also counteract aPKC and Aurora A kinase phosphorylation of Lgl, regulating apical-basal polarity in follicle epithelial cells (Moreira et al, 2019). In *Saccharomyces cerevisiae*, Sds22, along with Ypi1, is critical for the nuclear import and stabilization of PP1-Glc7 (Cheng and Chen, 2015; Peggie et al, 2002). Depletion of either Sds22 or Ypi1 results in mitotic arrest (Pedelini et al, 2007). Recent biochemical studies (Kueck et al, 2024) and studies in human cells (Cao et al, 2024) have demonstrated that SDS22 stabilizes nascent PP1 but needs to be removed from PP1 for the enzyme to be active. These recent findings suggest that while SDS22 is required for the biogenesis of PP1 holoenzymes, its removal from PP1 is essential to have an active phosphatase. This dual role of SDS22 explains how SDS22 behaves as an inhibitor of PP1 in biochemical assays in vitro but as an activator in vivo (Cao et al, 2024; Cao et al, 2022; Kueck et al, 2024; Lesage et al, 2007).

In *C. elegans* GSP-1 and GSP-2, orthologues of human PP1β and PP1α, contribute to diverse processes such as germline immortality (Billmyre et al, 2019), sister chromatid cohesion in meiosis (Hsu et al, 2000; Tzur et al, 2012), centriole duplication (Peel et al, 2017), PAR-2 localization in the early embryos (Calvi et al, 2022), and the maintenance of epidermal junctions in adult worms (Beacham et al, 2022). Associated regulatory proteins have been identified for specific functions: for example, SDS-22 and SZY-2 modulate GSP-1 and GSP-2 activity in centriole duplication (Peel et al, 2017), and APE-1 directs localization of GSP-2 to epidermal cell-cell junctions (Beacham et al, 2022). However, the regulation of GSP-1 and GSP-2 during polarity establishment in one-cell embryos remains unknown.

Here we show that SDS-22 depletion results in lower GSP-1 and GSP-2 levels and in a partial rescue of the polarity defects caused by reduced PAR-2 phosphorylation in the *pkc-3(ne4246)* mutant at the semi-restrictive temperature (24 °C). These results establish that SDS-22 contributes to cell polarity by regulating GSP-1/-2 levels and are consistent with and complement the recent data in mammalian cells showing that SDS22 is important to control the stability of the PP1 phosphatase (Cao et al, 2024).

## Results

### Depletion of SDS-22 partially suppresses polarity defects of a temperature-sensitive *pkc-3* mutant

We have previously found that PP1 phosphatases GSP-1 and GSP-2 counteract the activity of the atypical protein kinase PKC-3 and allow cortical localization of PAR-2 in *C. elegans* embryos. However, the regulator(s) of GSP-1 and GSP-2 in polarity establishment is not known. In immunoprecipitations followed by mass spectrometry analysis of mNG::GSP-2 from embryos, we identified three interactors that have been previously reported as conserved PP1 regulators: SDS-22, APE-1 and SYZ-2(I-2) (Appendix Fig. S1; Datasets EV1 and EV2).

When the temperature-sensitive mutant *pkc-3(ne4246)* is grown at semi-permissive temperature, the residual PKC-3 activity is not sufficient to exclude PAR-2 from the anterior cortex. These embryos cannot establish polarity and die. Depletion of GSP-2 in this strain suppresses PAR-2 mislocalization and the resulting polarity defects, and rescues embryonic lethality (Calvi et al, 2022). We first asked whether depletion of any of these three identified regulators suppresses the embryonic lethality of *pkc-3(ne4246); gfp::par-2* embryos at the semi-permissive temperature of 24 °C (temperature used in all experiments with the *pkc-3(ne4246)* mutant, unless otherwise stated), similar to depletion of GSP-2. None of these three regulators was able to suppress the embryonic lethality (Fig. 1A). SDS-22 depletion alone resulted in 73.6% of embryonic lethality in the control strain, consistent with previous findings (Peel et al, 2017). We then investigated if any of these regulators was able to rescue the aberrant PAR-2 localization of *pkc-3(ne4246)* embryos. Neither APE-1 nor SYZ-2 depletion rescued the symmetric PAR-2 localization observed in *pkc-3(ne4246); gfp::par-2* embryos (Fig. EV1A). In contrast, depletion of SDS-22 resulted in PAR-2 localization being enriched in the posterior cortex in 87.5% of the one-cell stage embryos compared to *pkc-3(ne4246); gfp::par-2* control embryos where PAR-2 was around the entire cortex (Fig. 1B,C) and PAR-2 was localized to the P1 blastomere after the first cell-division (Movie EV1). Depletion of SDS-22 in otherwise wild-type *gfp::par-2* embryos did not impair anterior-posterior polarity, with PAR-2 restricted to the posterior cortex (Fig. 1B), indicating that the lethality caused by loss of SDS-22 is not the result of impaired PAR polarity establishment. However, similar to what has been observed in GSP-2 depleted embryos (Calvi et al, 2022), the PAR-2 domain was smaller (Figs. 1D and EV1B). SDS-22 depleted embryos also displayed a smaller size (Fig. EV1C).

We then investigated whether SDS-22 depletion, similar to GSP-2 depletion, rescued aberrant PAR-2 localization in another genetic background. When the PP1 binding motif in PAR-2 (164 RLFF 167) is optimized by a single amino acid substitution (L165V, resulting in RVFF, which, based on previous studies shows better binding to PP1 (Meiselbach et al, 2006)), PAR-2 abnormally occupies the anterior cortex in early embryos. This phenotype is rescued by depletion of GSP-2, suggesting that it depends on higher activity of GSP-2 on PAR-2 (Calvi et al, 2022). SDS-22 depletion also rescued the anterior PAR-2 domain observed in *gfp::par-2(L165V)* embryos as anterior PAR-2 was observed in only 45.5% of *gfp::par-2(L165V); sds-22(RNAi)* embryos compared to 100% *gfp::par-2(L165V); ctrl(RNAi)* embryos (Fig. EV2).

Polarity controls the posterior positioning of the mitotic spindle, which results in an asymmetric cell division producing a larger anterior AB cell and a smaller posterior P1 cell. In *gfp::par-2* embryos AB cell accounts for 56.5% of the total embryo length (Fig. 1E,F). In contrast, polarity defects in the *pkc-3(ne4246)* mutant lead to reduced cell size asymmetry with a smaller AB cell, occupying 54.2% of the embryo length (Fig. 1E,F). Consistent with the suppression of PAR-2 polarity defect, depletion of SDS-22 in *pkc-3(ne4246)* embryos resulted in a bigger AB cell (57.1% embryo length), comparable to the size of AB in control *gfp::par-2* embryos (Fig. 1E,F).

Our data show that depletion of SDS-22 results in a smaller PAR-2 domain, partially suppresses the polarity defects of a *pkc-3* temperature-sensitive strain and the aberrant PAR-2 localization observed in the PAR-2(L165V) mutant strain. As SDS-22 is a

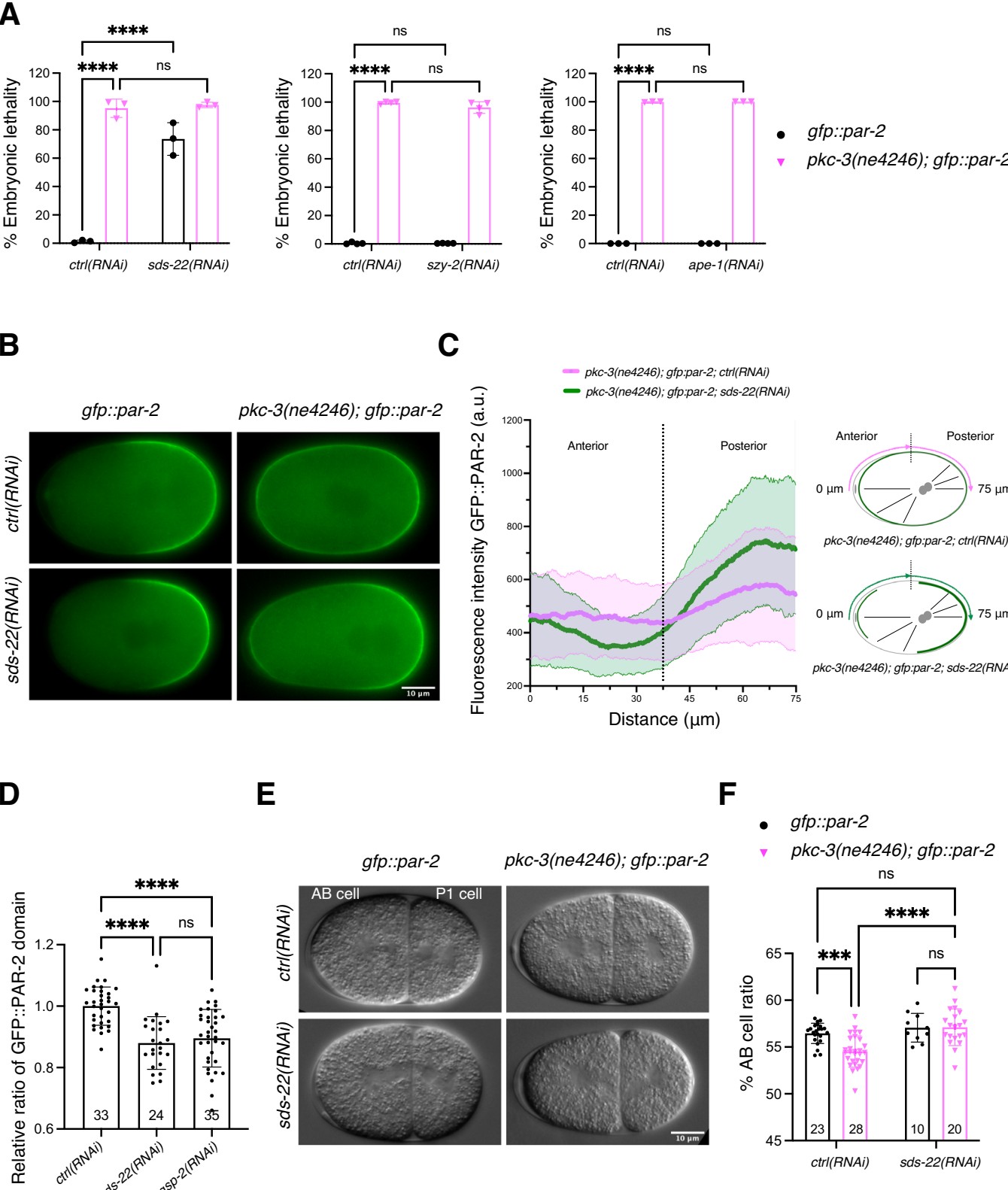

◄ **Figure 1.  SDS-22 depletion partially suppresses the polarity defects of the *pkc-3(ne4246)* mutant.**

(A) Embryonic lethality of *gfp::par-2* and *pkc-3(ne4246); gfp::par-2* embryos with the indicated depletions. The reported values correspond to the percentage of unhatched embryos over the total progeny (larvae and unhatched embryos) in all figures. *gfp::par-2; ctrl(RNAi)*, n = 4975, *gfp::par-2; sds-22(RNAi)*, n = 3923, *pkc-3(ne4246); gfp::par-2; ctrl(RNAi)*, n = 2247, and *pkc-3(ne4246); gfp::par-2; sds-22(RNAi)*, n = 1611. *gfp::par-2; szy-2(RNAi)*, n = 3550; *pkc-3(ne4246); gfp::par-2; szy-2(RNAi)*, n = 2588. *gfp::par-2; ape-1(RNAi)*, n = 720; *pkc-3(ne4246); gfp::par-2; ape-1(RNAi)*, n = 930. *sds-22(RNAi)* and *ape-1(RNAi)*, N = 3. *szy-2(RNAi)*, N = 4. The mean is shown and error bars indicate Standard Deviation (SD). The P values were determined using two-way ANOVA Tukey's multiple comparisons test. For plots of embryonic lethality, each dot inside the bar represents the average lethality of each independent experiment, unless otherwise indicated. (B) Representative frames of time-lapse videos of embryos at the pronuclear meeting stage: *gfp::par-2; ctrl(RNAi)*, n = 12 (PAR-2 is posterior in 100% of them), *gfp::par-2; sds-22(RNAi)*, n = 12 (100% posterior PAR-2), *pkc-3(ne4246); gfp::par-2; ctrl(RNAi)*, n = 42 (0% posterior PAR-2, PAR-2 is all around the cortex in all of them), and *pkc-3(ne4246); gfp::par-2; sds-22(RNAi)*, n = 48 (87.5% have PAR-2 enriched in the posterior cortex). N = 5. (C) GFP::PAR-2 intensity line profile along half of the embryo cortex starting from the anterior, as shown in the scheme on the right. *pkc-3(ne4246); gfp::par-2; ctrl(RNAi)*, n = 31, *pkc-3(ne4246); gfp::par-2; sds-22(RNAi)*, n = 32. N = 5. (D) Quantification of the GFP::PAR-2 domain size at pronuclear meeting in live embryos with the indicated RNAi conditions. Representative images are shown in Fig. EV1B. The images and quantification were based on a subset of embryos shown in Fig. 4. Sample size (*n*) is indicated inside the bars in the graph. N = 4. Mean is shown and error bars indicate SD. The P values were determined using one-way ANOVA. In D and F, each dot represents a single embryo. (E) Still images of two-cell embryos of the indicated genotypes taken from DIC time-lapse movies. (F) Quantification of the AB cell size as a percentage of the whole embryo size. Sample size (*n*) is indicated inside the bars in the graph. N = 3. Mean is shown and error bars indicate SD. The P values were determined using two-way ANOVA "Tukey's multiple comparisons test". For all the panels, RNA interference was performed by feeding. Scale bar is 10 μm, anterior is to the left and posterior to the right. In all plots, ns P > 0.05, *P < 0.05, **P < 0.01, ***P < 0.001, ****P < 0.0001. Exact P values are provided in Dataset EV3. *n* number of embryos analyzed, *N* number of independent experiments.

conserved PP1 regulator, our data suggest that SDS-22 positively regulates GSP-2.

## The conserved E153 of SDS-22 is important for SDS-22 interaction with GSP-1 and GSP-2

We have identified SDS-22 in immunoprecipitation with GSP-2 from embryos, confirming a previous finding that SDS-22 immunoprecipitates with both GSP-1 and GSP-2 from full worm lysates (Peel et al, 2017). We were able to reproduce the interaction of SDS-22 with both GSP-1 and GSP-2 in a two-hybrid assay (Fig. 2).

SDS-22 is a leucine-rich repeat (LRR) protein and does not contain the degenerate RLxF PP1 docking motif that has been found in most PP1 regulators. Instead, a conserved glutamic acid in the LRR domain is important for its interaction with the PP1 catalytic subunits (Fig. 2) (Ceulemans et al, 2002; Eiteneuer et al, 2014; Heroes et al, 2019). This site is conserved in *C. elegans* and we asked whether it is important for the interaction of SDS-22 with GSP-1 and GSP-2. Substitution of the glutamine E153 with Alanine (referred to as SDS-22(E153A)) abrogated the interaction with both GSP-1 and GSP-2 in the two-hybrid assays (Fig. 2).

These data identify the glutamic acid residue E153 in the LRR domain of SDS-22 as crucial for the interaction with GSP-1 and GSP-2, consistent with what has been demonstrated for the human orthologs (Ceulemans et al, 2002; Eiteneuer et al, 2014; Heroes et al, 2019).

## Polarity defects are partially rescued in *pkc-3(ne4246); sds-22(E153A); gfp::par-2* embryos

Our two-hybrid data show that SDS-22 interacts with GSP-1 and GSP-2 through the E153 residue. We introduced this substitution in vivo and asked whether *sds-22(E153A)* mutant embryos display a smaller PAR-2 domain, as observed in SDS-22 depleted embryos (Fig. 1B,D). PAR-2 was restricted to the posterior in *gfp::par-2; sds-22(E153A)* as in control embryos, but the length of the domain was decreased (Fig. 3A,B; Movie EV2), similar to what was observed in SDS-22 and GSP-2 depleted embryos. However, while SDS-22 depletion resulted in 73.6% embryonic lethality (Fig. 1A), *gfp::par-2; sds-22(E153A)* only exhibited 5.3% embryonic lethality compared

to the *gfp::par-2* strain (Fig. 3C). Because of this, we decided to test whether *sds-22(E153A)* was able to rescue the lethality and the polarity defects of *pkc-3(ne4246)* embryos. We introduced the E153A mutation in *pkc-3(ne4246); gfp::par-2* strain, let them grow at the semi-permissive temperature (24 °C) and quantified embryonic lethality. We found that the lethality of the embryos was not rescued at this temperature (Fig. 3D).

Decreasing the temperature to 22 °C reduced the embryonic lethality in the *pkc-3(ne4246); gfp::par-2* strain to about 76.8% (Fig. 3E). In these more permissive conditions, *pkc-3(ne4246); sds-22(E153A); gfp::par-2* exhibited only 26.4% embryonic lethality (Fig. 3E), indicating that the *sds-22(E153A)* mutation can partially rescue the lethality of *pkc-3(ne4246)* at this temperature.

When we looked at polarity we found that PAR-2, despite still weakly detectable at the anterior, was more enriched at the posterior cortex in the *pkc-3(ne4246); sds-22(E153A); gfp::par-2* mutant embryos compared to *pkc-3(ne4246); gfp::par-2* (Fig. 3F,G; Movie EV3). The rescue of embryonic lethality and polarity phenotypes was not due to a reduced expression level of SDS-22(E153A) because SDS-22(E153A)::GFP showed similar protein intensity as SDS-22::GFP (Appendix Fig. S2).

Our data show that a mutation that disrupts the interaction of SDS-22 with GSP-1 and GSP-2 in a two-hybrid assay, can lead to a partial rescue of the lethality and the polarity defect of *pkc-3(ne4246); gfp::par-2* embryos, suggesting that the interaction of SDS-22 with the PP1 phosphatases contributes to polarity establishment.

## Loss or mutation of SDS-22 results in hyper-phosphorylation of a PP1 substrate

In *C. elegans* embryos GSP-1 and GSP-2 redundantly regulate PAR-2 localization (Calvi et al, 2022). GSP-2 is the predominant PP1 phosphatase that allows for PAR-2's localization to the posterior cortex. Depletion of GSP-2 reduces the size of the PAR-2 domain in the *gfp::par-2* strain while depletion of GSP-1 does not lead to observable defects in PAR-2 domain size. Only when GSP-1 and GSP-2 are co-depleted, PAR-2 remains mostly cytoplasmic (Fig. 4A (Calvi et al, 2022)). If SDS-22 was essential to regulate the activity of both GSP-1 and GSP-2, its depletion should result in a loss of

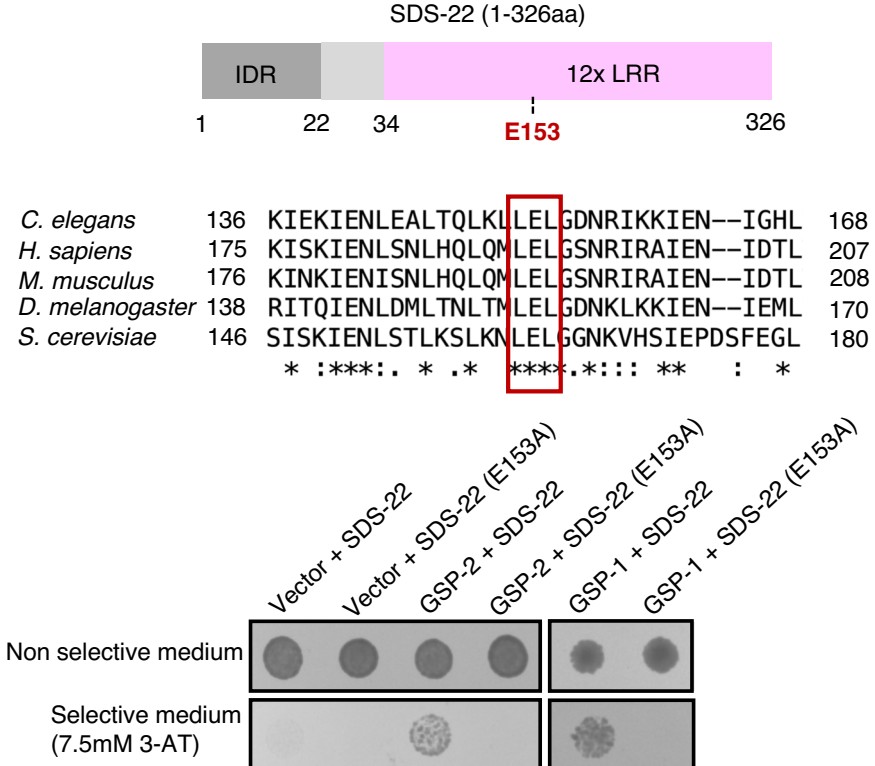

**Figure 2. SDS-22 interacts with GSP-1 and GSP-2 via the conserved E153 residue.**

Upper panel: schematic representation of the full-length (1–326 aa) SDS-22. IDR: intrinsically disordered region; LRR: leucine-rich repeats; E153: glutamic acid at amino acid residue 153. Middle panel: sequence alignment of one PP1 interacting region in SDS22 from *C. elegans* (Uniprot P45969), *H. sapiens* (Uniprot Q15435), *M. musculus* (Uniprot Q3UM45), *D. melanogaster* (Uniprot Q9VEK8), and *S. cerevisiae* (Uniprot P36047). The reported PP1-binding residue E192 of *H. sapiens* is present in *C. elegans* as E153, and this site is conserved in all species listed (marked by a red square). Figure adapted from alignment obtained with the Uniprot alignment tool (https://www.uniprot.org/align). Lower panel: yeast two-hybrid assays showing the interaction between SDS-22 (wild-type and the E153A mutant) and GSP-1 and GSP-2. Interaction between bait and prey results in yeast growth on selective medium.

PAR-2 cortical localization, similar to what is observed in the co-depletion of GSP-1 and GSP-2. As shown above, depletion of SDS-22 in the *gfp::par-2* strain did not lead to cytoplasmic localization of PAR-2 (Figs. 1B and 4A) but resulted in a smaller PAR-2 domain (Figs. 1D and 4B), as observed in the GSP-2 depletion.

These data suggest that SDS-22 regulates GSP-2 rather than GSP-1. If this was true, depletion of both SDS-22 and GSP-1 should result in PAR-2 remaining in the cytoplasm, similar to GSP-1 and GSP-2 co-depletion. We therefore co-depleted SDS-22 and GSP-1 and, as control, SDS-22 and GSP-2. The double depletion was efficient as shown in Appendix Fig. S3. In embryos co-depleted of SDS-22 and GSP-1, PAR-2 still formed a posterior cortical domain. Co-depletion of SDS-22 and GSP-2 resulted in an even more pronounced reduction of PAR-2 domain compared to the single depletion of SDS-22 (Fig. 4A,B), suggesting that SDS-22 plays a different role than simply activating GSP-2 (see below).

To understand the role of SDS-22 in the regulation of GSP-1 and GSP-2, we investigated whether SDS-22 depletion or mutation decreases PP1 activity. To quantify the activity of the PP1 phosphatase, we examined the phosphorylation levels of a well characterized substrate, histone H3, as previously described (Hsu et al, 2000; Qian et al, 2011). Histone H3 is phosphorylated at the Ser 10 residue by the Aurora B kinase in the early phases of mitosis

and it is dephosphorylated by PP1 at anaphase (Xin et al, 2020). We found that phospho-histone H3 (Ser10) levels on the chromosomes during anaphase were higher in SDS-22 depleted one-cell embryos (Fig. 5A) compared to control depletion, indicating that depletion of SDS-22 leads to deficient dephosphorylation of PP1 substrates.

Similar to the observation in SDS-22 depleted embryos (Fig. 5A), phospho-histone H3 (Ser10) levels were higher in the *sds-22(E153A)* mutant embryos (Fig. 5B), suggesting that the interaction between SDS-22 and GSP-1 and GSP-2 is important to maintain efficient dephosphorylation of a PP1 substrate.

To conclude, while our genetic data on PAR-2 cortical localization suggest that GSP-1/-2 activity does not strictly require SDS-22, depletion or mutation of SDS-22 does result in a reduced activity of these phosphatases, as shown by phospho-histone H3 (Ser10) levels. These data show that the regulation of GSP-1/-2 by SDS-22 is not specific to cell polarity.

## SDS-22 maintains GSP-1 and GSP-2 protein levels by protecting them from proteasomal degradation

Regulatory subunits of phosphatases can regulate their activity, localization or protein levels (Aggen et al, 2000; Verbinnen et al, 2017; Virshup and Shenolikar, 2009). Recent work in human cells

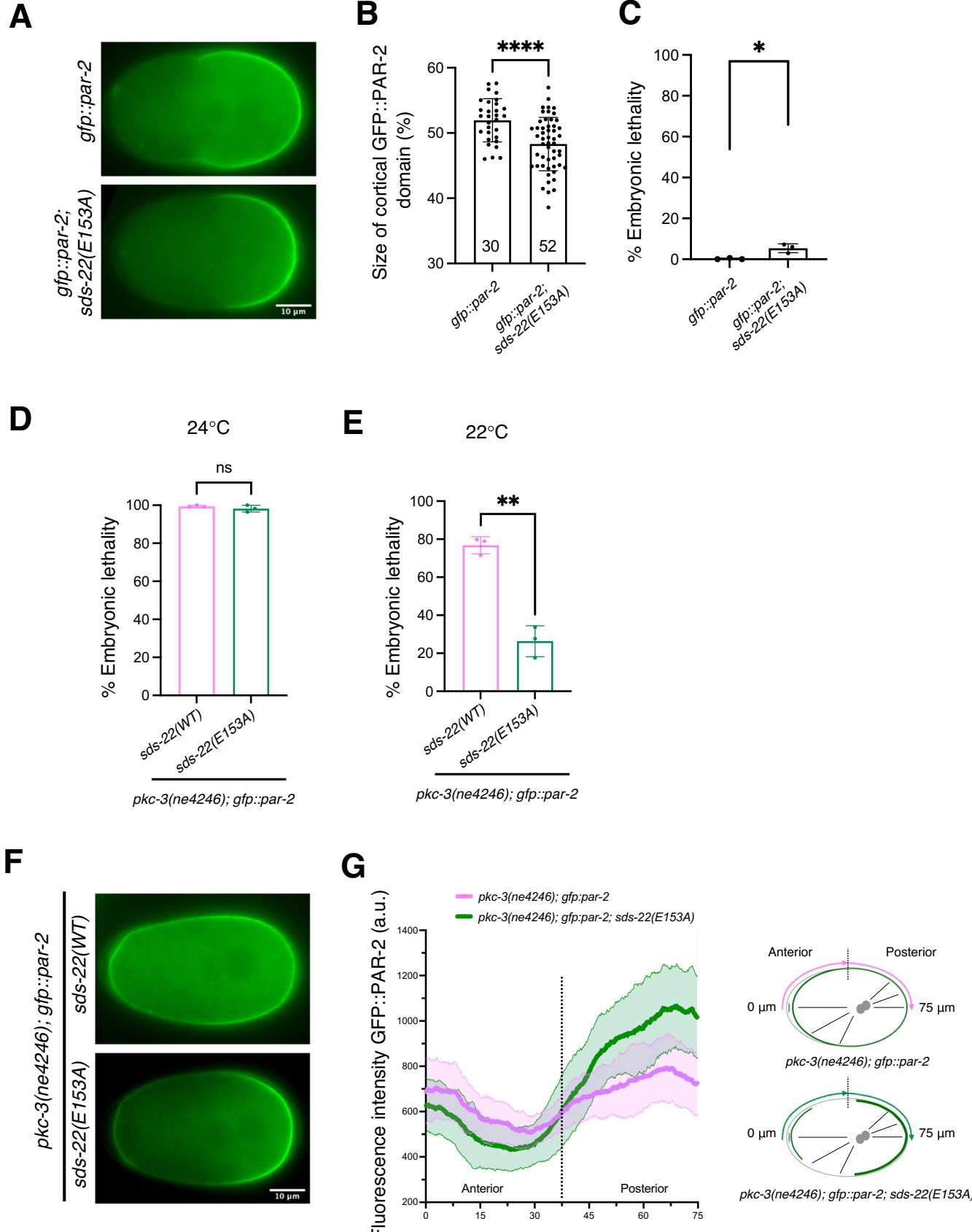

◄ **Figure 3. Mutation of SDS-22 E153 residue partially rescues polarity defects of *pkc-3(ne4246)* mutant.**

(A) Still images at the pronuclear meeting stage from time-lapse videos of *gfp::par-2* and *gfp::par-2; sds-22(E153A)*. (B) Quantification of the GFP::PAR-2 domain size at pronuclear meeting in live zygotes. Sample size (*n*) is indicated inside the bars in the graph, each dot represents a single embryo. N = 3. Mean is shown and error bars indicate SD. The *P* value was determined using two-tailed unpaired Student's *t* test. (C) Percentage of embryonic lethality of *gfp::par-2* and *gfp::par-2; sds-22(E153A)*. *gfp::par-2*, *n* = 757; *gfp::par-2; sds-22(E153A)*, *n* = 2644. N = 3. Mean is shown and error bars indicate SD. The *P* value was determined using two-tailed unpaired Student's *t* test. (D, E) Percentage of embryonic lethality of *pkc-3(ne4246); gfp::par-2* and *pkc-3(ne4246); gfp::par-2; sds-22(E153A)* at the semi-permissive temperature of 24 °C (D) and of 22 °C (E). For (D), *pkc-3(ne4246); gfp::par-2*, *n* = 569, and *pkc-3(ne4246); gfp::par-2; sds-22(E153A)*, *n* = 545. For (E), *pkc-3(ne4246); gfp::par-2*, *n* = 2171, and *pkc-3(ne4246); gfp::par-2; sds-22(E153A)* *n* = 1867. N = 3. Mean is shown and error bars indicate SD. The *P* value was determined using two-tailed unpaired Student's *t* test. (C–E) Dots represent the mean lethality of each independent experiment. (F) Still images from time-lapse videos of *pkc-3(ne4246); gfp::par-2* and *pkc-3(ne4246); gfp::par-2; sds-22(E153A)* embryos at the pronuclear meeting stage. Worms were grown at the semi-permissive temperature of 22 °C as in (E). (G) GFP::PAR-2 intensity line profile along half of the embryo cortex starting from the anterior, as shown in the scheme on the right. *pkc-3(ne4246); gfp::par-2* and *pkc-3(ne4246); gfp::par-2; sds-22(E153A)* *n* = 19 for both genotypes. N = 3. Worms were grown at the semi-permissive temperature of 22 °C as in (E, F). Scale bar is 10 μm, anterior is to the left and posterior to the right. In all plots, nsP > 0.05, *P < 0.05, **P < 0.01, ***P < 0.001, ****P < 0.0001. Exact *P* values are provided in Dataset EV3. *n* number of embryos analyzed, *N* number of independent experiments.

and in vitro has shown that SDS22 coordinates the assembly of the PP1 holoenzyme (Cao et al, 2024; Kueck et al, 2024). To understand how SDS-22 contributes to the function of GSP-1 and GSP-2 in *C. elegans*, we investigated whether SDS-22 regulated the localization or levels of GSP-1 and GSP-2. We depleted SDS-22 in the *mNG::gsp-2* and *gfp::gsp-1* strains and observed that the localization of GSP-2 and GSP-1 was not altered by SDS-22 depletion (Fig. 6A). However, the intensity of mNG::GSP-2 and GFP::GSP-1 was reduced compared to the control strains (Fig. 6B,C). As fusing proteins to GFP can result in their destabilization (Sokolovski et al, 2015), we independently verified this result using an OLLAS::GSP-2 fusion and antibodies to GSP-1. Both OLLAS::GSP-2 and GSP-1 levels were reduced as observed by western blot analysis in SDS-22 depleted embryos (Fig. 6D–F). Depletion of SDS-22 did not result in reduced *gsp-1* and *gsp-2* mRNA levels (Fig. EV3A). SDS-22 is therefore essential to maintain proper levels of GSP-1 and GSP-2 in *C. elegans* embryos.

We next asked whether SDS-22 maintains GSP-1 and GSP-2 levels in vivo via their physical interaction with the proteins. We generated strains in which the E153 site of SDS-22 is mutated in the genetic background of *mNG::gsp-2*, *ollas::gsp-2* and *gfp::gsp-1* separately. *sds-22(E153A)* mutation caused 99.1% embryonic lethality in *mNG::gsp-2*, 37.0% embryonic lethality in *ollas::gsp-2* and 32.3% in *gfp::gsp-1* (Fig. EV3B–D). These data suggest that tagged GSP-2 and GSP-1 strains are hypomorph even though there are no observable growth defects nor embryonic lethality in these strains. Because of the high embryonic lethality of *mNG::gsp-2; sds-22(E153A)* homozygous mutant, we maintained the strain as a *sds-22(E153A)* heterozygote (see methods). Consistent with the phenotypes of *sds-22(RNAi)*, SDS-22(E153A) mutation led to a reduction of the mNG::GSP-2 (Fig. 6G,H) and GFP::GSP-1 intensity (Fig. 6G,I). Consistently, both OLLAS::GSP-2 and GSP-1 protein levels were reduced in *sds-22(E153A)* embryos (Fig. 6J–L), similar to what we observed in SDS-22 depleted embryos.

We then investigated whether the levels of GSP-1 and/or GSP-2 affected the levels of SDS-22. We performed single depletion and co-depletion of GSP-1 and GSP-2 in *sds-22::gfp* strain. We found that the levels of SDS-22 were weakly reduced by single depletion of GSP-1 or GSP-2 (around 8% of reduction in both conditions) but were un-affected by co-depletion of both GSP-1 and GSP-2 (Appendix Fig. S4A,B).

Our data show that SDS-22 depletion results in reduction of GSP-1 and GSP-2 levels. Embryos co-depleted of SDS-22 and either

GSP-1 or GSP-2 still polarized (Fig. 4), suggesting that reduced levels of the phosphatase catalytic subunits are sufficient to maintain polarity. To test this, we depleted GSP-1 by injecting dsRNA and co-depleted GSP-2 by feeding using diluted bacteria to obtain a partial depletion. With this method, GSP-1 was about 20% of the initial levels and GSP-2 was about 30% (Fig. EV4A,B). PAR-2 remained localized at the posterior cortex, indicating that this degree of reduction of GSP-1/-2 levels was not sufficient to abrogate PAR-2 cortical localization, as it is observed when dsRNA to both subunits is injected (Fig. EV4C). As in the GSP-2 depletion alone, the PAR-2 domain was smaller (Fig. EV4C,D). These findings (Figs. 4 and EV4) suggest that low amounts of GSP-1 and GSP-2 are sufficient to polarize PAR-2.

In budding yeast, when Sds22 is mutated, PP1-Glc7 is prone to misfold and form aggregates with heat-shock proteins, which are cleared by the proteasome (Cheng and Chen, 2015). Additionally, in human cells, SDS22 stabilizes nascent PP1 (Cao et al, 2024). We hypothesized that GSP-1 and GSP-2 are degraded by the proteasome when SDS-22 is depleted or mutated. To test this hypothesis, we reduced proteasomal activity in SDS-22-depleted or mutated embryos by depleting the proteasomal subunits RPN-6.1 or RPN-7, which localize to the 19S regulatory complex and are essential for the proteolytic activity (Fernando et al, 2022; Papaevgeniou and Chondrogianni, 2014). Since depletion of these subunits results in worms with very little to no progeny (Fernando et al, 2022), we analyzed the protein intensity of GSP-1 and GSP-2 in the -1 and -2 oocytes in the germline and investigated whether reducing proteasomal activity would restore GSP-1 and GSP-2 levels.

When RPN-6.1 or RPN-7 were depleted, there was an increase of GSP-1 and GSP-2 intensity both in the cytoplasm and in the nucleus of the oocytes (Fig. 7A–F; Appendix Fig. S5), indicating that GSP-1 and GSP-2 are subject to proteasomal degradation in the germline of *C. elegans*. Consistent with the observation in embryos (Fig. 6A), both cytoplasmic and nuclear GSP-2 levels in the germline were reduced by about 50% after depletion of SDS-22 (Fig. 7B,C). When SDS-22 was depleted together with RPN-6.1 or RPN-7, the levels of GSP-2 were restored (Fig. 7A–C; Appendix Fig. S5A). The efficiency of SDS-22 depletion in the RPN-6.1 or RPN-7 co-depletion was verified (Appendix Fig. S6). In *sds-22(RNAi)* oocytes, the levels of GFP::GSP-1 exhibited a smaller decrease (17.4%) in the nucleus and an even smaller reduction in the cytoplasm (8.8%, n.s.), which was not statistically significant (Fig. 7D–F). The reduced nuclear and cytoplasmic intensities of GSP-1 following SDS-22 depletion were

## A

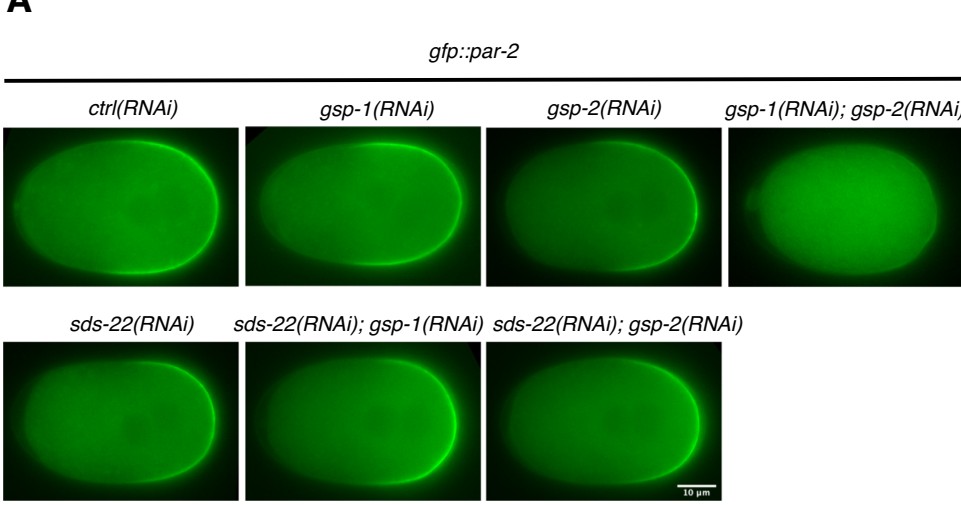

## B

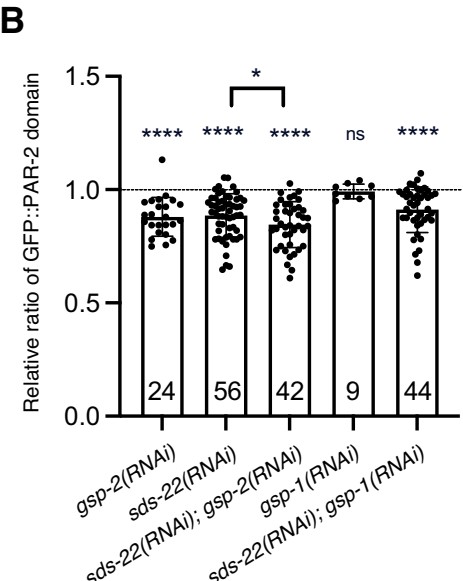

**Figure 4. SDS-22 does not genetically regulate GSP-1, GSP-2 or both.**

(A) Still images from time-lapse videos of *gfp::par-2* zygotes at pronuclear meeting with the indicated depletions. RNA interference was performed by injection. Scale bars is 10 μm, anterior is to the left and posterior to the right. (B) Quantification of the relative ratio of GFP::PAR-2 size domain with indicated RNAi conditions. Each condition was compared to *ctrl(RNAi)* treatment within the same experiment. The *P* value was determined using two-tailed unpaired Student's *t* test. Sample size (*n*) is indicated inside the bars in the graph. N = 4–5. Mean is shown and error bars indicate SD. nsP > 0.05, *P < 0.05, **P < 0.01, ***P < 0.001, ****P < 0.0001. Exact *P* values are provided in Dataset EV3. *n* number of embryos analyzed, *N* number of independent experiments.

also rescued by co-depletion of RPN-6.1 or RPN-7 (Fig. 7D–F; Appendix Fig. S5B), similar to the recovery of GSP-2 levels (Fig. 7B,C). To further test the role of SDS-22 on GSP-1 stability, we measured GFP::GSP-1 levels in the -1 and -2 oocytes in the *gfp::gsp-1; sds-22(E153A)* strain. In the germline of *gfp::gsp-1; sds-22(E153A)* worms, GSP-1 levels were reduced (28.8% in the cytoplasm; 31.0% in the nucleus) (Fig. 7G–I). The stronger reduction in the E153A mutant compared to the SDS-22 depletion could be due to the fact that RNA interference of *sds-22* was performed from the L4 larval stage.

Consistent with this, depletion of SDS-22 starting at the L1 larval stage resulted in a stronger GFP::GSP-1 reduction, comparable to the one observed in the E153A mutant (Appendix Fig. S7). Depletion of RPN-6.1 or RPN-7 in the *gfp::gsp-1; sds-22(E153A)* increased the levels of GSP-1 (Fig. 7G–I; Appendix Fig. S5C), supporting our hypothesis that SDS-22, through its interaction with GSP-1, protects GSP-1 from proteasome mediated degradation. Since we use the embryonic lethality phenotype of the *mNG::gsp-2; sds-22(E153A)* strain to recognize the homozygote *sds-22(E153A)*, this precluded the

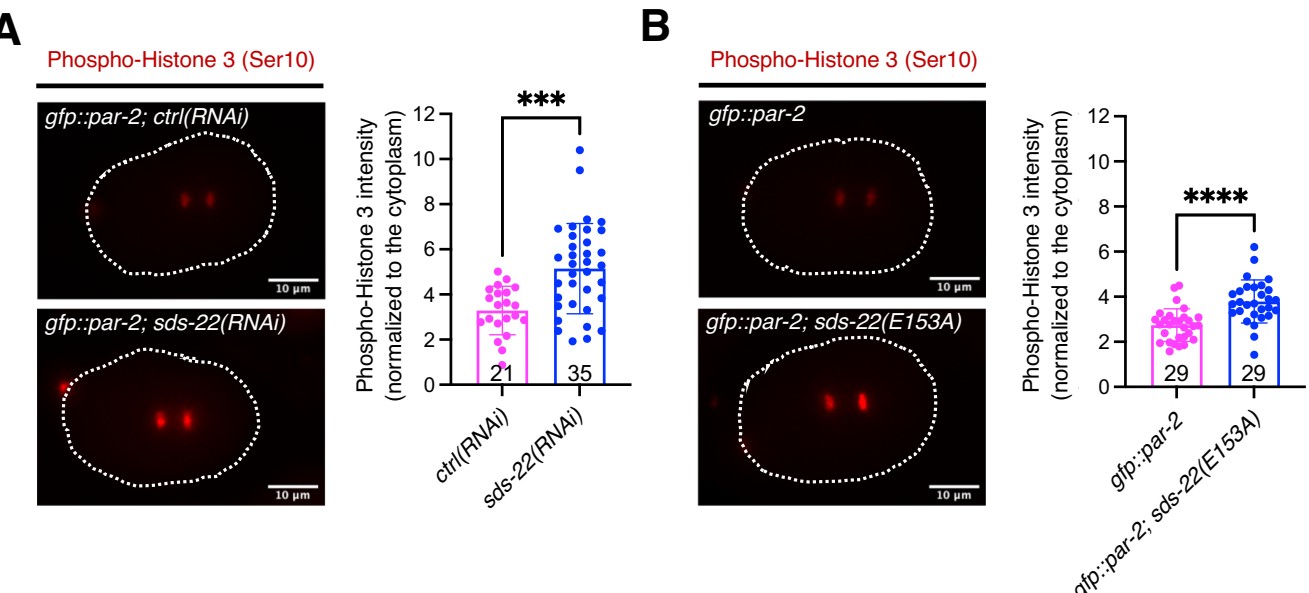

**Figure 5.  Loss or mutation of SDS-22 decreases GSP-1 and GSP-2 activity.**

(A) Left, images of a *gfp::par-2; ctrl(RNAi)* anaphase embryo and a *gfp::par-2; sds-22(RNAi)* anaphase embryo stained for phospho-histone H3 (Ser10) (in red). Right, quantifications. (B) Left, images of a *gfp::par-2* and a *gfp::par-2; sds-22(E153A)* embryo at anaphase stained for phospho-histone H3(Ser10) (in red). Right, quantifications. For both A and B, sample size (n) is indicated inside the bars in the graph, each dot represents an embryo. N = 4. Phospho-histone H3 (Ser10) intensity was normalized to cytoplasmic levels. Mean is shown and error bars indicate SD. The P values were determined using two-tailed unpaired Student's t test. ***P < 0.001, ****P < 0.0001. Exact P values are provided in Dataset EV3. n number of embryos analyzed, N number of independent experiments. The dashed lines indicate the outline of embryos. RNA interference was performed by feeding. Scale bar is 10 μm.

possibility to analyze the germlines of homozygote *mNG::gsp-2; sds-22(E153A)* worms depleted of RNP-6.1 or RPN-7, as these worms do not have progeny (Fernando et al, 2022) and we therefore cannot distinguish the *sds-22(E153A)* homozygote from the *sds-22(E153A)* heterozygote (see "Methods" for details).

To further investigate if reducing proteasomal activity restores protein levels of GSP-1/-2 in the embryos, we depleted RPN-12, a subunit that leads to less proteasomal activity reduction when depleted, compared to RPN-6.1 and RPN-7, but worms remained fertile (Fernando et al, 2022). Depletion of RPN-12 in the GFP::GSP-1 and in mNG::GSP-2 strains resulted in increased levels of the two proteins (Fig. EV5A,B). We then performed RPN-12 depletion in the *gfp::gsp-1; sds-22(E153A)* which can be maintained as homozygote and found that the levels of GFP::GSP-1 were restored in this strain comparable to the levels observed in the control (Fig. EV5C,D). We then asked whether the smaller PAR-2 domain observed in the *sds-22(E153A)* mutant embryos can be rescued by depleting this proteasomal subunit. We found that depletion of RPN-12 in the *gfp::par-2* strain resulted in a longer PAR-2 domain. In *gfp::par-2; sds-22(E153A)* RPN-12 depleted embryos the PAR-2 domain size was indistinguishable from the *gfp::par-2* strain (Fig. EV5E,F), suggesting that RPN-12 may regulate PAR-2 length by maintaining the proper PP1 phosphatase levels. This is consistent with our previous data showing that RPN-12 depletion can rescue the lethality of a *par-2(ts)* strain (Labbe et al, 2006).

In summary, these data suggest that SDS-22 is important to maintain the levels of GSP-1 and GSP-2 by protecting them from proteasome mediated degradation.

## Discussion

Cell polarity establishment in *C. elegans* embryos is crucial for subsequent asymmetric cell divisions and development. Previous studies have shown that the balanced activity of the kinase PKC-3 and the PP1 phosphatases GSP-1 and GSP-2 is essential for the posterior cortical localization of PAR-2 (Calvi et al, 2022; Hao et al, 2006; Motegi et al, 2011). GSP-1/-2 are PP1 catalytic subunits that rely on regulatory subunits to direct their cellular localization or facilitate their activity within a functional holoenzyme. Here we find that a conserved PP1 regulator, SDS-22, when depleted, results in a smaller PAR-2 domain and can partially rescue the polarity defects of a *pkc-3(ne4246)* mutant. We demonstrate that SDS-22 contributes to the activity of GSP-1/-2 by maintaining their protein levels. These data suggest that the role of SDS-22 in polarity is indirect via the regulation of GSP-1/-2 levels. In support of this, SDS-22 depletion results in broader GSP-1/-2 dependent phenotypes such as increased Phospho-H3 (Ser10) (Fig. 5) and centriole duplication defects in later-stage embryos (Peel et al, 2017).

SDS-22 is a conserved protein previously shown to be important for cell polarity. In *Drosophila*, Sds22 is essential for maintaining the morphology and apical-basal polarity of follicle epithelial cells (Grusche et al, 2009) and it facilitates the cortical localization of the lethal giant larvae (Lgl) protein in these cells (Moreira et al, 2019). In mammalian and yeast cells, SDS22 is required for mitotic progression (Duan et al, 2016; Posch et al, 2010; Wurzenberger et al, 2012). Even though SDS22 appears as a conserved regulator of PP1 phosphatases, the exact mechanism by which SDS22 regulates PP1 activity was still debated.

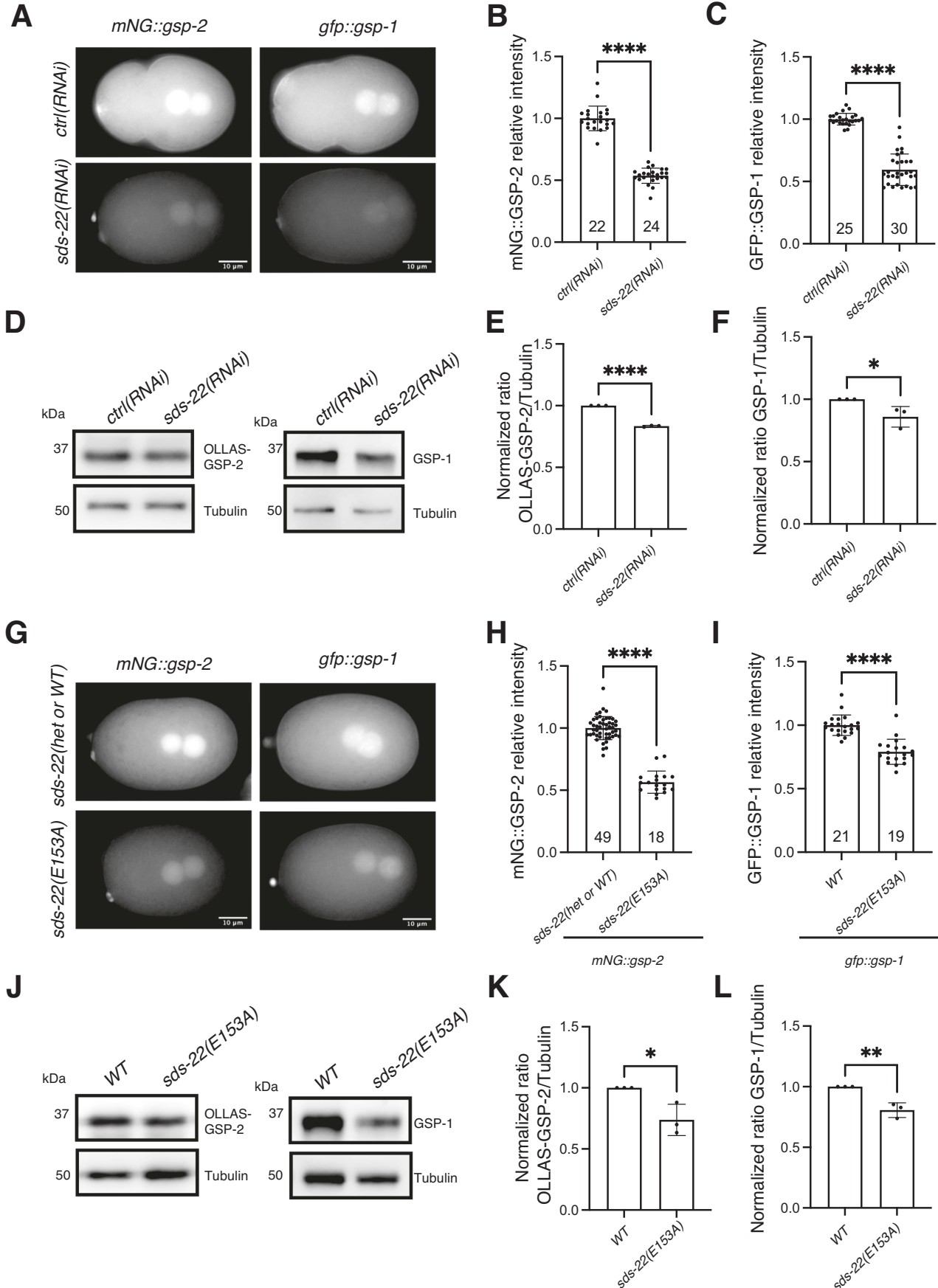

**Figure 6. SDS-22 depletion or mutation causes a reduction of GSP-1 and GSP-2 protein levels.**

(A) Representative images of *mNG::gsp-2* and *gfp::gsp-1* embryos in *ctrl(RNAi)* and *sds-22(RNAi)*. (B, C) Quantification of relative mNG::GSP-2 (B) and GFP::GSP-1 (C) levels in *sds-22(RNAi)* normalized to *ctrl(RNAi)*. Mean is shown and error bars indicate SD. $N = 4$. ****$P < 0.0001$. (D) Western Blot of embryonic extracts from *ollas::gsp-2* and *N2* in *ctrl(RNAi)* and *sds-22(RNAi)* embryos. Tubulin is used as a loading control. (E, F) Quantification of fold change of OLLAS::GSP-2 (E) and GSP-1 (F) normalized to Tubulin levels. $N = 3$. (G) Representative images of *mNG::gsp-2; sds-22(E153A/+ or +/+*, a mixture of wild-type and heterozygous of the E153A mutant, see methods), *mNG::gsp-2; sds-22(E153A)*, *gfp::gsp-1* and *gfp::gsp-1; sds-22(E153A)*. (H, I) Quantification of relative mNG::GSP-2 (H) and GFP::GSP-1 (I) levels with SDS-22(E153A) mutation normalized to wild-type. *mNG::gsp-2*, $N = 3$. *gfp::gsp-1*, $N = 4$. ****$P < 0.0001$. (J) Western Blot of embryonic extracts from *ollas::gsp-2, ollas::gsp-2; sds-22(E153A)*, *N2* and *sds-22(E153A)* embryos. Tubulin is used as a loading control. (K, L) Quantification of fold change of OLLAS::GSP-2 (K) and GSP-1 (L) normalized to Tubulin levels. $N = 3$. (A–F) RNA interference was performed by feeding. (B, C, E, F, H, I, K, L) Mean is shown, error bars indicate SD, the $P$ values were determined using two-tailed unpaired Student's *t* test. Exact $P$ values are provided in Dataset EV3. *n* number of embryos analyzed, *N* number of independent experiments. Dots in (B, C, H, I) represent individual embryo measurements and sample size (*n*) is indicated inside the bars in the graph; each dot in (E, F, K, L) represent the measurement of each independent experiment.

Some studies have shown that SDS22 positively contributes to PP1 activity in mitosis (MacKelvie et al, 1995; Stone et al, 1993) and plays a role in localizing PP1 and in protein folding of the PP1 complex (Cheng and Chen, 2015; Peggie et al, 2002). However, other studies suggested that SDS22 inhibits PP1 activity in vitro (Lesage et al, 2007). Two recent studies from the Bollen and Meyer groups show that SDS22 plays a key role in the biogenesis of PP1 holoenzymes (Cao et al, 2024; Kueck et al, 2024). On the one hand, SDS22 stabilizes newly translated PP1; on the other hand, SDS22 and Inhibitor 3 (I3) bind to PP1, forming a ternary complex that inhibits PP1 activity. This inhibitory complex is dissociated by the AAA ATPase p97/Valosin, allowing PP1 to bind canonical regulatory subunits to form a functional holoenzyme. Given that SDS-22 both stabilizes PP1 levels and inhibits its activity, this dual role clarifies the apparent contradiction: while SDS-22 is essential for PP1 activity in vivo (because it is essential for the biogenesis/stability), it inhibits PP1 activity in vitro (as it needs to be removed to have an active PP1), and it is removed by p97/Valosin in vivo resulting in active PP1 (Cao et al, 2024; Kueck et al, 2024).

Our data in *C. elegans* support and complement these studies. We show that GSP-1/-2 activity does not strictly require SDS-22, since depletion of SDS-22 does not abolish polarity, as observed in the co-depletion of GSP-1 and GSP-2 (Calvi et al, 2022). Embryos co-depleted of SDS-22 and GSP-1 or of SDS-22 and GSP-2, are also able to polarize, indicating that SDS-22 plays a different role than simply activating GSP-1, GSP-2 or both. We find that in *C. elegans* SDS-22 is required to maintain the wild-type levels of both GSP-1 and GSP-2. We propose that the reduced levels of GSP-2 explain why loss of SDS-22 partially rescues PAR-2 defect in *pkc-3(ne4246)* mutant allele on one side (Fig. 1B). On the other side, since embryos still have GSP-1 and GSP-2 proteins they are able to polarize after depletion of SDS-22, or co-depletion of SDS-22 with GSP-1 or SDS-22 with GSP-2 in *gfp::par-2* embryos (Fig. 4). In these conditions, depletion of SDS-22, whether alone or in conjunction with GSP-1 or GSP-2, only results in a partial loss of the PP1 subunits GSP-1 and/or GSP-2 and in a reduction in catalytic activity, which is not sufficient to abrogate polarity, as also shown by partial depletion of GSP-1 and GSP-2 (Fig. EV4). Consistent with reduced GSP-2 levels, SDS-22 depleted or E153A mutant embryos also have a smaller PAR-2 domain. However, since these embryos also show reduced cortical ruffling (Movies EV1 and 2) and are smaller (Fig. EV1C) we cannot exclude that these two phenotypes also contribute to the smaller size of the PAR-2 domain.

How SDS-22 maintains PP1 levels has not been fully clarified. In budding yeast, it has been reported that when Sds22 is mutated, PP1/Glc7 forms aggregates in the nucleus, despite the overall protein level remaining unaltered (Peggie et al, 2002). These aggregates co-localize with heat-shock proteins that require clearance by the proteasome (Cheng and Chen, 2015). Co-expression of PP1 with chaperones GroEL/ES(HSP-60) in bacteria (Peti et al, 2013) or with SDS22 in mammalian cells (Choy et al, 2019) increases PP1 yield and solubility. This suggests that in the absence of SDS22, PP1 is prone to mislocalization or misfolding, leading to aggregation. Interestingly, Peel et al found SDS-22 co-immunoprecipitated with TCP-1 and HSP-60 (Peel et al, 2017), both members of the chaperonin family critical for protein folding (Hayer-Hartl et al, 2016; Kubota et al, 1995). Collectively, these findings suggest that, by facilitating protein folding, SDS22 protects PP1 from degradation. In the *C. elegans* germline and embryos GSP-1/-2 do not form visible aggregates when SDS-22 is depleted but they undergo degradation in a proteasome-dependent manner (Fig. 7). Together with previous findings (Cao et al, 2024; Cheng and Chen, 2015), our data suggest a role of SDS-22 in protecting PP1s from proteasomal degradation.

Recent studies of the Bollen and Meyer laboratories (Cao et al, 2024; Kueck et al, 2024) suggest that there is a canonical regulator of GSP-1 and GSP-2 in embryonic polarity that we have not yet identified. In the immunoprecipitations of GSP-2 from embryos we have identified APE-1 and SZY-2(I-2). APE-1 has been shown to direct GSP-2, but not GSP-1, localization to epidermal junctions and modulate GSP-2 activity (Beacham et al, 2022). SZY-2(I-2) regulates both GSP-1/-2 in centriole duplication from the four-cell stage of *C. elegans* embryos (Peel et al, 2017). However, our data suggest that neither APE-1 nor SZY-2 functions as canonical PP1 regulators in PAR-2's regulation, as depletion of either did not rescue the polarity defects of *pkc-3(ne4246)* mutant (Figs. 1A and EV1A). We cannot exclude the possibility that SDS-22 may also function as a canonical regulator of the PP1 holoenzymes in controlling PAR-2 localization in *C. elegans* embryos. Further investigation is required to determine whether a yet-to-be-identified regulator of the PP1 holoenzyme is responsible for loading PAR-2 to the posterior cortex.

Another open question is whether and how the activity of PP1 catalytic subunits is asymmetrically modulated in the one-cell embryo. Our previous work (Calvi et al, 2022) suggests that the activity of GSP-1/-2 might be asymmetrically restricted by the anterior-enriched polo-like kinase PLK-1, which consequently gives rise to asymmetric posterior localization of PAR-2. One possibility is that PLK-1 directly phosphorylates and regulates the activity of GSP-1 and/or GSP-2. Consistent with this possibility, a recent study shows that in *Drosophila* cells POLO kinase phosphorylates and inhibits PP1 (Moura et al, 2025). Another possibility is that

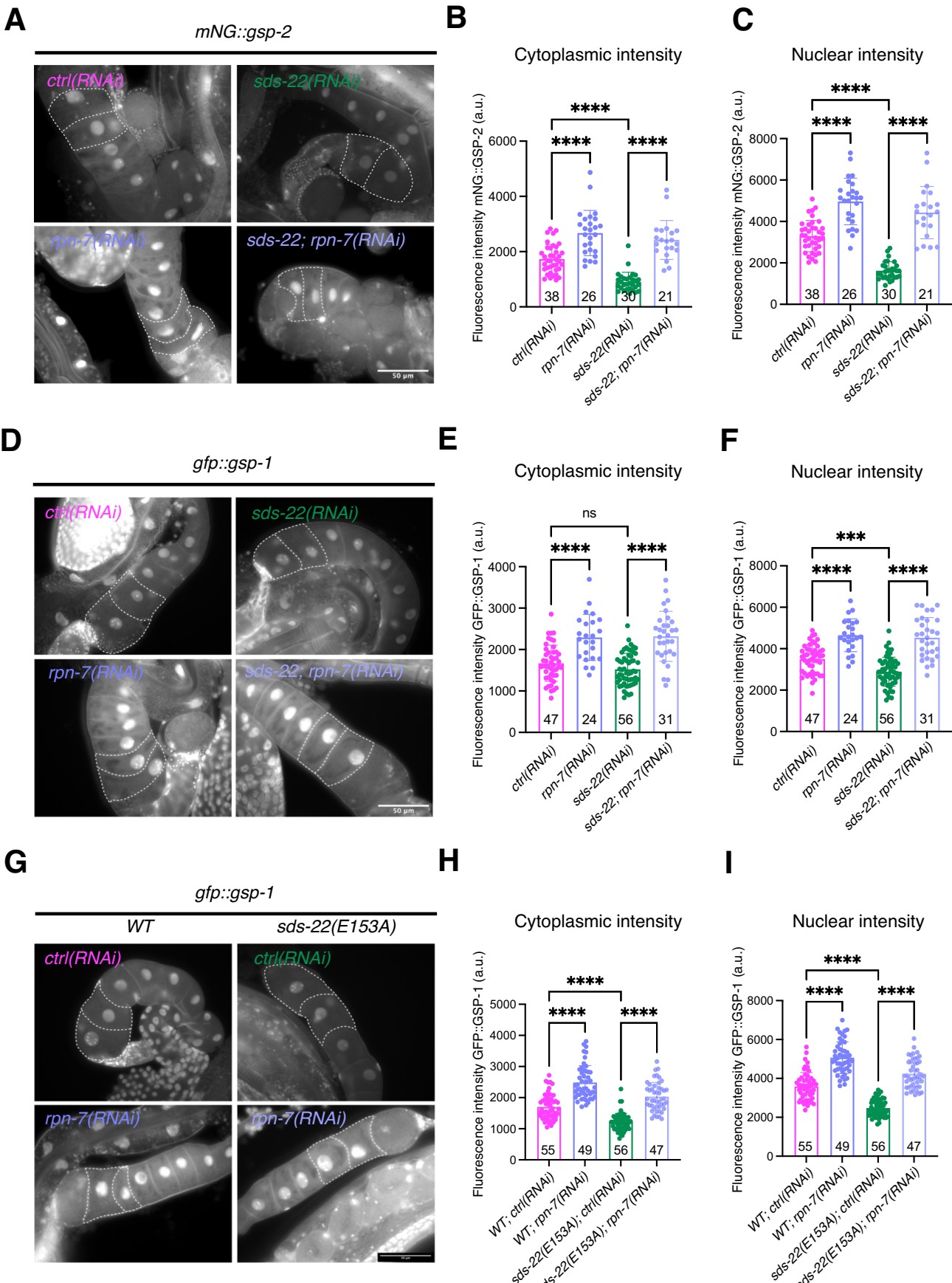

◄ **Figure 7. SDS-22 protects GSP-1 and GSP-2 from proteasomal degradation.**

(A) Representative images of *mNG::gsp-2* germlines in *ctrl(RNAi)*, *sds-22(RNAi)*, *rpn-7(RNAi)* and *sds-22(RNAi); rpn-7(RNAi)*. (B, C) Quantification of mNG::GSP-2 intensity levels in the cytoplasm (B) and nucleus (C) of -1 and -2 oocytes. N = 3. (D) Representative images of *gfp::gsp-1* germlines in *ctrl(RNAi)*, *sds-22(RNAi)*, *rpn-7(RNAi)* and *sds-22(RNAi); rpn-7(RNAi)*. (E, F) Quantification of GFP::GSP-1 intensity levels in the cytoplasm (E) and nucleus (F) of -1 and -2 oocytes. N = 3. (G) Representative midsection images of *gfp::gsp-1*, *gfp::gsp-1;sds-22(E153A)* germlines in *ctrl(RNAi)* and *rpn-7(RNAi)*. (H, I) Quantification of GFP::GSP-1 intensity levels in the cytoplasm (H) and nucleus (I) of -1 and -2 oocytes. N = 3. For (B, C, E, F, H, I), mean is shown and error bars indicate SD. Each dot represents the measurement of one germline. The P values were determined using one-way ANOVA "Tukey's multiple comparisons test". Sample size (n) is indicated inside the bars in the graph. For all panels, RNA interference was performed by feeding. The scale bars are 50 µm. In all plots, nsP > 0.05, *P < 0.05, **P < 0.01, ***P < 0.001, ****P < 0.0001. Exact P values are provided in Dataset EV3. n number of embryos analyzed, N number of independent experiments. Imaging of the *ctrl(RNAi)* and *sds-22(RNAi)* has been performed in parallel to the imaging of Appendix Fig. S5, and it is therefore the same data.

PLK-1 regulates the regulatory subunit(s) of GSP-1/-2. Notably, in mammalian cells, SDS22 is phosphorylated and inhibited by Plk1 during mitosis (Duan et al, 2016). Further studies will be required to understand whether additional pathways regulate the activity of this phosphatase during polarity establishment.

# Methods

## Worm strains

Worms (listed in the "Reagents and tools table") were maintained on NGM plates seeded with OP50 bacteria using standard methods (Brenner, 1974). The strains containing the temperature sensitive *pkc-3(ne4246)* mutation, *pkc-3(ne4246); gfp::par-2* (Ng et al, 2022) and *pkc-3(ne4246); gfp::par-2; sds-22(E153A)*, were kept at 15 °C. All the other strains were maintained at 20 °C.

Mutant strains were generated using CRISPR/Cas9 technology as described (Arribere et al, 2014). The single guide RNAs, repair templates used to generate the mutants, PCR primers used to amplify the sequence with mutation, as well as enzymes used for the screening, are listed in "Reagents and tools table". For the strain ZU341 (*gfp::par-2; sds-22(E153A)*), two independent isolates were analyzed. The ZU362 *mNG::gsp-2; sds-22(E153A)* was maintained as a *sds-22(E153A)* heterozygote. *mNG::gsp-2; sds-22(E153A)* adult worms are viable but produce 100% dead progeny, indicating that this mutation behaves as a maternal effect lethal in the *mNG::gsp-2* background.

### Reagents and tools table

| Reagent/resource | Reference or source | Identifier or catalog number |
|---|---|---|
| **Experimental models** | | |
| *C. elegans:* N2 wild-type (Bristol) | Caenorhabditis Genetics Center | N2 |
| *par-2(it328[gfp::par-2]) III* | Caenorhabditis Genetics Center | KK1273 |
| *pkc-3(ne4246) II; par-2(it328[gfp::par-2]) III* | Ng et al, 2022 | NWG0124 |
| *par-2(it328[gfp::par-2]) (L165V) III* | Calvi et al, 2022 | ZU316 |
| *gsp-2(it151[mNeongreen::gsp-2]) III* | Courtesy of Dhanya Cheerambathur | OD4092 |
| *gsp-1(it94[gfp::gsp-1]) III* | Courtesy of Dhanya Cheerambathur | OD3350 |
| *sds-22::gfp II* | Courtesy of Erik Griffin | EGD762 |
| *sds-22(E153A) II; gfp::par-2 III;* | This study | ZU341 |
| *sds-22(E153A) II; gfp::gsp-1 V* | This study | ZU351 |
| *sds-22(WT) II; gfp::gsp-1 V* | This study | ZU352 |
| *sds-22(E153A) II* | This study | ZU358 |
| *sds-22(E153A)::gfp II* | This study | ZU361 |
| *sds-22(E153A) hetro II; mNG::gsp-2 III* | This study | ZU362 |
| *pkc-3(ne4246) II; sds-22(E153A) II; par-2(it328[gfp::par-2]) III;* | This study, from Sunybiotech | PHX8659 |
| *ollas::gsp-2 III (gsp-2(syb10502))* | This study, from Sunybiotech | PHX10502 |
| *sds-22(E153A) hetro II; ollas::gsp-2 III* | This study | ZU381 |
| *ollas::gsp-2 III* | This study | ZU382 |
| **Recombinant DNA** | | |
| pDONR201 | Invitrogen | 11798-014 |
| pDEST32 | Invitrogen | 10043562 |

| Reagent/resource | Reference or source | Identifier or catalog number |
|---|---|---|
| pDEST22 | Invitrogen | 10043562 |
| pDONR201-SDS-22 cDNA (f.l.) | This study | pMG1584 |
| pDEST32-SDS-22 cDNA (f.l.) | This study | pMG1585 |
| pDONR201-SDS-22 cDNA (f.l., E153A) | This study | pMG1588 |
| pDEST32-SDS-22 cDNA (f.l., E153A) | This study | pMG1589 |
| pDEST22-GSP-1 cDNA (f.l.) | This study | pMG1593 |
| pDEST22-GSP-2 cDNA (f.l.) | This study | pMG1260 |
| pRB1017 | Addgene | Plasmid #59936 |
| pDEST-L4440 | Addgene | Plasmid #1654 |
| pDD162 | Addgene | Plasmid #47549 |
| pJA58 | Addgene | Plasmid #59933 |
| **Antibodies** | | |
| Rabbit anti-GSP-1 | Peel et al, 2017 | |
| Rat anti-OLLAS | Thermo Fisher | MA5-16125 |
| Rabbit anti-phospho Histone H3(Ser10) | Upstate | 06-570 |
| Mouse anti-Tubulin DM1A | Sigma | T9026 |
| HRP-conjugated anti-mouse | Bio-Rad | 170-6516 |
| HRP-conjugated anti-rabbit | Bio-Rad | 170-6515 |
| HRP-conjugated anti-rat | Bio-Rad | 2504-2504 |
| Alexa-Fluor-488-mouse | Invitrogen | A-11029 |
| Alexa-Fluor-568-rabbit | Invitrogen | A-11036 |
| **RNAi clones used in this study** | | |
| *control* | C06A6.2 | Ahringer library (Ahringer, 2006; Kamath et al, 2003) |
| *gsp-1* | F29F11.6 | Ahringer library |
| *gsp-2* | F56C9.1 | Calvi et al, 2022 |
| *sds-22* | T09A5.9 | Ahringer library |
| *szy-2* | Y32H12A.4 | Ahringer library |
| *ape-1* | F46F3.4 | This study |
| *rpn-6.1* | F57B9.10 | Ahringer library |
| *rpn-7* | F49C12.8 | Ahringer library |
| *rpn-12* | ZK20.5 | Ahringer library |
| **Oligonucleotides and other sequence-based reagents** | | |
| sgRNA1 for SDS-22(E153A) mutation forward | TCTTGGCAGCTCAAACTGCTCGAAC | |
| sgRNA1 for SDS-22(E153A) mutation reverse | AAACGTTCGAGCAGTTTGAGCTGCC | |
| sgRNA2 for SDS-22(E153A) mutation forward | TCTTGTTTCAATTTTATTGCTGACA | |
| sgRNA2 for SDS-22(E153A) mutation reverse | AAACTGTCAGCAATAAAATTGAAAC | |
| Repair template for SDS-22(E153A) mutation | AAATTGACAAAGCTCGAGACTCTT TAttTaGTCAGCAATAAAATTGAA AAAATCGAAAATTTAGAg GCccTGACGCAaCTaAAgtTGCTaG cACTaGGAGATAATCGgtaaatatggaa | |
| Primers for SDS-22(E153A) verification forward | GTCCAACGACAAATCCGCTGA | |
| Primers for SDS-22(E153A) verification reverse | GGTTCGACGCCATGAATATCTTGT | |
| Primers for pDONR201-SDS-22 cDNA (f.l.) forward | GGGGACAAGTTTGTACAAAAAAGCAG GCTTGATGTCCAACGACAAATCCGC | |
| Primers for pDONR201-SDS-22 cDNA (f.l.) reverse | GGGGACCACTTTGTACAAGAAAGCTGG GTGTTATTCAATTGGCTTTCTGCAC | |
| Primers for pDONR201-SDS-22 cDNA (f.l.,E153A) forward | CAAACTGCTCGcACTGGGAG | |

| Reagent/resource | Reference or source | Identifier or catalog number |
|---|---|---|
| Primers for pDONR201-SDS-22 cDNA (f.l.,E153A) reverse | AGCTGCGTCAAAGCTTCTAAAT | |
| qPCR *gsp-1* forward | TTCTGGAACTCGAAGCACCT | |
| qPCR *gsp-1* reverse | CTCTGCTTTCCACGATCGAC | |
| qPCR *gsp-2* forward | AGCTCCGTTGAAAATTTGCG | |
| qPCR *gsp-2* reverse | TCCCTCGGTCCACATAATCG | |
| qPCR *sds-22* forward | ATCTAACTCACACTCGCGCT | |
| qPCR *sds-22* reverse | AACGAAGAAATTGTGGGGCT | |
| qPCR *gpd-1* forward | ACCATGAGAAGTACGACGCT | |
| qPCR *gpd-1* reverse | GTGCACTGTCGTCATGAGTC | |
| qPCR *tba-1* forward | CCAACCTGAACCGCATCATC | |
| qPCR *tba-1* reverse | GCTCGAAGCAACTGTTGGTG | |
| RNAi clone primer T7 | CGTAATACGACTCACTATAG | |
| RNAi *ape-1* forward | TGAATTGCCAACCGAACAAATGG | |
| RNAi *ape-1* reverse | CCTGCATCAAAACTGAGCTCATCTTC | |
| **Chemicals, enzymes and other reagents** | | |
| 3-aminotriazole | Sigma | 09540 |
| IPTG | Applichem | A1008 |
| cOmplete™, Protease Inhibitor Cocktail | Roche | 4693159001 |
| PhosSTOP | Roche | 4906845001 |
| ChromoTek mNeonGreen-Trap Agarose | Chromotek | No. ntma |
| TRIzol™ Reagent | Invitrogen | 15596026 |
| Isopropanol | Sigma | I9516 |
| Gateway® LR Clonase™ II Enzyme Mix | Invitrogen | 11791-020 |
| Gateway® BP Clonase™ II Enzyme Mix | Invitrogen | 11789-020 |
| Proteinase K | Invitrogen | 25530-015 |
| T7 RiboMAX™ Express Large Scale RNA Production | Promega | P1320 |
| Taq DNA polymerase | Roche | 03734927001 |
| Pfu DNA polymerase | Promega | M7745 |
| HindIII | New England Biolabs | R0104S |
| BsrBI | New England Biolabs | R0102S |
| PrimeScript Reverse Transcription Kit | Takara | RR037B |
| FastStart Universal SYBR Green Master (Rox) | Merck | 4913850001 |
| poly-D-lysine | Sigma | P1024 |
| **Software** | | |
| Graphpad Prism 10 | Graphpad Software | https://www.graphpad.com/scientificsoftware/prism/ |
| Fiji ImageJ | National Institutes of Health | https://imagej.net/software/fiji/ |
| Design & Analysis 2 (DA2) software | Thermo Fisher | https://www.thermofisher.com/ch/en/home/technical-resources/software-downloads/quantstudio-6-7-pro-real-time-pcr-system.html |
| **Other** | | |
| Nikon ECLIPSE Ni-U microscope | Nikon | N/A |
| Applied Biosystems QuantStudio 12 K Flex Real-Time PCR system | Thermo Fisher | N/A |

## RNA interference

Clones from the Ahringer feeding library (Ahringer, 2006; Kamath et al, 2003) were used as listed in "Reagents and tools table". The clone

C06A6.2 served as the control for the injection experiments, as previously done (Bondaz et al, 2019). A fragment of APE-1 was amplified from cDNA and cloned into the final pDEST-L4440 vector using Gateway technology (primers listed in "Reagents and tools table").

Double-stranded RNA (dsRNA) for injections was produced with the Promega Ribomax RNA production system.

dsRNA was injected in L4/young adult hermaphrodites, which were then incubated at 20 °C. For the co-depletion of SDS-22 and GSP-1/-2, and co-depletion of GSP-1 and GSP-2, a mixture of 1:1 dsRNAs was injected (Figs. 4 and EV4C; Appendix Figs. S3,4). Embryos from injected worms were analyzed 16-20 h after injection.

RNA interference by feeding was performed on plates containing 1 mM IPTG. The L4440 vector was used as control. For the pkc-3(ne4246) strains and controls, worms were incubated on feeding plates at semi-restrictive temperatures of 24 °C (Figs. 1, 3D and EV1A) or 22 °C (Fig. 3E–G). L4 larvae were added to RNAi feeding plates and incubated for 24 h. For co-depletion of SDS-22 and RPN-6.1/-7 (Fig. 7; Appendix Figs. S5 and 6), a mixture of 1:1 fresh bacteria culture was plated. Worms were incubated on feeding plates at 20 °C for 24 h (Figs. 5A, 6A–F, 7, EV1B and EV2; Appendix Figs. S5 and 6). For RPN-12 depletion, L1 worms were added to the RNAi feeding plates and incubated for 72 h (Fig. EV5). For partial depletion of GSP-2 and GSP-1, young adult worms were injected with gsp-1 dsRNA and subsequently placed on RNAi feeding plates seeded with a 1:10 mixture of gsp-2 RNAi bacteria and L4440 control bacteria (Fig. EV4). As reported by Conte et al, RNAi feeding is less effective following prior dsRNA injection (Conte et al, 2015); however, since we aimed at a partial depletion, this is not of concern for this experiment. For depletion of SDS-22 from L1 larval stage, worms were added to RNAi feeding plates and incubated for 72 h (Appendix Fig. S7).

## Immunoprecipitation and mass spectrometry

For immunoprecipitation experiments from embryos of mNeon-Green tagged GSP-2, worms were grown in 3× PEP plates and embryos were harvested from gravid worms by bleaching (500 mM NaOH and 5% bleach). The N2 strain was used as a control. The mNeonGreen::GSP-2 and control immunoprecipitations were performed in triplicate. Packed embryos were frozen in liquid nitrogen and stored at −80 °C. Embryos were then thawed, resuspended in immunoprecipitation buffer (100 mM KCl, 50 mM Tris (pH: 7.5), 1 mM MgCl$_2$, 1 mM DTT, 5% glycerol, 0.2% NP-40, 1 mM EDTA, and protease and phosphatase inhibitor cocktail (Roche)). The embryos were ground using a mill (Retsch MM301) and silica beads (Lysing matrix C, MP Biomedicals). The protein homogenate was centrifuged at 14,000 rpm for 30 min at 4 °C. The protein concentration in the supernatant was determined in a Bradford assay (Bio-Rad Laboratories) using a UV/Vis Spectrophotometer (Labgene Scientific). About 3 mg of embryonic extract was incubated at 4 °C for 1.5 h with 10 μl of mNeonGreen-Trap Agarose beads (Chromotek), previously washed three times with IP buffer. After the incubation with the embryonic extract, beads were washed three times with IP buffer, and three times with the buffer containing 100 mM KCl, 50 mM Tris (pH: 7.5), 1 mM MgCl$_2$, 1 mM DTT, 1 mM EDTA to remove nonspecific binding. Elution of mNeonGreen-tagged GSP-2 was performed by incubation of the beads for 10 min at room temperature with 25 μl of 0.15% TFA. The pH of the elution was corrected to pH: 7–8 with 0.5 M Tris (pH 7.9). 3 μl of the elutions were used to analyse the immunoprecipitation by western blot. 22 μl of the elutions were snap frozen and sent to Biognosys (https://biognosys.com) for mass spectrometric analysis. The elutions were subjected to denaturation, reduction, alkylation, digestion and C18 clean up. The peptide digests were acquired in DIA mode using an Orbitrap Exploris 480 (Thermo Fisher) and analyzed with directDIA (Spectronaut 15).

## Imaging of live embryos

Adult hermaphrodites were dissected on a coverslip in a drop of Egg Buffer (118 mM NaCl, 48 mM KCl, 2 mM CaCl$_2$, 2 mM MgCl$_2$, and 25 mM HEPES, pH 7.5). Embryos were mounted on a 3% agarose pad for imaging. Time-lapse recordings, with frames captured every 10 s, were conducted using a Nikon ECLIPSE Ni-U microscope equipped with a Nikon DS-U3 Digital Camera and a 60×/1.25 numerical aperture (NA) objective. Imaging was performed between 20 and 22 °C.

## Immunostaining and imaging of fixed embryos

Fixation and staining of embryos were carried out as described previously (Calvi et al, 2022). The primary antibody phospho-Histone 3 (Ser10) (Millipore, rabbit) was diluted 1:500 in PBS-Tween with 1% BSA and slides incubated overnight at 4 °C. Slides were then washed and incubated with the secondary antibody (4 μg/ml Alexa-Fluor 568–coupled anti-rabbit antibody) and 1 μg/ml DAPI to visualize DNA. Images were acquired using a Nikon ECLIPSE Ni-U microscope, equipped with a Nikon DS-U3 Digital Camera, and using a 60×/1.25 numerical aperture (NA) objective.

## Embryonic lethality

To assess the embryonic lethality, young adult worms were singled on NGM plates seeded with OP50 and incubated at 20 °C for 24 h. Adults were then removed and the plates were incubated for an additional 24 h at 20 °C (Figs. 3C and EV3B–D, and other sds-22(E153A) mutants in N2 and in sds-22::gfp genetic backgrounds). The mNG::gsp-2; sds-22(E153A) strain was maintained as a sds-22(E153A) heterozygote, since in the homozygote form it is 99.1% embryonic lethal. To quantify the lethality of the homozygote, L4/young adult worms (homozygote wild-type, heterozygote, or homozygote sds-22(E153A) mutant) were singled on OP50 seeded-NGM plates, allowed to lay eggs for 24 h, and subsequently lysed for genotyping. Homozygote sds-22 wild-type and heterozygote worms expressing wild-type sds-22 from one allele and mutant sds-22(E153A) from the other allele were used as a control. The embryonic lethality can be used to differentiate homozygotes from heterozygotes and wild types for quantification (e.g. Fig. 6G,H). However, since depletion of RNP-6.1 or RPN-7 resulted in few or no progeny (Fernando et al, 2022), this precluded the possibility to analyze the germlines of homozygote mNG::gsp-2; sds-22(E153A) worms with RNP-6.1 or RPN-7 depletion, as we cannot distinguish the homozygote from the heterozygote.

SDS-22(E153A) mutation in wild-type N2 exhibited 8.3% of lethality, while sds-22(E153A)::gfp showed a lethality of 51.9%.

For the RNAi feeding and lethality assay of pkc-3(ne4246) mutant, L4 worms were initially transferred to 1 mM IPTG plates seeded with feeding bacteria and incubated overnight at 15 °C. The following day, young adult worms were singled onto feeding plate and incubated for 24 h at 24 °C (Figs. 1A and 3D) and 22 °C (Fig. 3E). After this period, the adult worms were removed and the plates were further incubated at 24 °C and 22 °C, respectively, for 24 h. The ratio of unhatched embryos to the total F1 progeny

(unhatched embryos plus larvae) was used to calculate the percentage of embryonic lethality.

## Yeast two-hybrid assay

The interaction between SDS-22 and GSP-1/-2 was assessed using a GAL4-based yeast two-hybrid (Y2H) system (Gateway, Invitrogen) using the MAV203 yeast strain. Full-length SDS-22 (1–326) fragments, both wild-type and mutant (E153A), were fused to the GAL4 DNA binding domain (DBD, Bait plasmid). Full-length cDNAs of GSP-1 and GSP-2 were fused to the GAL4 activation domain (AD, Prey plasmid). The SDS-22 wild-type and mutant fragments, GSP-1 and GSP-2 were first cloned into the pDONR201, then transferred to the pDEST32 (GAL4DBD) and pDEST22 vector (GAL4AD) separately using Gateway technology. Mutations were introduced by Pfu site-directed mutagenesis. A list of plasmids and primers used for the Y2H assay is provided in "Reagents and tools table". Transformants were selected on synthetic defined (SD) medium lacking leucine and tryptophan. Interactions were tested by spotting single colonies containing the desired plasmids on medium lacking leucine, tryptophan and histidine, with 7.5 mM of 3AT (3-amino-1,2,3- triazole, Sigma). Picture of the plates were captured using the Fusion FX6 EDGE Imaging System (Vilber) equipped with an Evo-6 Scientific Grade CCD camera.

## Western blot

Embryos obtained by hypochlorite treatment from two 5 cm plates of adult worms were resuspended in SDS sample buffer and denatured at 95 °C for 5 min. Approximately 3000 embryos per sample were loaded onto a 10% SDS-PAGE gel, followed by Western blotting with ECL detection (Vilber, Fusion FX). Primary antibodies [(anti-TUBULIN (1/2500, mouse, Sigma), anti-GSP-1 (1/1000, rabbit, courtesy of Kevin F. O'Connell) and anti-OLLAS (1/1000, rat, Thermo Fisher)] were incubated overnight at 4 °C. HRP-conjugated secondary antibodies were incubated for 45 min at room temperature.

Band signal measurement for anti-TUBULIN, anti-GSP-1 and anti-OLLAS was performed in Fiji ImageJ by drawing equal-sized ROIs over each band. Mean intensity values were used for quantification, and background was subtracted using the mean intensity of an equivalent ROI placed below the band of interest. The ratio of ctrl(RNAi) treatment was normalized to 1.

## Quantitative-PCR analysis

Total RNA was extracted from *C. elegans* embryos using a TRIzol-based protocol adapted from the Greer Lab (https://greerlab.wustl.edu/items/c-elegans-rna-extraction/). For each volume of packed embryos, 10 volumes of cold (4 °C) TRIzol was added. For cDNA synthesis, 150 ng of total RNA per sample was reverse transcribed using the PrimeScript Reverse Transcription Kit (TAKARA). Quantitative PCR (qPCR) was performed in 384-well plates using SYBR Green as the fluorescent dye on an Applied Biosystems QuantStudio 12 K Flex system, with support from the iGE3 genomics platform (https://ige3.genomics.unige.ch/). Relative gene expression levels were calculated using the ΔΔCt method, with normalization to the geometric mean of two housekeeping genes, *tba-1* and *gpd-1*. Expression values were plotted as fold changes relative to housekeeping genes, and statistical significance was assessed using paired *t* tests on ΔCt values. Primers for qPCR analysis were listed in "Reagents and tools table".

## Image analysis and measurement

### Cortical PAR-2 intensity measurement (Figs. 1B,C and 3F,G)
The line file of GFP::PAR-2 intensity along the entire cortex was measured in one-cell stage embryos at pronuclear meeting. The stage of pronuclear meeting is defined as the first timepoint in which the two pronuclei contact each other. In *pkc-3(ne4246)* embryos, the two pronuclei exhibited a tendency to meet more centrally compared to controls (Fig. 1B; Movie EV1), as shown in (Kirby et al, 1990; Rodriguez et al, 2017). Using ImageJ software, a free-line 5-pixel-wide was positioned at the anterior pole of the cortex (position 0 μm) and traced manually along the whole cortex to the posterior pole (position 75 μm) and back to the anterior pole. The background was subtracted from each value. The intensity of the line profile (from 0 to 75 μm) of the segment traced was shown in an arbitrary unit (a.u.). Images were in 16-bit format.

### Cortical PAR-2 size measurement (Figs. 1D, 3B, 4, EV4C,D and EV5E,F)
The size of cortical GFP::PAR-2 domain was determined by measuring the length of the PAR-2 domain normalized for the total perimeter of the embryo at pronuclear meeting. The perimeter of the embryo and the length of the PAR-2 domain were measured by the Qupath software with machine learning, as detailed previously (Vaudano et al, 2024). The length of the PAR-2 domain is represented as a percentage of the total perimeter of the embryo.

### AB cell size measurement (Fig. 1E,F)
The total length of the embryo was measured manually from the anterior to the posterior pole using ImageJ software. AB cell size was measured from the anterior pole to the cleavage midline when the embryo just divided into two cells. The AB cell ratio was calculated as the proportion of AB cell size relative to the total embryo length (AB cell plus P1 cell length).

### Protein intensity of mNG::GSP-2, GFP::GSP-1, SDS-22::GFP in embryos and germlines (Figs. 6, 7, EV4A,B and EV5A–D; Appendix Figs. S2, S4-7)
For embryos, the mean intensity of cytoplasmic mNG::GSP-2, GFP::GSP-1, SDS-22::GFP was determined by tracing a fixed-length line in the cytoplasm from the anterior to the posterior of the embryo (above the nucleus) at pronuclear meeting stage using the ImageJ software. The background was subtracted from each value.

For the embryonic intensity of mNG::GSP-2 in the *sds-22(E153A)* mutant genetic background, which is embryonic lethal, L4/young adult worms (homozygote wild-type, heterozygote or homozygote *mNG::GSP-2; sds-22(E153A)* mutant) were singled, let them lay a few eggs, and subsequently dissected for imaging. Homozygous *mNG::GSP-2; sds-22(E153A)* mutant worms were recognized by looking at the progeny in the original plates, as homozygote *mNG::GSP-2; sds-22(E153A)* worms only lay dead eggs. Homozygote *sds-22* wild-type and heterozygote worms expressing wild-type *sds-22* from one allele and mutant *sds-22(E153A)* from the other allele were used as a control.

For germlines, mean cytoplasmic and nuclear intensities of mNG::GSP-2, GFP::GSP-1, SDS-22::GFP were measured by

drawing a fixed circle in the cytoplasm and/-or nucleus of -1 and -2 oocytes using ImageJ. Background intensity was subtracted. The mean intensities of the cytoplasmic and nuclear regions in -1 and -2 oocytes were calculated and reported.

The germline intensity of mNG::GSP-2 in the *sds-22(E153A)* mutant cannot be measured because there were no viable embryos after depletion of RNP-6.1 or RNP-7, therefore the genotype of homozygous *mNG::gsp-2; sds-22(E153A)* (also embryonic lethal) cannot be differentiated from the homozygote wild-type and heterozygote.

### Phospho-Histone 3 (Ser10) intensity measurement (Fig. 5)

One-cell stage embryos at anaphase were analyzed. A segment 5-pixel-wide and of constant length, centered at the chromosomes was traced using ImageJ software. For each embryo, the average of three segments (upper, center, and lower chromosomes) was used for quantification. The intensity of the line profile of each embryo was normalized to the average of the value in the cytoplasm at position 0 of the segment traced. The mean intensities of the phospho-Histone 3 (Ser10) on the chromosomes were calculated and reported.

## Statistical analysis

The investigators were not blinded to allocation during experiments and outcome assessment. No statistical method was used to predetermine the sample size. Sample sizes analysis tools and statistics were chosen according to commonly used and accepted standards in the field.

Statistical analysis was conducted using GraphPad Prism 10. Details on the statistical tests, sample sizes, number of replicates, and interpretation of error bars are specified in each figure legend, within the "Results" section, or in the "Methods" section, and summarized in Dataset EV3. Data was assumed to have normal distribution.

Significance was defined as: ns, $P > 0.05$, $*P < 0.05$, $**P < 0.01$, $***P < 0.001$, $****P < 0.0001$.

## Data availability

All the strains and reagents generated in this study are available from the corresponding author upon request. All raw data associated with the experiments can be found at https://doi.org/10.26037/yareta:rij6bpbhwvfalhmajupmiucr5m.

The source data of this paper are collected in the following database record: biostudies:S-SCDT-10_1038-S44319-025-00624-0.

## Peer review information

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

## Acknowledgements

A special thanks to Nate Goehring, Erik Griffin, Dhanya Cheerambathur, and Kevin F. O'Connell for strains and antibodies. We would like to thank all the present and past members of the Gotta, Meraldi and Steiner laboratories, Patrick Meraldi and Florian Steiner for discussions and suggestions. Thanks to Sofia Barbieri, Victoria von Glasenapp and Florian Steiner for comments on the manuscript. We would like to thank Biognosys (https://biognosys.com/) for performing the proteomic analysis and the iGE3 genomics platform (https://ige3.genomics.unige.ch/) for performing qPCR analysis. Some strains were provided by the CGC, which is funded by NIH Office of Research Infrastructure Programs (P40 OD010440). This research was funded by the Swiss National Science Foundation (grant number 31003A_175850) and the University of Geneva.

## Author contributions

**Yi Li**: Conceptualization; Formal analysis; Investigation; Visualization; Writing—original draft; Writing—review and editing. **Ida Calvi**: Conceptualization; Investigation; Writing—review and editing. **Monica Gotta**: Conceptualization; Supervision; Funding acquisition; Writing—original draft; Project administration; Writing—review and editing.

Source data underlying figure panels in this paper may have individual authorship assigned. Where available, figure panel/source data authorship is listed in the following database record: biostudies:S-SCDT-10_1038-S44319-025-00624-0.

## Disclosure and competing interests statement

The authors declare no competing interests.

# Expanded View Figures

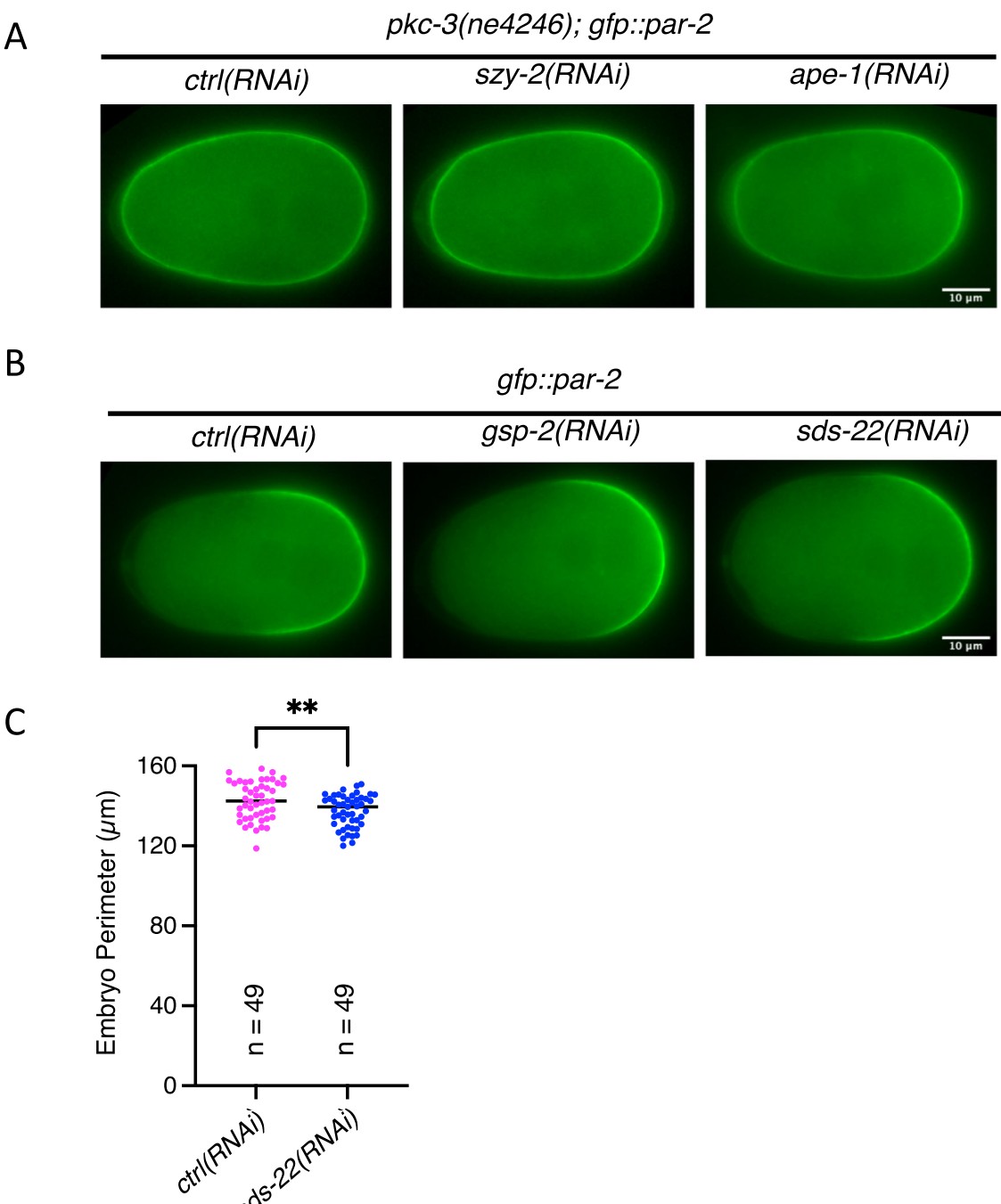

**Figure EV1.  PAR-2 localization after depletion in different genetic backgrounds of candidate regulators of PP1.**

(A) Still images of embryos at the pronuclear meeting stage taken from time-lapse videos: *pkc-3(ne4246); gfp::par-2; ctrl(RNAi)*, n = 18, *szy-2(RNAi)*, n = 17, N = 4, and *ape-1(RNAi)*, n = 16, N = 2. (B) Still images of embryos at the pronuclear meeting stage taken from time-lapse videos of *gfp::par-2*, comparing *ctrl(RNAi)*, n = 33, *gsp-2(RNAi)*, n = 24 and *sds-22(RNAi)*, n = 35. N = 3. The quantification of the size of the PAR-2 domain is displayed in Fig. 1D. (C) Quantification of embryo perimeter with indicated RNAi conditions. For both *ctrl(RNAi)* and *sds-22(RNAi)*, n = 49, N = 6. For the images in (B) and quantifications of the perimeter in (C) we used a subset of the embryos of the experiment of Fig. 4. The P value was determined using two-tailed unpaired Student's *t* test. Mean is shown and each dot represents a single embryo. ns $P > 0.05$, *$P < 0.05$, **$P < 0.01$, ***$P < 0.001$, ****$P < 0.0001$. Exact P values are provided in Dataset EV3. RNA interference was performed by feeding. For all embryos, the scale bar is 10 µm, anterior is to the left and posterior to the right. *n* number of embryos analyzed, *N* number of independent experiments.

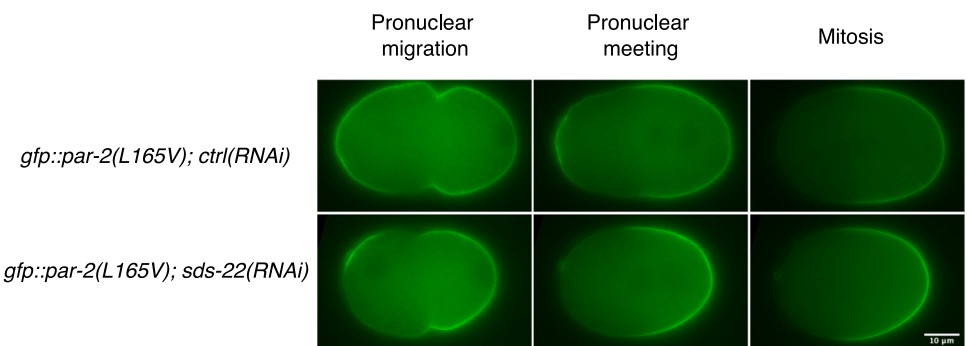

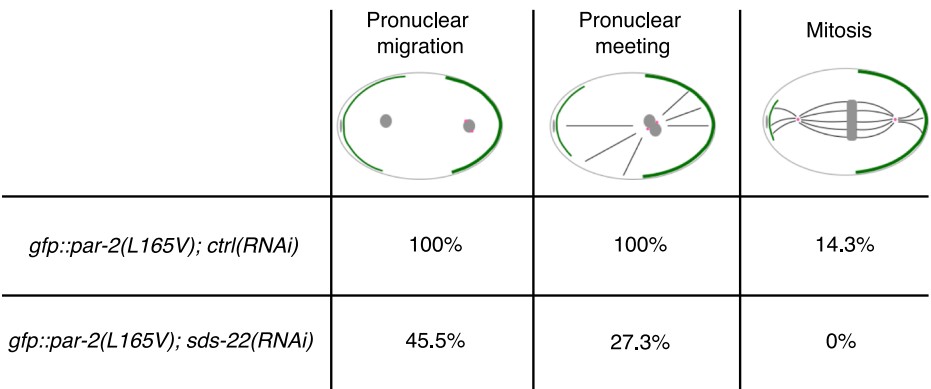

| | Pronuclear migration | Pronuclear meeting | Mitosis |
|---|---|---|---|
| *gfp::par-2(L165V); ctrl(RNAi)* | 100% | 100% | 14.3% |
| *gfp::par-2(L165V); sds-22(RNAi)* | 45.5% | 27.3% | 0% |

**Figure EV2.  Depletion of SDS-22 rescues aberrant PAR-2 localization in *gfp::par-2(L165V)* mutant.**

Top: still images from time lapse videos of *gfp::par-2(L165V)* of one-cell embryos at different cell division stages, *ctrl(RNAi)*, *n* = 9 and *sds-22(RNAi)*, *n* = 11, *N* = 3. For all embryos, the scale bar is 10 μm, anterior is to the left and posterior to the right. Bottom: the table shows the percentage of the phenotype represented schematically at the top. RNA interference was performed by feeding. *n* number of embryos analyzed, *N* number of independent experiments.

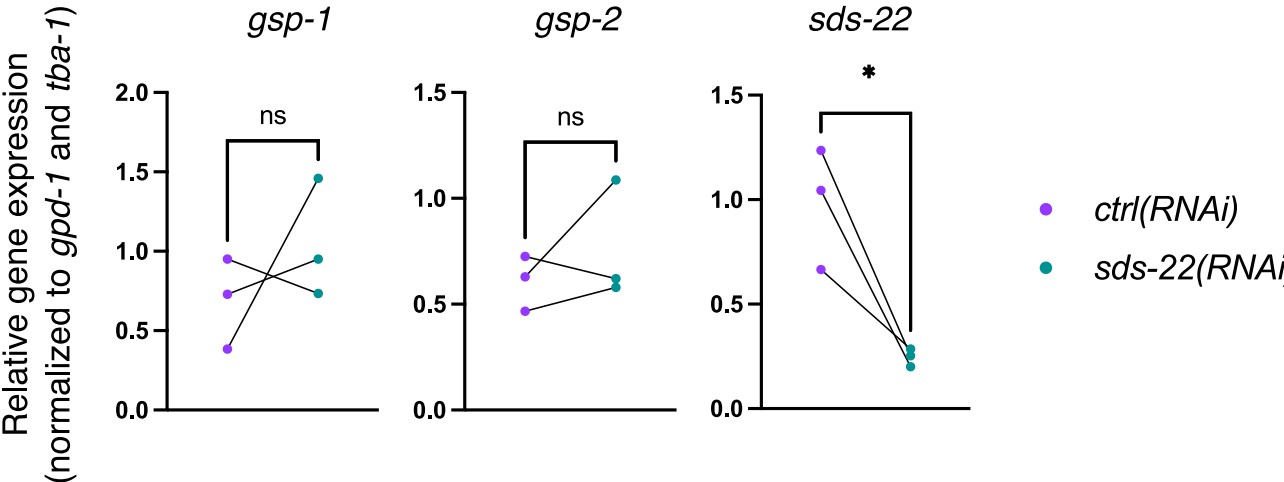

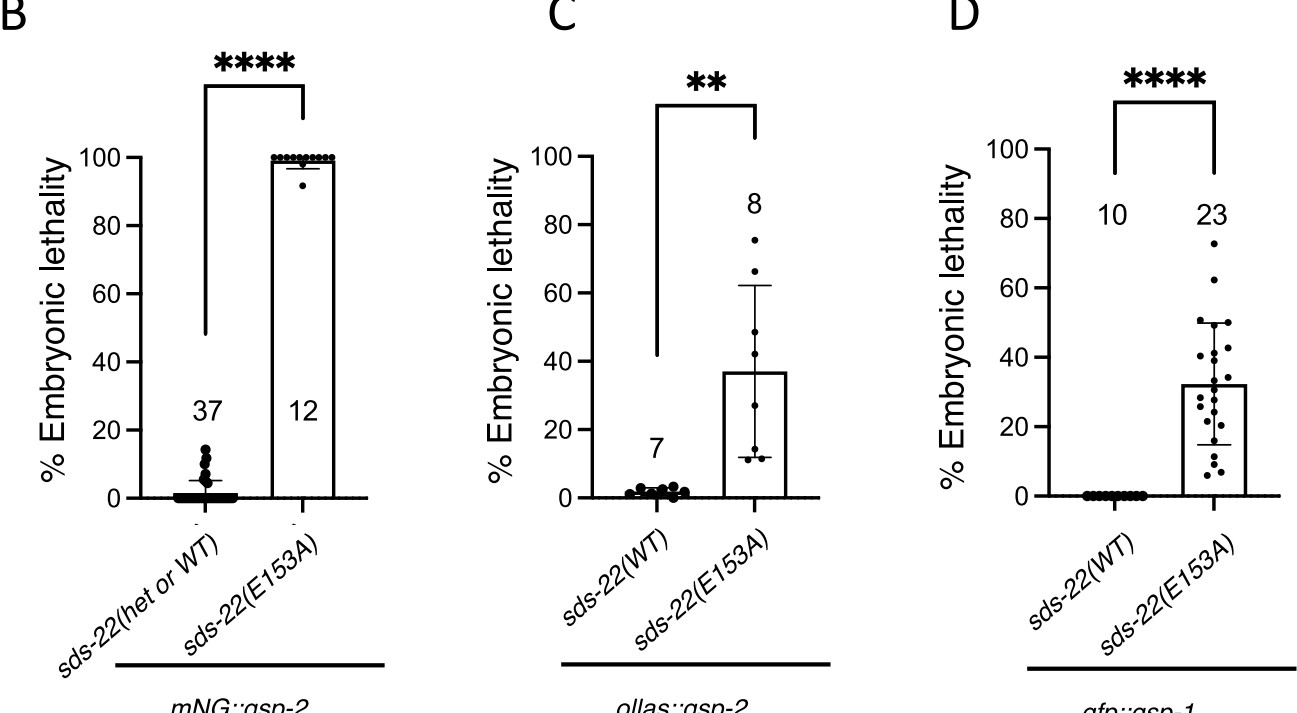

**Figure EV3. The E153A substitution in SDS-22 leads to embryonic lethality in the *mNG::gsp-2*, *ollas::gsp-2* and *gfp::gsp-1* genetic backgrounds.**

(A) Quantitative PCR of *gsp-1*, *gsp-2* and *sds-22* in *N2* following SDS-22 depletion. The geometrical mean between two control genes (*gpd-1* and *tba-1*) was used to calculate the relative gene expression. The statistical significance has been calculated by paired *t* tests comparing ΔCt values of different samples. Each dot is the value of one experiment. $N = 3$. RNA interference was performed by feeding. (B–D) Embryonic lethality of embryos with the *sds-22(E153A)* mutation in the genetic background of *mNG::gsp-2* (B), *ollas::gsp-2* (C) and *gfp::gsp-1* (D). The reported values correspond to the percentage of unhatched embryos over the total progeny (larvae and unhatched embryo). *mNG::gsp-2; sds-22(E153A/+ or +/+)*, $n = 1703$, *mNG::gsp-2; sds-22(E153A)*, $n = 568$. $N = 3$. *ollas::gsp-2*, $n = 476$, *ollas::gsp-2; sds-22(E153A)*, $n = 470$. $N = 2$. *gfp::gsp-1*, $n = 713$, *gfp::gsp-1; sds-22(E153A)*, $n = 1783$. $N = 2$. In this plot, each dot represents the quantified lethality of one single plate. Mean is shown and error bars indicate SD. The *P* values were determined using two-tailed unpaired Student's *t* test. In all plots, ns $P > 0.05$, *$P < 0.05$, **$P < 0.01$, ****$P < 0.0001$. Exact *P* values are provided in Dataset EV3. *n* number of embryos analyzed, *N* number of independent experiments.

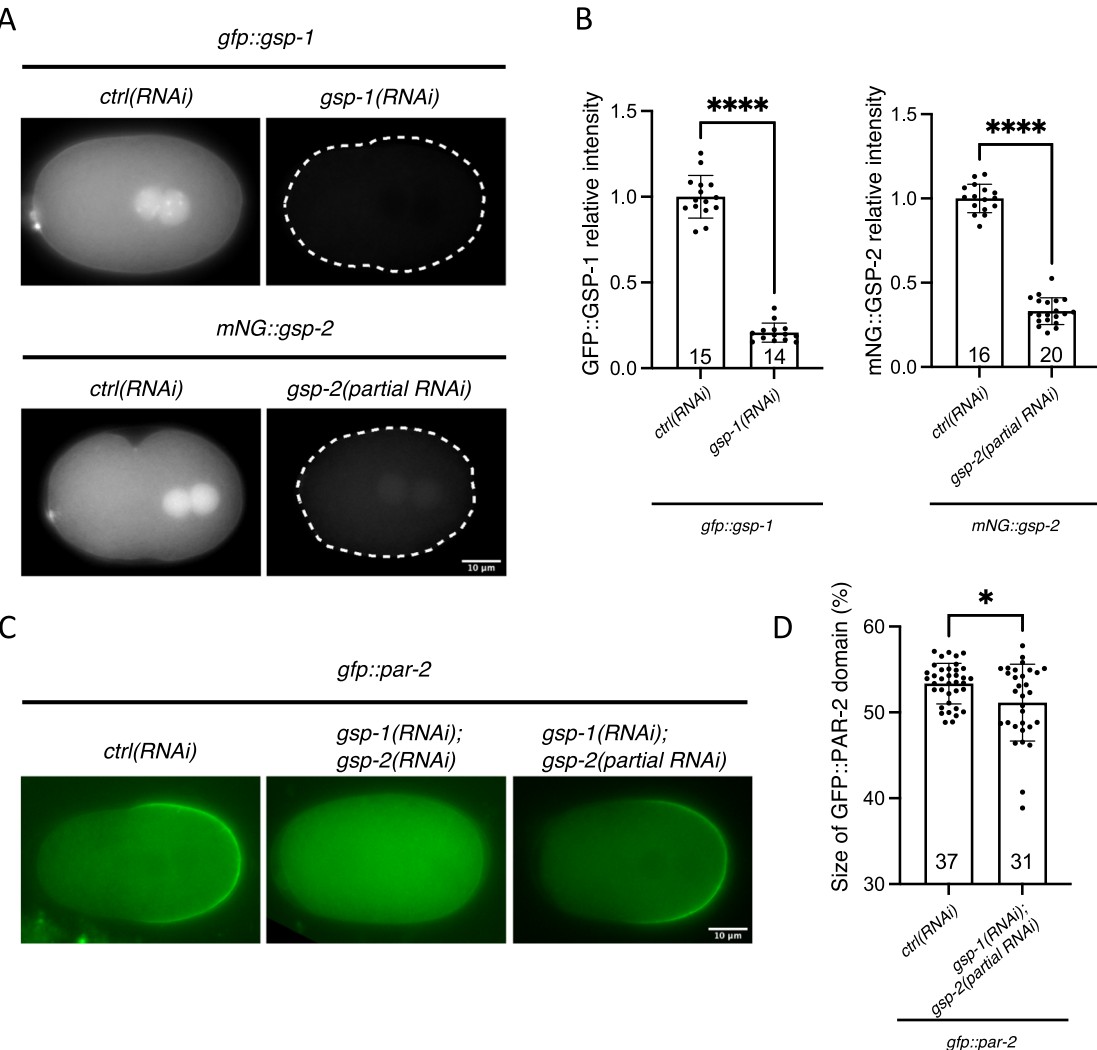

**Figure EV4.  PAR-2 polarizes with minimum amounts of GSP-1 and GSP-2.**

(A) Upper, representative images of *gfp::gsp-1* embryos in *ctrl(RNAi)* and *gsp-1(RNAi)*. RNA interference was performed by injection. Lower, representative images of *mNG::gsp-2* embryos in *ctrl(RNAi)* and *gsp-2(RNAi)*. RNA interference was performed by feeding diluted bacteria to obtain a partial depletion. (B) Quantification of relative GFP::GSP-1 and mNG::GSP-2 levels. N = 2. (C) Representative images of *gfp::par-2* embryos with the indicated depletion. Co-depletion of GSP-1 and GSP-2 was performed by injection (middle panel) and by injection of GSP-1 dsRNA and feeding diluted GSP-2 RNAi bacteria (right panel). (D) Quantification of the GFP::PAR-2 domain size at pronuclear meeting in live zygotes. N = 3. In all plots, mean is shown and error bars indicate SD. Sample size (*n*) is indicated inside the bars in the graph, each dot represents a single embryo. *P < 0.05, ****P < 0.0001. The P values were determined using two-tailed unpaired Student's *t* test. Exact P values are provided in Dataset EV3. *n* number of embryos analyzed, N number of independent experiments.

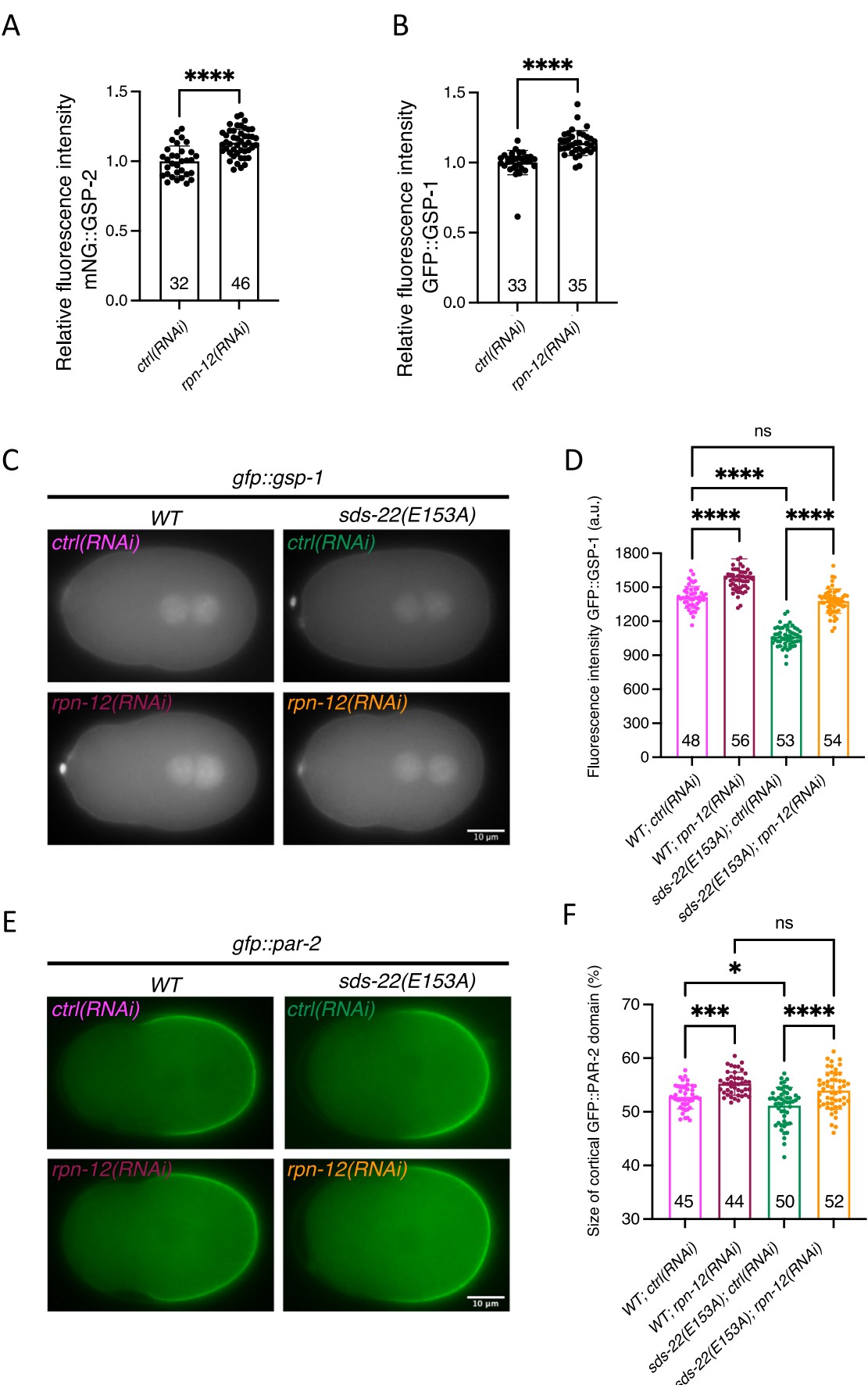

◀ **Figure EV5. Depletion of RPN-12 increases GSP-1/-2 levels and PAR-2 domain length.**

(A, B) Quantification of mNG::GSP-2 and GFP::GSP-1 in *ctrl(RNAi)* and *rpn-12(RNAi)*. $N = 2$. The *P* values were determined using two-tailed unpaired Student's *t* test. (C) Representative images of *gfp::gsp-1* and *gfp::gsp-1; sds-22(E153A)* embryos in *ctrl(RNAi)* and *rpn-12(RNAi)*. (D) Quantification of GFP::GSP-1 intensity levels. $N = 3$. (E) Representative images of *gfp::par-2* and *gfp::par-2; sds-22(E153A)* embryos in *ctrl(RNAi)* and *rpn-12(RNAi)*. (F) Quantification of the GFP::PAR-2 domain size at pronuclear meeting in live zygotes. $N = 3$. For all embryos, the scale bar is 10 μm, anterior is to the left and posterior to the right. Mean is shown and error bars indicate SD. Each dot represents the measurement of one embryo. The *P* values were determined using one-way ANOVA "Tukey's multiple comparisons test". Sample size (*n*) is indicated inside the bars in the graph. In all plots, ns$P > 0.05$, *$P < 0.05$, **$P < 0.01$, ***$P < 0.001$, ****$P < 0.0001$. Exact *P* values are provided in Dataset EV3. *n* number of embryos analyzed, *N* number of independent experiments.

