## [Peer Review File · EMBO Reports]

SDS-22 stabilizes GSP-1/-2 PP1 subunits contributing to polarity establishment in *C. elegans* embryos

Monica Gotta, Yi Li, and Ida Calvi

Corresponding author(s): Monica Gotta (monica.gotta@unige.ch)

Review Timeline:

Transfer Date:	12th May 25
Editorial Decision:	27th May 25
Revision Received:	24th Jul 25
Editorial Decision:	26th Sep 25
Revision Received:	8th Oct 25
Accepted:	20th Oct 25

Editor: Deniz Senyilmaz Tiebe / Martina Rembold

**Transaction Report: This manuscript was transferred to
EMBO reports following peer review at Review Commons.**

**Review
COMMONS**

Review #1

1. Evidence, reproducibility and clarity:

Evidence, reproducibility and clarity (Required)

Summary

This work from Li et al. identifies a role for SDS-22 in maintaining normal levels of the PP1 subunits GSP-1/2 in *C. elegans*. Prior work in the lab identified a role for the catalytic PP1 subunits GSP-1/2 in opposing the phosphorylation of the polarity protein PAR-2 by the polarity kinase PKC-3(aPKC). To identify potential regulatory subunits in this process, they performed IP-MassSpec on GSP-2 and pulled out a number of potential regulatory subunits, including SDS-22. While polarity was the primary motivation for the study, their subsequent analysis did not point to a specific function in cell polarity, but rather pointed to a general effect on stabilising levels of GSP-1/2 against potential proteasome-mediated degradation. Consistent with this hypothesis, SDS-22 depletion or mutation of the SDS-22-GSP-1/2 interaction partially recapitulated the phenotypes of GSP-1/2 depletion including increased phosphorylation of histone H3. Consistent with reduced PP1 activity, there were modest effects on polarity as seen in GSP-2 depleted embryos, including slightly reducing PAR-2 domain size and partially restoring PAR-2 asymmetry in embryos carrying a temperature sensitive *pkc-3* mutation.

Major Comments:

1. Overall, the evidence supporting the core finding that SDS-22 is required for normal GSP-1/2 levels is strong and well documented. The experiments were performed well and controls, statistics, replicates were appropriate. Our only slight reservation was whether the effect of *sds-22*(RNAi) on stability may be overstated due to the use of GFP fusions to GSP-1/2 for this analysis. The authors note these alleles are hypomorphic, potentially raising the possibility that GFP tags destabilise the proteins and make them more prone to degradation. Ideally this would be repeated with an untagged allele via Western (e.g. Peel et al 2017 for relevant antibodies).

2. The role for SDS-22 in polarity is rather weak. Both the SDS-22 depletion phenotypes and the ability of SDS-22 depletion to suppress *pkc-3*(ts) polarity phenotypes are modest (and weaker in than GSP-2 depletion). For example, the images in Figure 1B appear striking, but from Movie S1 it is clear that this isn't a full rescue as PAR-2 is initially uniformly enriched on the cortex (rather than mostly cytoplasmic) and it is never fully cleared. In the movie, the clearance at the point of pronuclear meeting is very modest. Quantitation might be helpful

here (i.e. as in Figure 3G). As the authors state, it seems that SDS-22 does not have a specific role in polarity beyond the general effect on GSP-1/2 levels. This does not undermine the core message of the paper, but we would recommend downplaying the conclusions with respect to contributing to polarity establishment. For example "...suggesting that SDS-22 regulates GSP-1/-2 activity to control the loading of PAR-2 to the posterior cortex in one-cell stage C. elegans embryos" implies a regulatory role for SDS-22 in polarity, but we would interpret it as simply helping reduce aberrant degradation of GSP-1/2 and this impacts a variety of cellular processes including polarity.

3. Specificity of SDS-22 effects on polarity. SDS-22 (or GSP-1/2) depletion is likely to have effects on many pathways. We wondered whether some of the polarity phenotypes may not be specifically due to changes in the PAR-2 phosphorylation cycle as implied.

One candidate is the actomyosin cortex. It was noticeable that control and *sds-22* embryos were different: In Movies S1, S2, and S3 control embryos show either stronger or more persistent cortical ruffling or pseudocleavage furrows. This is also visible in Figure 3A. Is it possible that disruption of SDS-22 reduces cortical flows (time, intensity or duration) and could this explain the small reduction in anterior PAR-2 spreading and thus the slightly smaller domain size measured in Figures 1B and 3A.

A potentially related issue could be embryo size. *sds-22* embryos generally seem to be smaller than wild-type (e.g. Figure 1B(left), 4A(left column), and particularly EV3). Is this consistently true? Could cell size effects change the ability of embryos to clear anterior PAR-2 domains as described in EV3? Klinkert et al (2018, biorXiv) note that reducing the size of *air-1*(RNAi) embryos reduces the frequency of bipolar PAR-2 domains.

We would stress that these comments relate to interpreting the polarity phenotypes and do not undermine the core finding that SDS-22 stabilises GSP-1/2.

****Minor Points****

- The link between lethality and polarity of the zygote is not always obvious and whether they are connected (or not) could probably be made clearer. Indeed, the source of lethality is unclear, particularly given that loss of SDS-22 on its own strongly impacts lethality with minimal effects on polarity (at least in the zygote).
- Formally, the conclusion that reduced GSP-1/2 in SDS-22 depletion conditions is due to increased proteasomal degradation is not shown directly as there is no data on rates just steady-state levels. We agree that the genetic data is strongly suggestive of this model and it is consistent with work of other labs. Thus this is the most likely scenario, but could in

principle reflect reduced expression that is balanced by reduced degradation.

- It is interesting that sds-22(E153A) caused a stronger decrease in oocyte GSP-1 levels than sds-22(RNAi) (Fig 7). The authors may want to comment on this result.
- "At polarity establishment, the PP1 phosphatases GSP-1/-2 dephosphorylate PAR-2 allowing its cortical posterior accumulation." This statement, possibly inadvertently, implies temporal regulation, which has not been shown.
- It would be ideal if the authors could explicitly state how they define pronuclear meeting. For example in Figure 1B, the embryos look like they are a few minutes past pronuclear meeting (e.g. compared to Figure 3), but maybe the pronuclei tend to meet more centrally in these conditions? Given that PAR-2 clearance is changing in time in some of these cases (based on looking at the movies), staging needs to be very accurate to get the best comparisons.
- In the interests of data-availability, upon publication the authors would deposit the raw mass spec data underlying Figure EV1.

****Referees cross-commenting****

We also generally agree with the comments of the other reviewers.

Our only concern with the main conclusion regarding GSP stabilization of GSP-1/2 is the impact of the gfp tags. Given that antibodies exist and have been used in Peel et al 2017 for exactly this purpose in *C. elegans* embryos, this does not seem excessively burdensome in our view and would strengthen the paper.

The remainder of our concerns can likely be addressed by modifications to the text and/or adding a caveats/limitations section to their discussion. As we noted, these mostly relate to the magnitude and specificity of the impact of SDS-22 on polarity and PAR-2 phosphorylation, which in our view is rather peripheral to the core conclusion (i.e. that SDS-22 stabilizes GSP-1/2).

2. Significance:

Significance (Required)

Overall, this is a careful and well-executed study identifying a conserved role for SDS-22 in stabilising PP1 catalytic subunits in *C. elegans* embryos and shows that this can broadly impact PP1 activity in this system. A mechanistic role for SDS-22 in PP1 function was recently demonstrated in (Cao et al, 2024), where it was shown to stabilise nascent catalytic subunits, but also in subunit recycling (Kuetsch et al 2024). The data here suggest

this role in stabilisation PP1 subunits is broadly relevant.

These data are also consistent with prior work from the lab demonstrating the role of PP1 in *C. elegans* zygote polarity. It adds to previous reports that compromised PP1 activity can impact cell polarity and further highlights the importance of considering regulation of protein phosphatases in cell polarity pathways. That said, the impact on polarity is rather modest, likely reflecting a general requirement for SDS-22 in supporting optimal PP1 activity rather than any specific role in polarity.

Field of expertise: cell polarity, cell and developmental biology.

3. How much time do you estimate the authors will need to complete the suggested revisions:

Estimated time to Complete Revisions (Required)

(Decision Recommendation)

Between 1 and 3 months

Yes

Review #2

1. Evidence, reproducibility and clarity:

Evidence, reproducibility and clarity (Required)

****Summary:**** The authors present a logical and clear set of data that support the model that SDS-22 is an important regulator of cell polarity via its ability to stabilize Protein Phosphatase 1 (PP1).

The authors use a clever combination of genetic manipulations and quantitative imaging to

show that loss of SDS-22 phenocopies loss of PP1, in that PAR-2 polarization is restored in nascent *C. elegans* zygotes following inactivation of PKC-3. Rescue of PAR-2 polarization also occurs when the authors mutate a conserved residue in SDS-2 that is predicted to form electrostatic interactions with PP1, suggesting that SDS-22 acts via PP1. Inactivation of SDS-22 results in decreased levels of PP1, and the authors provide evidence that this is via proteasomal degradation by showing that PP1 levels are restored by knockdown of proteasomal subunits.

Overall the manuscript is well-written, the experiments rigorous, and the methods and data likely to be reproducible.

****Major comment:**** Consistent with the model that PP1 activity is reduced in the absence of SDS-22, the authors show that a surrogate PP1 target (phospho-histone H3) becomes hyper-phosphorylated. To strengthen the study, the authors could consider performing an OPTIONAL experiment (see below) of assaying the phosphorylation status of PAR-2 itself, as this is proposed to be the target of both PKC-3 and PP1, and represent the mechanism of PAR-2 polarization.

****Referees cross-commenting****

In principle I agree with many of the thoughtful comments by the other reviewers. They point out many potential areas for both enhancing the strength of the findings and including a more nuanced interpretation of the results. However, I also feel that the experiments proposed to deal with their concerns might not be so straightforward to pursue for unforeseen technical reasons and may actually take substantially longer than anticipated. The same is true for my proposed experiment to assess phosphorylation status of PAR-2, which is why I have indicated it as optional. I ask the other reviewers to consider if any of their proposed experiments might also be considered optional. I also thank them for their critical assessments of the paper! They were helpful for me and I'm sure will also be for the authors.

2. Significance:

Significance (Required)

This study brings clarity to the contentious role of SDS-22 by showing that it appears to promote PP1 activity by counteracting the phosphatase degradation process *in vivo*. This complements a previously hypothesized function of SDS-22, while contrasting with other proposed functions of SDS-22 as a regulator of PP1 localization or stimulator of PP1

degradation. Thus, the authors' discoveries in *C. elegans* represent a significant advance in our understanding of protein phosphatase regulation, a long-standing question in biology and a central process in all cellular systems. The study also points to potential mechanisms for modulating phosphatase activity in other contexts, across different organisms and disease states. Basic science researchers will be interested in the findings, with potential to attract additional interest from physiologists and even drug designers.

One limitation of the study is that the authors use PAR-2 polarization as a readout of PKC-3 and PP1 activity without showing directly that PAR-2 phosphorylation status is changing in response to their genetic manipulations, including SDS-22 inactivation. The PAR-2 membrane localization is thought to be inhibited by PKC-3-dependent phosphorylation and promoted by PP1-dependent dephosphorylation. Is there a possibility of examining whether PAR-2 phosphorylation is elevated in SDS-22 RNAi or mutant animals? Previously, Hao et. al., 2006; doi.org/10.1016/j.devcel.2005.12.015. showed in Figure 2 that PAR-2 runs as a doublet band on western blots with the phosphorylated form of PAR-2 appearing to correlate with the slightly higher molecular weight band. This was used to infer the ratio of phosphorylated to dephosphorylated PAR-2. I'm wondering if it might be possible for the authors to perform a similar analysis of their existing GFP::PAR-2? It appears from their previous paper on PP1 regulation of PAR-2 polarization (Calvi et. al., 2022; doi: 10.1083/jcb.202201048.) that they might also be detecting a similar doublet (Figure S5F and associated source file), so perhaps it is a doable experiment? Regardless, this is not an essential experiment as the study is already significant and rigorous!

I am a cell biologist that uses *C. elegans* to understand the function of conserved protein complexes that regulate the development and function of animal tissues.

3. How much time do you estimate the authors will need to complete the suggested revisions:

Estimated time to Complete Revisions (Required)

(Decision Recommendation)

Less than 1 month

4. Review Commons values the work of reviewers and encourages them to get credit for their work. Select 'Yes' below to register your reviewing activity at Web of Science Reviewer Recognition Service (formerly Publons); note that the content of your review will not be visible on Web of Science.

No

Review #3

1. Evidence, reproducibility and clarity:

Evidence, reproducibility and clarity (Required)

****Summary:**** your understanding of the study and its conclusions

This is a follow up on Gotta's lab paper, which shows that the PP1 catalytic subunits GSP-1 and GSP-2 are involved in the polarization of the *C. elegans* zygote (10.1083/jcb.202201048). Here, the authors report that SDS-22, an interactor of PP1, regulates PP1 function in the zygote. Depleting SDS-22, similar to depleting GSP-2, rescues the polarity defects caused by the inactivation of aPKC in the zygote. This suggests that SDS-22 plays a role in promoting GSP-2's function in polarity. The mechanism behind this may involve SDS-22 protecting GSP-1 and GSP-2 from degradation by the proteasome.

****Major comments:**** major issues affecting the conclusions

Overall, the authors' conclusions are supported by their data. The data and methods are presented clearly, with appropriate replicates and statistics. Here I propose two experiments to strengthen the link between some of their data and their claims. These experiments could take a month or two to complete.

Experiment 1

It would be helpful if the authors could show that blocking the proteasome in the zygote restores GSP-1/-2 levels in the absence of SDS-22 or even better in the SDS-22(E153A) mutant. This would provide more direct evidence to support their claim that SDS-22 regulates polarity by protecting PP1 from proteasomal degradation. While they are currently conducting this experiment in the germline, they cannot assess polarity there. However, in the zygote, they would be able to examine the PAR-2 domain (polarity). To do this, the authors could permeabilise the embryos and apply a proteasome inhibitor.

Experiment 2

The posterior localization of PAR-2 after co-RNAi of GSP-1 and SDS-22 contrasts with the absence of PAR-2 at the cortex when both GSP-1 and GSP-2 are depleted. This difference may be due to the partial reduction of GSP-2 levels when SDS-22 is depleted, compared to the more substantial reduction of GSP-2 upon GSP-2 RNAi. Have the authors considered combining full depletion of GSP-1 with partial depletion of GSP-2 to see if PAR-2 remains present and localized to the posterior? This experiment could help clarify the discrepancy between the phenotypes and further support the role of SDS-22 in regulating GSP-2 protein levels. Additionally, by titrating PP1, the authors may be able to determine the minimum amount of PP1 needed to establish the PAR-2 domain.

****Minor comments:**** important issues that can confidently be addressed

In the introduction (line 83), it's unclear what reconciles the contradictory data. I also have difficulty understanding this point in the discussion (line 435).

Additionally, in the results section (line 389), it's not clear why the gonads cannot be studied in the strain with dead embryos. Are the gonads also altered in a way that prevents their observation?

****Referees cross-commenting****

Overall, I agree with the other reviewers' comments. The suggested experiments would help strengthen the connection between SDS-22 and cell polarity, as well as its role in relation to the proteasomal-mediated degradation of GSP-1/-2 and its impact on cell polarity. These experiments seem feasible and could provide stronger support for the authors' claims about these regulatory mechanisms. Alternatively, the authors may consider moderating some of their conclusions if these experiments are not conducted.

2. Significance:

Significance (Required)

General assessment: strengths and limitations

This study enhances our understanding of how phosphatases regulate cell polarity, specifically in the *C. elegans* zygote, a key model system for studying cell polarity. The study could be further strengthened by the experiments mentioned above. Additionally, see the comment on how to increase the impact of the work (Audience section).

***Advance:** compare the study to existing published knowledge

This study is the first to characterize the role of SDS-22 in the polarization of the *C. elegans* zygote. As the authors discuss, their results align with and complement existing knowledge of SDS-22 in other cell types. Together with the literature, this work highlights the complexity of PP1 regulation, suggesting that different PP1 outcomes may be achieved by combining SDS-22 with various PP1 co-regulators.

Audience that will be interested or influenced by this research

These results will be of interest to scientists studying cell signalling and cell polarity. There is currently strong focus on understanding the regulation of phosphatases. In cell polarity research, the spatial regulation of phosphatases is particularly important for understanding the asymmetric activation of signalling pathways. SDS-22 does not appear to control the spatial localization or activity of PP1, but rather its overall protein levels. As the authors note in the discussion, this suggests that other factors may be involved in the polarization of PP1 signalling. In supplementary figure S1, the authors provide a volcano plot showing candidate PP1 interactors. Providing the list of positive hits would increase the impact of the study and benefit the research community. It would also help explain why the authors chose to follow up on SDS-22 in this study. Furthermore, this could advance the identification of factors involved in the polarization of PP1 signalling.

My expertise

Cell polarity, cell signalling, embryo development.

3. How much time do you estimate the authors will need to complete the suggested revisions:

Estimated time to Complete Revisions (Required)

(Decision Recommendation)

Between 1 and 3 months

4. Review Commons values the work of reviewers and encourages them to get credit for their work. Select 'Yes' below to register your reviewing activity at Web of Science Reviewer Recognition Service (formerly Publons); note that the content of your review will not be visible on Web of Science.

Yes

Dear Monica,

Thank you for transferring your manuscript to EMBO Reports, which was previously reviewed at Review Commons.

Referees express interest in the proposed role of SDS-22 in polarity establishment in *C. elegans* embryos. However, they also raise concerns that need to be addressed to consider publication in EMBO Reports.

Having looked at all documents, we would like to invite you to submit a revised manuscript as in your revision plan. Please revise your manuscript with the understanding that the referee concerns (as in their reports) must be fully addressed and their suggestions taken on board. Please address all referee concerns in a complete point-by-point response. Acceptance of the manuscript will depend on a positive outcome of a second round of review. It is EMBO reports policy to allow a single round of major experimental revision only and acceptance or rejection of the manuscript will therefore depend on the completeness of your responses included in the next, final version of the manuscript.

We realize that it is difficult to revise to a specific deadline. In the interest of protecting the conceptual advance provided by the work, we recommend a revision within 3 months. Please discuss the revision progress ahead of this time with me if you require more time to complete the revisions, or if you have questions or comments regarding the revision (also by video chat).

1. A data availability section providing access to data deposited in public databases is missing (where applicable).
2. Your manuscript contains statistics and error bars based on $n=2$. Please use scatter plots in these cases.

You can submit the revision either as a Scientific Report or as a Research Article. For Scientific Reports, the revised manuscript can contain up to 5 main figures and 5 Expanded View figures, and it should not exceed 27000 characters. If the revision leads to a manuscript with more than 5 main figures it will be published as a Research Article. In this case the Results and Discussion section should be separate. If a Scientific Report is submitted, these sections have to be combined. This will help to shorten the manuscript text by eliminating some redundancy that is inevitable when discussing the same experiments twice. In either case, all materials and methods should be included in the main manuscript file.

4) a .docx formatted letter INCLUDING the reviewers' reports and your detailed point-by-point responses to their comments. As part of the EMBO publication's Transparent Editorial Process, EMBO reports publishes online a Review Process File (RPF) to accompany accepted manuscripts. This File will be published in conjunction with your paper and will include the referee reports, your point-by-point response and all pertinent correspondence relating to the manuscript.

<https://www.embopress.org/page/journal/14693178/authorguide#transparentprocess>

5) a complete author checklist, which you can download from our author guidelines <https://www.embopress.org/page/journal/14693178/authorguide>. Please insert information in the checklist that is also reflected in the manuscript. The completed author checklist will also be part of the RPF.

6) Please note that all corresponding authors are required to supply an ORCID ID for their name upon submission of a revised manuscript (). Please find instructions on how to link your ORCID ID to your account in our manuscript tracking system in our Author guidelines

Additional information on source data and instruction on how to label the files are available:
<https://www.embopress.org/page/journal/14693178/authorguide#sourcedata>

9) Our journal encourages inclusion of *data citations in the reference list* to directly cite datasets that were re-used and obtained from public databases. Data citations in the article text are distinct from normal bibliographical citations and should directly link to the database records from which the data can be accessed. In the main text, data citations are formatted as follows: "Data ref: Smith et al, 2001" or "Data ref: NCBI Sequence Read Archive PRJNA342805, 2017". In the Reference list, data citations must be labeled with "[DATASET]". A data reference must provide the database name, accession number/identifiers and a resolvable link to the landing page from which the data can be accessed at the end of the reference. Further instructions are available at <http://www.embopress.org/page/journal/14693178/authorguide#referencesformat>

12) Please also note our reference format:
<http://www.embopress.org/page/journal/14693178/authorguide#referencesformat>

13) All Materials and Methods need to be described in the main text using our 'Structured Methods' format, which is required for all research articles. According to this format, the Methods section includes a Reagents and Tools Table (listing key reagents,

experimental models, software and relevant equipment and including their sources and relevant identifiers) followed by a Methods and Protocols section describing the methods using a step-by-step protocol format. The aim is to facilitate adoption of the methodologies across labs. More information on how to adhere to this format as well as a downloadable template (.docx) for the Reagents and Tools Table can be found in our author guidelines:

I look forward to seeing a revised version of your manuscript when it is ready. Please let me know if you have questions or comments regarding the revision.

Kind regards,

Deniz

Deniz Senyilmaz Tiebe, PhD
Senior Scientific Editor
EMBO Reports

We thank the reviewers for their insightful comments that helped us improve and clarify our manuscript. We agree with Reviewer 1 and 3 that SDS-22 has more general functions in cellular processes by maintaining GSP-1/2 levels, rather than only regulating cell polarity. We have now modified our conclusion accordingly in the text (all changes are highlighted in yellow) and we hope that it is now clearer and better explained.

We have quantified the endogenous levels of GSP-1 and OLLAS-tagged GSP-2, both of which are reduced in SDS-22-depleted or mutant backgrounds, consistent with observations made using GFP-tagged strains. Furthermore, by depleting other proteasomal subunit, we achieved a mild inhibition of proteasomal function, which led to increased GSP-1 and GSP-2 levels in the embryos, and rescued PAR-2 domain defects in SDS-22(E153A) mutant embryos. Despite our attempts to measure phosphorylation levels of GFP-PAR-2 we did not obtain clear (and quantifiable) results. We thank the reviewer for having labelled the experiment as optional.

Below we include a point-by-point response to the comments of the reviewers.

Reviewer #1 (Evidence, reproducibility and clarity (Required)):

Major Comments:

(1) Overall, the evidence supporting the core finding that SDS-22 is required for normal GSP-1/2 levels is strong and well documented. The experiments were performed well and controls, statistics, replicates were appropriate. Our only slight reservation was whether the effect of *sds-22*(RNAi) on stability may be overstated due to the use of GFP fusions to GSP-1/2 for this analysis. The authors note these alleles are hypomorphic, potentially raising the possibility that GFP tags destabilise the proteins and make them more prone to degradation. Ideally this would be repeated with an untagged allele via Western (e.g. Peel et al 2017 for relevant antibodies).

We thank the reviewer for the general comments, which is indeed very important. To address this point we have requested GSP-1 and GSP-2 antibodies reported in Peel et al and Tzur et al (Peel *et al*, 2017; Tzur *et al*, 2012). The published GSP-1 antibody has been used in Western blot, and the GSP-2 antibody has been used in both immunostaining and Western blot analysis. Despite our efforts, we were not able to detect GSP-2 neither on Western blots nor on immunostainings with the aliquot we have received. On the opposite, GSP-1 antibodies worked well on Western blot.

We have measured the GSP-1 levels in SDS-22 depleted embryos (N = 3, Fig. 6D,F) and we observed reduced levels, confirming our initial result. The reduction is lower compared to the one observed in the GFP strain, suggesting that the GFP fusion contributes in part to the instability. We have also measured the GSP-1 levels in *sds-22*(E153A) embryos and, consistently, we saw a similar reduction (N = 3, Fig. 6J,L).

Since we were unable to detect endogenous GSP-2, we have generated an OLLAS-tagged GSP-2 strain. OLLAS is a commonly used tag consisting of 14 amino acids (Park *et al*, 2008), with an additional 4 amino acids as a linker, a tag much smaller than mNeonGreen (approximately 270 amino acids). We have crossed the strain with *sds-22*(E153A) and found that the strain exhibited 37.0% lethality (compared to 99.1% of the *mNG::gsp-2; sds-22*(E153A)) and can be maintained in the homozygote form, which was not the case with *mNG::gsp-2; sds-22*(E153A) (Fig. EV3B,C). We have measured the OLLAS-GSP-2 levels in SDS-22 depleted and *sds-22*(E153A) embryos and we observed a reduction of around 20% (N = 3, Fig. 6D,E,J,K).

These data have now been added in Fig 6 and the western blots with the GFP strains have been removed. We have also added a sentence to the results stating the potential caveat of measuring GFP fusions (see Results, Line 326-328, page 13).

(2) The role for SDS-22 in polarity is rather weak. Both the SDS-22 depletion phenotypes and the ability of SDS-22 depletion to suppress *pkc-3*(ts) polarity phenotypes are modest (and weaker in than GSP-2 depletion). For example, the images in Figure 1B appear striking, but from Movie S1 it is clear that this isn't a full rescue as PAR-2 is initially uniformly enriched on the cortex (rather than mostly cytoplasmic) and it is never fully cleared. In the movie, the clearance at the point of pronuclear meeting is very modest. Quantitation might be helpful here (i.e. as in Figure 3G). As the authors state, it seems that SDS-22 does not have a specific role in polarity beyond the general effect on GSP-1/2

levels. This does not undermine the core message of the paper, but we would recommend downplaying the conclusions with respect to contributing to polarity establishment. For example "...suggesting that SDS-22 regulates GSP-1/-2 activity to control the loading of PAR-2 to the posterior cortex in one-cell stage *C. elegans* embryos" implies a regulatory role for SDS-22 in polarity, but we would interpret it as simply helping reduce aberrant degradation of GSP-1/2 and this impacts a variety of cellular processes including polarity.

We agree with the reviewer that the rescue of *pkc-3ts* polarity defects by SDS-22 depletion is not as strong as GSP-2 depletion, and as suggested, we have re-quantified the phenotype, as we did in Fig. 3G and added the quantification in Fig. 1C.

To fully address this comment, we have clarified the role of SDS-22 in the text in several locations (all highlighted in the text). We have added "partial" rescue and modified conclusions in the results and discussion. The changes are all highlighted and the major ones are also below:

From

To conclude, while our genetic data on PAR-2 cortical localization suggest that SDS-22 is not required to fully activate GSP-1 and/or GSP-2, depletion or mutation of SDS-22 results in a reduced activity of the phosphatases.

To: Result Line 311-314, page 12

To conclude, while our genetic data on PAR-2 cortical localization suggest that SDS-22 is not required to fully activate GSP-1 and/or GSP-2, depletion or mutation of SDS-22 results in a reduced activity of the phosphatases, as shown by phospho-histone H3 (Ser10) levels. These data show that the regulation of GSP-1/-2 by SDS-22 is not specific to cell polarity.

We have also rephrased our conclusion according to Reviewer 1's suggestion.

From Introduction

Here we show that SDS-22 depletion rescues the polarity defects caused by reduced PAR-2 phosphorylation in the *pkc-3(ne4246)* mutant at the semi-restrictive temperature (24°C), similarly to the depletion of GSP-2. Depletion of SDS-22 results in lower GSP-1 and GSP-2 protein levels which can be rescued by depleting proteasomal subunits. These results establish SDS-22 as a regulator of PAR polarity establishment in the *C. elegans* one-cell embryo and are consistent with and complement the recent data in mammalian cells showing that SDS22 is important to control the stability of the PP1 phosphatase (Cao *et al.*, 2024).

To: Introduction Line 95-99, Page 4

*Here we show that SDS-22 depletion results in lower GSP-1 and GSP-2 levels and in a partial rescue of the polarity defects caused by reduced PAR-2 phosphorylation in the *pkc-3(ne4246)* mutant at the semi-restrictive temperature (24°C). These results establish that SDS-22 contributes to cell polarity by regulating GSP-1/-2 levels and are consistent with and complement the recent data in mammalian cells showing that SDS22 is important to control the stability of the PP1 phosphatase (Cao *et al.*, 2024).*

From Discussion:

Depletion of SDS-22, or mutation of its E153 residue (E153A) important for SDS-22-PP1 interaction resulted in reduced GSP-1/-2 protein levels, decreased dephosphorylation of a PP1 substrate, and a smaller PAR-2 domain, suggesting that SDS-22 regulates GSP-1/-2 activity to control the loading of PAR-2 to the posterior cortex in one-cell stage *C. elegans* embryos.

To: Discussion Line 456-458, page 17

*Here we find that a conserved PP1 regulator, SDS-22, when depleted, results in a smaller PAR-2 domain and can partially rescue the polarity defects of a *pkc-3(ne4246)* mutant. We demonstrate that SDS-22 contributes to the activity of GSP-1/-2 by maintaining their protein levels.*

Add new discussion to Discussion Line 458-461, page 17:

*These data suggest that the role of SDS-22 in polarity is indirect via the regulation of GSP-1/-2 levels. In support of this, SDS-22 depletion results in broader GSP-1/-2 dependent phenotypes such as increased Phospho-H3 (Ser10) (Fig 5) and centriole duplication defects in later-stage embryos (Peel *et al.*, 2017).*

(3) Specificity of SDS-22 effects on polarity. SDS-22 (or GSP-1/2) depletion is likely to have effects on many pathways. We wondered whether some of the polarity phenotypes may not be specifically due to changes in the PAR-2 phosphorylation cycle as implied.

One candidate is the actomyosin cortex. It was noticeable that control and *sds-22* embryos were different: In Movies S1, S2, and S3 control embryos show either stronger or more persistent cortical ruffling or pseudocleavage furrows. This is also visible in Figure 3A. Is it possible that disruption of SDS-22 reduces cortical flows (time, intensity or duration) and could this explain the small reduction in anterior PAR-2 spreading and thus the slightly smaller domain size measured in Figures 1B and 3A.

We have noticed that SDS-22 depletion results in less ruffling and a reduced pseudocleavage furrow. To properly address this question, we should have a condition in which we can rescue the cortical flow reduction in the SDS-22 depletion and measure the PAR-2 domain. Since we do not know how SDS-22 reduces the flows, we could not come up with a clean experiment to address this question. We have addressed this limitation in the text, by adding in the discussion:

Discussion Line 493-496, page 19:

Consistent with GSP-2 reduced levels, SDS-22 depleted or E153A mutant embryos also have a smaller PAR-2 domain. However, since these embryos also show reduced cortical ruffling (Movie EV1,2) and are smaller (Fig EV1C) we cannot exclude that these two phenotypes also contribute to the smaller size of the PAR-2 domain.

A potentially related issue could be embryo size. *sds-22* embryos generally seem to be smaller than wild-type (e.g. Figure 1B(left), 4A(left column), and particularly EV3). Is this consistently true? Could cell size effects change the ability of embryos to clear anterior PAR-2 domains as described in EV3? Klinkert et al (2018, bioRxiv) note that reducing the size of *air-1*(RNAi) embryos reduces the frequency of bipolar PAR-2 domains.

We have quantified the perimeter of the embryos and as seen by the quantification, there is a weak but significant decrease of size in the absence of SDS-22, and in the SDS-22(E153A) mutant. We have now added the data of the RNAi in the supplementary information and mentioned it in the results (line 127, page 5) and the discussion (line 495, page 19).

Results Line 127, page 5:

SDS-22 depleted embryos also displayed a smaller size (Fig EV1C).

Klinkert et al reported that reducing the size of *air-1*(RNAi) embryos by depletion of ANI-2, a homolog of the actomyosin scaffold protein anillin, reduces the frequency of bipolar PAR-2 domains (Klinkert et al, 2018). In the image shown in the paper on bioRxiv, the PAR-2 domain appears small but there are no quantifications and these data have been removed from the published paper.

From published data, a smaller embryo size does not appear to correlate with smaller PAR-2 domain. Chartier et al show that depletion of ANI-2 reduces embryo size without changing the relative anterior PAR-6 domain (Chartier et al, 2011), thereby suggesting that the posterior PAR-2 domain should not change either. In addition, Hubatsch et al reported that small embryos depleted of *ima-3* tend to have larger PAR-2 domains, whereas larger embryos depleted of *C27D9.1* exhibit smaller PAR-2 domains (Hubatsch et al, 2019), which is the opposite of what we see. We do not believe that the smaller PAR-2 domain is the important message of our paper. Our main question was whether PAR-2 was cortical or not and since GSP-2 had a smaller domain, we decided to quantify the PAR-2 domain length in the different RNAi conditions and mutants. Since RNAi of *C27D9.1* which makes embryos bigger, results in a small PAR-2 domain, again we do not know how to experimentally address this question. As for the point above, we have highlighted this limitation in the discussion (see our reply to the previous point, now it is in Discussion Line 494-496, page 19).

We would stress that these comments relate to interpreting the polarity phenotypes and do not undermine the core finding that SDS-22 stabilises GSP-1/2.

We thank the reviewer and we hope that with the experiments we have performed and the text changes, their comments are properly addressed.

Minor Points

- The link between lethality and polarity of the zygote is not always obvious and whether they are connected (or not) could probably be made clearer. Indeed, the source of lethality is unclear, particularly given that loss of SDS-22 on its own strongly impacts lethality with minimal effects on polarity (at least in the zygote).

We apologize for the lack of clarity. In many cases, we have reported embryonic lethality as information, not with a precise scope to correlate the lethality with the phenotype. We know that embryonic lethality is normally associated with severe polarity defects. As example, in the *par-2(RAFA)* mutant and in the *pkc-3ts* mutant at temperatures around 24-25°C cortical polarity is lost, embryos divide symmetrically and synchronously and die (Calvi *et al*, 2022; Rodriguez *et al*, 2017) and many more references for the PAR mutants (Kemphues *et al*, 1988; Kirby *et al*, 1990; Morton *et al*, 1992). We and others have also shown that depletion of GSP-2 can rescue the lethality of *pkc-3(ts)* but only at a semipermissive temperature when there is still residual PKC-3 activity (Calvi *et al.*, 2022; Fievet *et al*, 2013). As our aim was to identify the regulator of GSP-2, we tested the potential regulators by RNAi in the *pkc-3(ts)*, with the assumptions that a regulator, similar to GSP-2, would rescue the *pkc-3(ts)* polarity defects and lethality. As it turns out, SDS-22 is not a canonical regulator of GSP-2. The partial rescue of the polarity defects is most likely the result of the fact that SDS-22 lowers the level of GSP-2. However, SDS-22 is probably involved in many other functions that involve GSP-1 and GSP-2 (as shown for example: (Beacham *et al*, 2022; Peel *et al.*, 2017)) and it is embryonic lethal. We do not know, however, whether the embryonic lethality is the results of the sum of the various functions of SDS-22 or it is due to a specific function.

To clarify it better, we have now explained the connection between polarity and lethality in the text, From Results:

We first asked whether depletion of any of these three regulators suppress the embryonic lethality of *pkc-3(ne4246); gfp::par-2* embryos at the semi-permissive temperature of 24°C (in which PKC-3 is partially active, temperature used in all experiments with the *pkc-3(ne4246)* mutant, unless otherwise stated), similar to depletion of the catalytic subunit GSP-2.

To Results Line 109-115, page 5:

*When the temperature sensitive mutant *pkc-3(ne4246)* is grown at semi-permissive temperature, the residual PKC-3 activity is not sufficient to exclude PAR-2 from the anterior cortex. These embryos cannot establish polarity and die. Depletion of the catalytic subunit GSP-2 in this strain suppresses PAR-2 mislocalization and the resulting polarity defects, and rescues embryonic lethality. We first asked whether depletion of any of these three identified regulators suppresses the embryonic lethality of *pkc-3(ne4246); gfp::par-2* embryos at the semi-permissive temperature of 24°C (temperature used in all experiments with the *pkc-3(ne4246)* mutant, unless otherwise stated) , similar to depletion of GSP-2.*

From Results:

We next asked whether *sds-22(E153A)* was able to rescue the lethality and the polarity defects of *pkc-3(ne4246)* embryos.

To Results Line 219-220, page 9:

*Because of this, we decided to test whether *sds-22(E153A)* was able to rescue the lethality and the polarity defects of *pkc-3(ne4246)* embryos.*

- Formally, the conclusion that reduced GSP-1/2 in SDS-22 depletion conditions is due to increased proteasomal degradation is not shown directly as there is no data on rates just steady-state levels. We agree that the genetic data is strongly suggestive of this model and it is consistent with work of other labs. Thus this is the most likely scenario, but could in principle reflect reduced expression that is balanced by reduced degradation.

We agree with the reviewer. To partially address this point, we have performed RT-PCR analysis to measure the levels of *gsp-1* and *gsp-2* mRNAs in control and SDS-22 depleted embryos. Our results showed no significant differences in *gsp-1* and *gsp-2* transcript levels under SDS-22 depletion (data in Figure EV3A, results Line 329-330).

- It is interesting that *sds-22(E153A)* caused a stronger decrease in oocyte GSP-1 levels than *sds-22(RNAi)* (Fig 7). The authors may want to comment on this result.

As we performed depletion of SDS-22 by RNAi feeding from L4 stage, we might not see strong reduction of GSP-1 in oocytes compared to that in *sds-22(E153A)* mutant, which carries an endogenous mutation of SDS-22 throughout the life cycle.

To address this point, we have performed an experiment in which SDS-22 was depleted starting from the L1 larval stage. As shown in Appendix Fig S7, RNAi feeding of SDS-22 from L1 stage resulted in a strong reduction of GSP-1 levels (38.0% in the cytoplasm; 47.1% in the nucleus). This reduction was more pronounced comparing to RNAi feeding of SDS-22 from L4 stage (8.8% in the cytoplasm; 17.4% in the nucleus, Fig. 7D-F) and comparable to *sds-22(E153A)* mutant (26.7% in the cytoplasm; 33.1% in the nucleus, see also Fig. 7G-I). These findings support our hypothesis that the difference shown in Fig. 7D-I might result from the relatively short RNAi treatment starting at L4, compared to endogenous SDS-22(E153A) mutation. We have now included a corresponding sentence in the text (Result Line 406-409, Page 16) and the data are shown in Appendix FigS7. However, since RNAi depletion of SDS-22 from the L1 stage also impairs germline formation (as shown in panel FigS7A), we kept the protocol using SDS-22 RNAi feeding in L4 worms for all other experiments in this study.

- "At polarity establishment, the PP1 phosphatases GSP-1/2 dephosphorylate PAR-2 allowing its cortical posterior accumulation." This statement, possibly inadvertently, implies temporal regulation, which has not been shown.

We have changed the sentence, as suggested by the reviewer:

To Introduction Line 58-59, page 3:

The PP1 phosphatases GSP-1/2 are required for PAR-2 cortical posterior accumulation and embryo polarization.

- It would be ideal if the authors could explicitly state how they define pronuclear meeting. For example in Figure 1B, the embryos look like they are a few minutes past pronuclear meeting (e.g. compared to Figure 3), but maybe the pronuclei tend to meet more centrally in these conditions? Given that PAR-2 clearance is changing in time in some of these cases (based on looking at the movies), staging needs to be very accurate to get the best comparisons.

We apologize for the lack of clarity. Pronuclear meeting is defined as the first timepoint in which the two pronuclei contact each other.

As noted by Reviewer 1, it is true that the pronuclei in *pkc-3ts* mutant tend to meet more centrally compared to control embryos. The same finding was also observed on PKC-3 inhibition (through depletion, mutation or inhibitor treatment) by Rodriguez et al (Rodriguez *et al.*, 2017). In addition, Kirby et al reported that mutations in the anterior PAR complex lead to meeting more in the center (Kirby *et al.*, 1990). We now specify this in the Material and Methods.

Add in Material and Methods Line 678-682, page 24:

*The stage of pronuclear meeting is defined as the first timepoint in which the two pronuclei contact each other. In *pkc-3(ne4246)* embryos, the two pronuclei exhibited a tendency to meet more centrally compared to controls (Fig 1B, Movie EV1), as shown in (Kirby *et al.*, 1990; Rodriguez *et al.*, 2017).*

As Reviewer 1 mentioned, accurate staging is crucial, as PAR-2 clearance can vary over time. The measurements were done in the first frame where pronuclei touch each other. However, in Fig. 1B we had shown one *pkc-3ts; sds-22(RNAi)* embryo one frame (10 seconds) later. We have now corrected this (see the updated Figure 1B).

- In the interests of data-availability, upon publication the authors would deposit the raw mass spec data underlying Figure EV1.

The reviewer is right, this was forgotten. We have now added as supplementary material the Dataset EV1 and EV2. Due to limited space of EV figures, we have now moved the mass spec graph to Appendix Fig S1.

****Referees cross-commenting****

We also generally agree with the comments of the other reviewers.

Our only concern with the main conclusion regarding GSP stabilization of GSP-1/2 is the impact of the gfp tags. Given that antibodies exist and have been used in Peel et al 2017 for exactly this purpose in *C. elegans* embryos, this does not seem excessively burdensome in our view and would strengthen the paper.

The remainder of our concerns can likely be addressed by modifications to the text and/or adding a caveats/limitations section to their discussion. As we noted, these mostly relate to the magnitude and specificity of the impact of SDS-22 on polarity and PAR-2 phosphorylation, which in our view is rather peripheral to the core conclusion (i.e. that SDS-22 stabilizes GSP-1/2).

Reviewer #1 (Significance (Required)):

Overall, this is a careful and well-executed study identifying a conserved role for SDS-22 in stabilising PP1 catalytic subunits in *C. elegans* embryos and shows that this can broadly impact PP1 activity in this system. A mechanistic role for SDS-22 in PP1 function was recently demonstrated in (Cao et al, 2024), where it was shown to stabilise nascent catalytic subunits, but also in subunit recycling (Kuetsch et al 2024). The data here suggest this role in stabilisation PP1 subunits is broadly relevant.

These data are also consistent with prior work from the lab demonstrating the role of PP1 in *C. elegans* zygote polarity. It adds to previous reports that compromised PP1 activity can impact cell polarity and further highlights the importance of considering regulation of protein phosphatases in cell polarity pathways. That said, the impact on polarity is rather modest, likely reflecting a general requirement for SDS-22 in supporting optimal PP1 activity rather than any specific role in polarity.

Field of expertise: cell polarity, cell and developmental biology.

Reviewer #2 (Evidence, reproducibility and clarity (Required)):

Summary: The authors present a logical and clear set of data that support the model that SDS-22 is an important regulator of cell polarity via its ability to stabilize Protein Phosphatase 1 (PP1).

The authors use a clever combination of genetic manipulations and quantitative imaging to show that loss of SDS-22 phenocopies loss of PP1, in that PAR-2 polarization is restored in nascent *C. elegans* zygotes following inactivation of PKC-3. Rescue of PAR-2 polarization also occurs when the authors mutate a conserved residue in SDS-2 that is predicted to form electrostatic interactions with PP1, suggesting that SDS-22 acts via PP1. Inactivation of SDS-22 results in decreased levels of PP1, and the authors provide evidence that this is via proteasomal degradation by showing that PP1 levels are restored by knockdown of proteasomal subunits.

Overall the manuscript is well-written, the experiments rigorous, and the methods and data likely to be reproducible.

Major comment: Consistent with the model that PP1 activity is reduced in the absence of SDS-22, the authors show that a surrogate PP1 target (phospho-histone H3) becomes hyper-phosphorylated. To strengthen the study, the authors could consider performing an OPTIONAL experiment (see below) of assaying the phosphorylation status of PAR-2 itself, as this is proposed to be the target of both PKC-3 and PP1, and represent the mechanism of PAR-2 polarization.

We thank the reviewer for this comment and also for pointing out that there is technical difficulty in the proposed experiment.

We have previously attempted to address this point in Calvi et al (Calvi *et al.*, 2022) using Western blot analysis, but without success (see below the data that we put in the response letter of Calvi et al, panel A and part of panel B). For this we used the *gfp::par-2* CRISPR strain and used a GFP antibody (shown below in panel A), as none of the anti-PAR-2 antibodies (neither the ones produced by us nor the ones produced by other laboratories) are working on Western blot. We observed several bands of

GFP::PAR-2 but were not able to determine if these represented phosphorylated forms or to compare the ratio of phosphorylated to unphosphorylated PAR-2. We did use λ -PPase in the embryonic extracts but we did not always observe a clear difference. We show three experiments below in panel A.

One possible explanation is that the role of GSP-1/-2 in PAR-2 dephosphorylation is specific to the very early embryos. As shown in panel B, despite PAR-2(RAFA) remaining cytoplasmic in one- and two-cell embryos, it can localize to internal cortices in four-cell stage embryos, similarly to the control. This suggests that in later embryos other mechanisms may intervene in PAR-2 cortical regulation. One limitation of our Western Blot is that it is not possible to isolate only early embryos, which are a minority in a mixed population of embryos. This may mask difference of phosphorylation status of PAR-2 in the early stages.

Despite our failure with *gsp-1/-2* RNAi, we have blotted PAR-2 using GFP antibody in *gfp::par-2* embryo lysates, with both control and *sds-22(RNAi)* treatment. In addition, we have also compared the GFP::PAR-2 bands between *gfp::par-2* and *gfp::par-2; sds-22(E153A)* mutant samples. As the reviewer can see in the blot, we could not observe a quantifiable difference between the different samples (see panels C and D).

In Hao et al (Hao *et al*, 2006) the authors used a transgene strain of PAR-2. We have depleted SDS-22 in a PAR-2 transgene but for some unknown reasons the GFP antibodies (the same we used for the blot shown below) did not detect anything (despite the fact that we could see GFP::PAR-2 under the microscope).

Figure for referee with unpublished data and its description has been removed upon request by the authors.

****Referees cross-commenting****

In principle I agree with many of the thoughtful comments by the other reviewers. They point out many potential areas for both enhancing the strength of the findings and including a more nuanced

interpretation of the results. However, I also feel that the experiments proposed to deal with their concerns might not be so straightforward to pursue for unforeseen technical reasons and may actually take substantially longer than anticipated. The same is true for my proposed experiment to assess phosphorylation status of PAR-2, which is why I have indicated it as optional. I ask the other reviewers to consider if any of their proposed experiments might also be considered optional. I also thank them for their critical assessments of the paper! They were helpful for me and I'm sure will also be helpful for the other authors.

Reviewer #2 (Significance (Required)):

This study brings clarity to the contentious role of SDS-22 by showing that it appears to promote PP1 activity by counteracting the phosphatase degradation process in vivo. This complements a previously hypothesized function of SDS-22, while contrasting with other proposed functions of SDS-22 as a regulator of PP1 localization or stimulator of PP1 degradation. Thus, the authors' discoveries in *C. elegans* represent a significant advance in our understanding of protein phosphatase regulation, a long-standing question in biology and a central process in all cellular systems. The study also points to potential mechanisms for modulating phosphatase activity in other contexts, across different organisms and disease states. Basic science researchers will be interested in the findings, with potential to attract additional interest from physiologists and even drug designers.

One limitation of the study is that the authors use PAR-2 polarization as a readout of PKC-3 and PP1 activity without showing directly that PAR-2 phosphorylation status is changing in response to their genetic manipulations, including SDS-22 inactivation. The PAR-2 membrane localization is thought to be inhibited by PKC-3-dependent phosphorylation and promoted by PP1-dependent dephosphorylation. Is there a possibility of examining whether PAR-2 phosphorylation is elevated in SDS-22 RNAi or mutant animals? Previously, Hao et. al., 2006; doi.org/10.1016/j.devcel.2005.12.015. showed in Figure 2 that PAR-2 runs as a doublet band on Western blots with the phosphorylated form of PAR-2 appearing to correlate with the slightly higher molecular weight band. This was used to infer the ratio of phosphorylated to dephosphorylated PAR-2. I'm wondering if it might be possible for the authors to perform a similar analysis of their existing GFP::PAR-2? It appears from their previous paper on PP1 regulation of PAR-2 polarization (Calvi et. al., 2022; doi: 10.1083/jcb.202201048.) that they might also be detecting a similar doublet (Figure S5F and associated source file), so perhaps it is a doable experiment? Regardless, this is not an essential experiment as the study is already significant and rigorous!

I am a cell biologist that uses *C. elegans* to understand the function of conserved protein complexes that regulate the development and function of animal tissues.

Reviewer #3 (Evidence, reproducibility and clarity (Required)):

Summary: your understanding of the study and its conclusions

This is a follow up on Gotta's lab paper, which shows that the PP1 catalytic subunits GSP-1 and GSP-2 are involved in the polarization of the *C. elegans* zygote (10.1083/jcb.202201048). Here, the authors report that SDS-22, an interactor of PP1, regulates PP1 function in the zygote. Depleting SDS-22, similar to depleting GSP-2, rescues the polarity defects caused by the inactivation of aPKC in the zygote. This suggests that SDS-22 plays a role in promoting GSP-2's function in polarity. The mechanism behind this may involve SDS-22 protecting GSP-1 and GSP-2 from degradation by the proteasome.

Major comments: major issues affecting the conclusions

Overall, the authors' conclusions are supported by their data. The data and methods are presented clearly, with appropriate replicates and statistics. Here I propose two experiments to strengthen the link between some of their data and their claims. These experiments could take a month or two to complete.

Experiment 1

It would be helpful if the authors could show that blocking the proteasome in the zygote restores GSP-1/-2 levels in the absence of SDS-22 or even better in the SDS-22(E153A) mutant. This would provide more direct evidence to support their claim that SDS-22 regulates polarity by protecting PP1 from proteasomal degradation. While they are currently conducting this experiment in the germline, they cannot assess polarity there. However, in the zygote, they would be able to examine the PAR-2 domain (polarity). To do this, the authors could permeabilise the embryos and apply a proteasome inhibitor.

This would be a straightforward experiment if we were using culture cells but it is a challenging experiment in the *C. elegans* embryo for different reasons. One problem with the set up is that much of the protein of the one-cell embryo is inherited from the oocyte and the reduction in SDS-22 depletion or mutant happens already in the germline (Fig. 6-7). Even if the proteasome function is inhibited in embryos, the whole division process only takes 20 minutes (and polarity establishment about 10 minutes) and we wonder whether the timing will be sufficient to effectively inhibit the proteasome, produce more protein and rescue the phenotype (knowing that proteasome inhibition has consequences on cell division (Dou *et al*, 2003) and also on cell polarity (Sugiyama *et al*, 2008). One alternative approach would be to apply the proteasome inhibitor to adult worms in liquid culture for several hours before dissection. This would aim to inhibit degradation in the germline, therefore allowing us to test whether GSP-1/-2 levels are restored in the embryos with SDS-22 disruption. However, proteasome inhibition in the germline impairs oogenesis (Shimada *et al*, 2006), suggesting that we might incur in the same problem.

To overcome this challenge, we tested a milder strategy by depleting proteasomal subunits whose knockdown impairs proteasome activity to a lesser extent than RPN-6 or RPN-7. As reported by Fernando *et al* (Fernando *et al*, 2022), depletion of RPN-9, -10, or -12 impairs proteasomal activity, but worms remain fertile. We tested these three subunits and our data indicate that RPN-12 depletion led to a mild but significant increase in both GSP-1 and GSP-2 levels in embryos (data shown in Fig. EV5A,B).

We have depleted RPN-12 in *gfp::gsp-1; sds-22(E153A)* strain (which can be maintained as homozygote) to test if GSP-1 levels are rescued. Our data showed that similar to germlines (Fig. 7G-I), RPN-12 depletion in *gfp::gsp-1; sds-22(E153A)* rescued the reduction of GSP-1 levels in embryos (see Fig. EV5C,D).

To further investigate if impaired proteasomal activity rescues polarity defects, we assessed the PAR-2 domain in *sds-22(E153A)* mutant following RPN-12 depletion. Our data show that the small PAR-2 domain was rescued (see Fig. EV5E,F). One caveat is that RPN-12 depletion alone causes a longer PAR-2 domain. So, these data suggest that mild reduction of proteasome activity can rescue the length of PAR-2 but other proteasome dependent pathways could be involved. It is interesting that RPN-12 was found by my lab as a suppressor of a *par-2(ts)* mutant (Labbe *et al*, 2006), consistent with a potential regulation of PP1 levels.

Experiment 2

The posterior localization of PAR-2 after co-RNAi of GSP-1 and SDS-22 contrasts with the absence of PAR-2 at the cortex when both GSP-1 and GSP-2 are depleted. This difference may be due to the partial reduction of GSP-2 levels when SDS-22 is depleted, compared to the more substantial reduction of GSP-2 upon GSP-2 RNAi. Have the authors considered combining full depletion of GSP-1 with partial depletion of GSP-2 to see if PAR-2 remains present and localized to the posterior? This experiment could help clarify the discrepancy between the phenotypes and further support the role of SDS-22 in regulating GSP-2 protein levels. Additionally, by titrating PP1, the authors may be able to determine the minimum amount of PP1 needed to establish the PAR-2 domain.

We agree with the reviewer that GSP-2 is only partially reduced in the co-depletion of SDS-22 and GSP-1, which likely explains why we did not observe the absence of PAR-2 as seen in co-depletion of GSP-1 and GSP-2 (Fig. 4).

As suggested by the reviewer, we have depleted GSP-1 by dsRNA injection and performed partial depletion of GSP-2 by feeding worms with a serial dilution (from 1:1 to 1:10) of GSP-2 RNAi bacteria with control bacteria. We achieved approximately 70% reduction of GSP-2 (see Fig. EV4A,B). In all

these conditions, PAR-2 was still able to localize to the cortex. With 30% of GSP-2 remaining in embryos (and about 20% of GSP-1), PAR-2 was polarized, though the PAR-2 domain was reduced in size (Fig. EV4C,D). This phenotype closely resembles that observed in SDS-22 depletion or in co-depletion of SDS-22 with either GSP-1 or GSP-2, suggesting that reduced amounts of GSP-1 and GSP-2 are sufficient to polarize PAR-2. We have now included these data in the manuscript (Figure EV4 and page 15, result line 371-380)

One limitation of this approach is that it is not possible to simultaneously quantify the levels of GSP-1 and GSP-2 and measure the PAR-2 domain size in the same embryo (which would be the most rigorous way to perform the experiment so that we know the amount of depletion and correlate with the PAR-2 domain length). We have performed the same depletion in parallel in the 3 different strains (the *mNG::gsp-2*, the *gfp::gsp-1* and the *gfp::par-2*) and assume that RNAi efficiency is comparable across strains. However, due to variability in RNAi depletion and the inability to directly correlate PP1 levels with PAR-2 domain size in individual embryos, we cannot precisely determine the minimum amount of PP1 required for PAR-2 polarization. It is also interesting that Rodrigues et al reported that the dosage of PAR proteins is not linear with the PAR asymmetry index in embryos (Rodrigues et al, 2024). We therefore cannot identify the threshold level of PP1 required for polarity but we can conclude that a strong reduction is not sufficient to abrogate polarity.

Minor comments: important issues that can confidently be addressed

In the introduction (line 83), it's unclear what reconciles the contradictory data. I also have difficulty understanding this point in the discussion (line 435).

We apologize for the lack of clarity and have now modified the text:

From the Introduction:

This underscores the complex roles of SDS22 in regulating PP1 function and reconciling the contradictory data obtained in vivo and in vitro (Cao et al., 2024; Cao et al, 2022; Kueck et al., 2024; Lesage et al, 2007).

To Introduction Line 80-84, page 4:

These two recent findings suggest that while SDS-22 is required for the biogenesis of PP1 holoenzymes, its removal from PP1 is essential to have an active phosphatase. This dual role of SDS-22 explains how SDS22 behaves as an inhibitor of PP1 in biochemical assays in vitro but as an activator in vivo (Cao et al, 2024; Cao et al, 2022; Kueck et al, 2024; Lesage et al, 2007).

From the Discussion:

These data reconcile the contradictory in vivo and in vitro observations.

To Discussion Line 476-479, page 17:

Given that SDS-22 both stabilizes PP1 levels and inhibits its activity, this dual role clarifies the apparent contradiction: while SDS-22 is essential for PP1 activity in vivo (because it is essential for the biogenesis/stability), it inhibits PP1 activity in vitro (as it needs to be removed to have an active PP1), while in vivo it is removed by p97/Valosin resulting in active PP1.

Additionally, in the results section (line 389), it's not clear why the gonads cannot be studied in the strain with dead embryos. Are the gonads also altered in a way that prevents their observation?

We explained this in the material and methods part, page 22.

To clarify it better in the main text, we have now modified

Results Line 377-378, page 15:

Since depletion of these subunits results in worms with very little to no progeny (Fernando et al., 2022)

Results Line 412-417, page 16:

*Since we use the embryonic lethality phenotype of the *mNG::gsp-2; sds-22(E153A)* strain to recognize the homozygote *sds-22(E153A)*, this precluded the possibility to analyze the germlines of homozygote *mNG::gsp-2; sds-22(E153A)* worms depleted of RNP-6.1 or RPN-7, as these worms do not have progeny (Fernando et al., 2022) and we therefore cannot distinguish the *sds-22(E153A)* homozygote from the *sds-22(E153A)* heterozygote (see material and methods for details).*

Referees cross-commenting

Overall, I agree with the other reviewers' comments. The suggested experiments would help strengthen the connection between SDS-22 and cell polarity, as well as its role in relation to the proteasomal-mediated degradation of GSP-1/-2 and its impact on cell polarity. These experiments seem feasible and could provide stronger support for the authors' claims about these regulatory mechanisms. Alternatively, the authors may consider moderating some of their conclusions if these experiments are not conducted.

Reviewer #3 (Significance (Required)):

General assessment: strengths and limitations

This study enhances our understanding of how phosphatases regulate cell polarity, specifically in the *C. elegans* zygote, a key model system for studying cell polarity. The study could be further strengthened by the experiments mentioned above. Additionally, see the comment on how to increase the impact of the work (Audience section).

Advance: compare the study to existing published knowledge This study is the first to characterize the role of SDS-22 in the polarization of the *C. elegans* zygote. As the authors discuss, their results align with and complement existing knowledge of SDS-22 in other cell types. Together with the literature, this work highlights the complexity of PP1 regulation, suggesting that different PP1 outcomes may be achieved by combining SDS-22 with various PP1 co-regulators.

Audience that will be interested or influenced by this research

These results will be of interest to scientists studying cell signalling and cell polarity. There is currently strong focus on understanding the regulation of phosphatases. In cell polarity research, the spatial regulation of phosphatases is particularly important for understanding the asymmetric activation of signalling pathways. SDS-22 does not appear to control the spatial localization or activity of PP1, but rather its overall protein levels. As the authors note in the discussion, this suggests that other factors may be involved in the polarization of PP1 signalling. In supplementary figure S1, the authors provide a volcano plot showing candidate PP1 interactors. Providing the list of positive hits would increase the impact of the study and benefit the research community. It would also help explain why the authors chose to follow up on SDS-22 in this study. Furthermore, this could advance the identification of factors involved in the polarization of PP1 signalling.

My expertise

Cell polarity, cell signalling, embryo development.

References

Beacham GM, Wei DT, Beyrent E, Zhang Y, Zheng J, Camacho MMK, Florens L, Hollopetter G (2022) The *Caenorhabditis elegans* ASPP homolog APE-1 is a junctional protein phosphatase 1 modulator. *Genetics* 222

Calvi I, Schwager F, Gotta M (2022) PP1 phosphatases control PAR-2 localization and polarity establishment in *C. elegans* embryos. *J Cell Biol* 221

Cao X, Lake M, Van der Hoeven G, Claes Z, Del Pino Garcia J, Lemaire S, Greiner EC, Karamanou S, Van Eynde A, Kettenbach AN *et al* (2024) SDS22 coordinates the assembly of holoenzymes from nascent protein phosphatase-1. *Nat Commun* 15: 5359

Cao X, Lemaire S, Bollen M (2022) Protein phosphatase 1: life-course regulation by SDS22 and Inhibitor-3. *FEBS J* 289: 3072-3085

Chartier NT, Salazar Ospina DP, Benkemoun L, Mayer M, Grill SW, Maddox AS, Labbe JC (2011) PAR-4/LKB1 mobilizes nonmuscle myosin through anillin to regulate *C. elegans* embryonic polarization and cytokinesis. *Curr Biol* 21: 259-269

Dou QP, Smith DM, Daniel KG, Kazi A (2003) Interruption of tumor cell cycle progression through proteasome inhibition: implications for cancer therapy. *Prog Cell Cycle Res* 5: 441-446

Fernando LM, Quesada-Candela C, Murray M, Ugoaru C, Yanowitz JL, Allen AK (2022) Proteasomal subunit depletions differentially affect germline integrity in *C. elegans*. *Front Cell Dev Biol* 10: 901320

Fievet BT, Rodriguez J, Naganathan S, Lee C, Zeiser E, Ishidate T, Shirayama M, Grill S, Ahringer J (2013) Systematic genetic interaction screens uncover cell polarity regulators and functional redundancy. *Nat Cell Biol* 15: 103-112

Hao Y, Boyd L, Seydoux G (2006) Stabilization of cell polarity by the *C. elegans* RING protein PAR-2. *Dev Cell* 10: 199-208

Hubatsch L, Peglion F, Reich JD, Rodrigues NT, Hirani N, Illukkumbura R, Goehring NW (2019) A cell size threshold limits cell polarity and asymmetric division potential. *Nat Phys* 15: 1075-1085

Kemphues KJ, Priess JR, Morton DG, Cheng NS (1988) Identification of genes required for cytoplasmic localization in early *C. elegans* embryos. *Cell* 52: 311-320

Kirby C, Kusch M, Kemphues K (1990) Mutations in the par genes of *Caenorhabditis elegans* affect cytoplasmic reorganization during the first cell cycle. *Dev Biol* 142: 203-215

Klinkert K, Levernier N, Gross P, Gentili C, von Tobel L, Pierron M, Busso C, Herrman S, Grill SW, Kruse K *et al* (2018) Aurora A depletion reveals centrosome-independent polarization mechanism in *C.elegans*. *bioRxiv*: 388918

Kueck AF, van den Boom J, Koska S, Ron D, Meyer H (2024) Alternating binding and p97-mediated dissociation of SDS22 and I3 recycles active PP1 between holophosphatases. *Proc Natl Acad Sci U S A* 121: e2408787121

Labbe JC, Pacquelet A, Marty T, Gotta M (2006) A genomewide screen for suppressors of par-2 uncovers potential regulators of PAR protein-dependent cell polarity in *Caenorhabditis elegans*. *Genetics* 174: 285-295

Lesage B, Beullens M, Pedelini L, Garcia-Gimeno MA, Waelkens E, Sanz P, Bollen M (2007) A complex of catalytically inactive protein phosphatase-1 sandwiched between Sds22 and inhibitor-3. *Biochemistry* 46: 8909-8919

Morton DG, Roos JM, Kemphues KJ (1992) par-4, a gene required for cytoplasmic localization and determination of specific cell types in *Caenorhabditis elegans* embryogenesis. *Genetics* 130: 771-790

Park SH, Cheong C, Idoyaga J, Kim JY, Choi JH, Do Y, Lee H, Jo JH, Oh YS, Im W *et al* (2008) Generation and application of new rat monoclonal antibodies against synthetic FLAG and OLLAS tags for improved immunodetection. *J Immunol Methods* 331: 27-38

Peel N, Iyer J, Naik A, Dougherty MP, Decker M, O'Connell KF (2017) Protein Phosphatase 1 Down Regulates ZYG-1 Levels to Limit Centriole Duplication. *PLoS Genet* 13: e1006543

Rodrigues NTL, Bland T, Ng K, Hirani N, Goehring NW (2024) Quantitative perturbation-phenotype maps reveal nonlinear responses underlying robustness of PAR-dependent asymmetric cell division. *PLoS Biol* 22: e3002437

Rodriguez J, Peglion F, Martin J, Hubatsch L, Reich J, Hirani N, Gubieda AG, Roffey J, Fernandes AR, St Johnston D *et al* (2017) aPKC Cycles between Functionally Distinct PAR Protein Assemblies to Drive Cell Polarity. *Dev Cell* 42: 400-415 e409

Shimada M, Kanematsu K, Tanaka K, Yokosawa H, Kawahara H (2006) Proteasomal ubiquitin receptor RPN-10 controls sex determination in *Caenorhabditis elegans*. *Mol Biol Cell* 17: 5356-5371

Sugiyama Y, Nishimura A, Ohno S (2008) Symmetrically dividing cell specific division axes alteration observed in proteasome depleted *C. elegans* embryo. *Mech Dev* 125: 743-755

Tzur YB, Egydio de Carvalho C, Nadarajan S, Van Bostelen I, Gu Y, Chu DS, Cheeseman IM, Colaiacovo MP (2012) LAB-1 targets PP1 and restricts Aurora B kinase upon entrance into meiosis to promote sister chromatid cohesion. *PLoS Biol* 10: e1001378

Dear Monica,

Thank you for submitting your revised manuscript. As you know, it has now been seen by all original referees and all three recommend publication after a few minor concerns have been addressed.

However, the editorial points below need to be addressed before I can accept the manuscript.

Please provide a point-by-point response to the referee and also to editorial concerns to speed up manuscript checks.

• We note that Moura et al, 2024 is a preprint, which should be cited accordingly. Citations to manuscripts posted on recognized preprint servers can be cited the following way:

In-text citation: (preprint: NAME1 et al, YEAR)

In the reference list: Author NAME1, Author NAME2, (YEAR) article title. bioRxiv doi: 1234/002.dfg123 [PREPRINT]

- Please remove the figures from the manuscript text, but their legends should be placed at the end of the manuscript.
- Please provide 3-5 keywords for your study. These will be visible in the html version of the paper and on PubMed and will help increase the discoverability of your work.
- Please remove Author Contributions from the manuscript text.
- We note the phrase "our unpublished data" in the text, which is not allowed as per journal policy. Please either show the data or remove the statement.
- Funding information needs to be part of Acknowledgments, Funding section heading should be removed.
- The legends of EV should be removed from the figure files and placed after the main figure legends at the end of the manuscript.
- Some error messages pop up upon opening DATASET EV files. Please double check.
- We note the following regarding the Appendix file:
 - o Please remove the two tables from the Appendix. Please submit Table EV2 as a separate file as Dataset EV. Please include its legend in a separate tab of the Excel file and please remember to update all callouts and file type.
 - o I think Table EV1 in the Appendix file should be a part of Reagents & Tools table.
 - o Please add page numbers and a Table of Contents where all items and their page numbers are listed.
- As per the movies, please remove their legends from the manuscript. Instead, each movie should be zipped up with its corresponding legend (in a readme file) so that we have 3 zip folder uploaded.
- Please rename the Material and Methods section as Methods.
- During our routine figure check, we note a possible cell reuse between Figure 1B and Figure EV1B. Please clarify, also in the legends of both affected panels.
- Our production/data editors have asked you to clarify several points in the figure legends - Figure Legends (main + EV):
 - o Please define the annotated p values ****/**/*/* as well as provide the exact p-values for the same in the legend of figure EV1 C as appropriate.
 - o Please note that the exact p values are not provided in the legends of figures 4B, 5A, B; 7B, C, E, F, H, I; EV3 A-D; EV4 B, D; EV5 A, B, D, F. *I note that the current Table EV2 in the Appendix file (which is to be converted into a Dataset EV, please see the relevant point above) is the table of p values. You can also refer to the table in the relevant legends.
 - o Please indicate the statistical test used for data analysis in the legend of figure EV1 C
- Papers published in EMBO Reports include a 'synopsis' and 'bullet points' to further enhance discoverability. Both are displayed on the html version of the paper and are freely accessible to all readers. The synopsis includes a short standfirst summarizing the study in 1 or 2 sentences (max 35 words) that summarize the paper and are provided by the authors and streamlined by the handling editor. I would therefore ask you to include your synopsis blurb and 3-5 bullet points listing the key experimental findings.
- In addition, please provide an image for the synopsis. This image should provide a rapid overview of the question addressed in the study but still needs to be kept fairly modest since the image size cannot exceed 550 (width) x 300-600 (height) pixels.

Thank you again for giving us to consider your manuscript for EMBO Reports, I look forward to your minor revision.

Kind regards,

Martina

--

Martina Rembold, PhD
Senior Scientific Editor
EMBO Reports

=====

Referee #1:

Overall, the authors have addressed the concerns raised in review. It was particularly useful to see the analysis using non-GFP/mNG tagged alleles as this alleviates our only major concern. We also appreciate the adjustments to language regarding conclusions. We have only a few minor comments that the authors may wish to address in revision.

Specific Comments:

Figure 1C, 3G: Reducing PP1 phosphatase activity in embryos should increase the total proportion of phosphorylated PAR-2 and hence reduce PAR-2 membrane binding. However, in the *pkc-3(ne4246)* background, posterior membrane levels of GFP::PAR-2 are higher in *sds-22(RNAi)* or *sds-22(E153A)* than controls (Fig 1C, 3G). Similar results were also seen in *gsp-2(RNAi)* (Fig 1C of Calvi et al., 2022). The authors may wish to comment on this somewhat paradoxical observation. It obviously fits with the idea that *gsp-2(RNAi)* partially rescues polarization, but one might not necessarily expect membrane binding to actually increase when the phosphatase is depleted.

Line 136: show > shows

Line 320: To conclude, while our genetic data on PAR-2 cortical localization suggest that SDS-22 is not required to fully activate GSP-1 and/or GSP-2. (I don't understand this statement - see below line 493)

Line 332: *C.elegans* > *C. elegans*

Line 493: Similar to above "We show that SDS-22 is not essential for full activation of both GSP-1 and GSP-2, since depletion of SDS-22 does not abolish polarity"

This wording is odd given that the data show that GSP-1/2 activity/levels is likely reduced when SDS-22 is depleted/mutant. It is just that without SDS-22 they don't reach their full potential. I would rephrase to "GSP-1/2 activity does not strictly require SDS-22, though SDS-22 is required for them to reach their full, wild-type levels." or something like that

Line 505: Consistent with GSP-2 reduced levels should be with reduced levels of GSP-2

Dataset EV1 and EV2 - It would be helpful to include a description these files (perhaps we missed it).

Figure EV4 - Perhaps worth noting Conte et al., *Curr. Protoc. Mol. Biol.*, 2015 which note that feeding by RNAi is less effective following previous injection of dsRNA. Might suggest that the magnitude of *gsp-2*(partial) is different between EV4A-B and EV4C-D. But the fact that the goal was partial depletion means this is a very minor point.

Note - embedded figures have various problems with fonts that are not seen in the attached figures at the end. Worth making sure there isn't a formatting issue.

Referee #2:

The authors have dedicated substantial effort in addressing all of the reviewers concerns. The manuscript is more clear and the conclusions better justified. I fully support publication. I noted that Figure 6 and the associated caption lack some p value notations, even though I see them in the source file. Overall I thank the authors for making it a pleasure to evaluate this revised manuscript.

Referee #3:

This paper builds upon previous work from Gotta's lab (10.1083/jcb.202201048), which demonstrated that the PP1 catalytic subunits GSP-1 and GSP-2 are involved in *C. elegans* zygote polarisation. In the current study, the authors show that SDS-22, a known interactor of PP1, regulates PP1 activity in the zygote. Notably, depletion of SDS-22-similar to GSP-2 depletion-rescues polarity defects caused by aPKC inactivation. This suggests that SDS-22 promotes GSP-2's function in polarity establishment. The proposed mechanism is that SDS-22 protects GSP-1 and GSP-2 from proteasomal degradation. The manuscript has undergone a first round of revisions, during which the authors have addressed most of the reviewers' major concerns. I am satisfied with the additional experiments and revisions, particularly those clarifying SDS-22's role in stabilizing GSP-1 and GSP-2 protein levels, and how this contributes to zygote polarisation. In areas where minor concerns could not be fully addressed, the authors have acknowledged these limitations and provided clarifications in the text. Overall, the study is carefully conducted, the conclusions are well supported, and the manuscript is clearly written. I find it suitable for publication.

Referee #1:

Overall, the authors have addressed the concerns raised in review. It was particularly useful to see the analysis using non-GFP/mNG tagged alleles as this alleviates our only major concern. We also appreciate the adjustments to language regarding conclusions. We have only a few minor comments that the authors may wish to address in revision.

Specific Comments:

Figure 1C, 3G: Reducing PP1 phosphatase activity in embryos should increase the total proportion of phosphorylated PAR-2 and hence reduce PAR-2 membrane binding. However, in the *pkc-3(ne4246)* background, posterior membrane levels of GFP::PAR-2 are higher in *sds-22(RNAi)* or *sds-22(E153A)* than controls (Fig 1C, 3G). Similar results were also seen in *gsp-2(RNAi)* (Fig 1C of Calvi et al., 2022). The authors may wish to comment on this somewhat paradoxical observation. It obviously fits with the idea that *gsp-2(RNAi)* partially rescues polarization, but one might not necessarily expect membrane binding to actually increase when the phosphatase is depleted.

Line 136: show > shows

Line 320: To conclude, while our genetic data on PAR-2 cortical localization suggest that SDS-22 is not required to fully activate GSP-1 and/or GSP-2. (I don't understand this statement - see below line 493)

Line 332: *C.elegans* > *C. elegans*

Line 493: Similar to above "We show that SDS-22 is not essential for full activation of both GSP-1 and GSP-2, since depletion of SDS-22 does not abolish polarity"

This wording is odd given that the data show that GSP-1/2 activity/levels is likely reduced when SDS-22 is depleted/mutant. It is just that without SDS-22 they don't reach their full potential. I would rephrase to "GSP-1/2 activity does not strictly require SDS-22, though SDS-22 is required for them to reach their full, wild-type levels." or something like that

Line 505: Consistent with GSP-2 reduced levels should be with reduced levels of GSP-2

Dataset EV1 and EV2 - It would be helpful to include a description these files (perhaps we missed it).

Figure EV4 - Perhaps worth noting Conte et al., *Curr. Protoc. Mol. Biol.*, 2015 which note that feeding by RNAi is less effective following previous injection of dsRNA. Might suggest that the magnitude of *gsp-2*(partial) is different between EV4A-B and EV4C-D. But the fact that the goal was partial depletion means this is a very minor point.

Note - embedded figures have various problems with fonts that are not seen in the attached figures at the end. Worth making sure there isn't a formatting issue.

Referee #2:

The authors have dedicated substantial effort in addressing all of the reviewers concerns. The manuscript is more clear and the conclusions better justified. I fully support publication. I noted that Figure 6 and the associated caption lack some p value notations, even though I see them in the source file. Overall I thank the authors for making it a pleasure to evaluate this revised manuscript.

Referee #3:

This paper builds upon previous work from Gotta's lab (10.1083/jcb.202201048), which demonstrated that the PP1 catalytic subunits GSP-1 and GSP-2 are involved in *C. elegans* zygote polarisation. In the current study, the authors show that SDS-22, a known interactor of PP1, regulates PP1 activity in the zygote. Notably, depletion of SDS-22-similar to GSP-2 depletion-rescues polarity defects caused by aPKC inactivation. This suggests that SDS-22 promotes GSP-2's function in polarity establishment. The proposed mechanism is that SDS-22 protects GSP-1 and GSP-2 from proteasomal degradation. The manuscript has undergone a first round of revisions, during which the authors have addressed most of the reviewers' major concerns. I am satisfied with the additional experiments and revisions, particularly those clarifying SDS-22's role in stabilizing GSP-1 and GSP-2 protein levels, and how this contributes to zygote polarisation. In areas where minor concerns could not be fully addressed, the authors have acknowledged these limitations and provided clarifications in the text.

Overall, the study is carefully conducted, the conclusions are well supported, and the manuscript is clearly written. I find it suitable for publication.

Rev_Com_number: RC-2025-02880

New_manu_number: EMBOR-2025-61928V2

Corr_author: Gotta

Title: SDS-22 stabilizes GSP-1/-2 PP1 subunits contributing to polarity establishment in *C. elegans* embryos

Dear Martina,

Please find below the answers to the points you and reviewer 1 raised. We hope we have properly fixed everything now and we look forward to hearing from you.

Best regards,

Monica

- We note that Moura et al, 2024 is a preprint, which should be cited accordingly. Citations to manuscripts posted on recognized preprint servers can be cited the following way:

In-text citation: (preprint: NAME1 et al, YEAR)

In the reference list: Author NAME1, Author NAME2, (YEAR) article title. bioRxiv doi: 1234/002.dfj123 [PREPRINT]

We have corrected in text and in the reference list accordingly.

- Please remove the figures from the manuscript text, but their legends should be placed at the end of the manuscript.

We have removed the figures and placed the figure legends at the end of the manuscript.

- Please provide 3-5 keywords for your study. These will be visible in the html version of the paper and on PubMed and will help increase the discoverability of your work.

The key words are: Cell polarity, PP1 phosphatases, SDS22, PAR proteins, Proteasomal degradation

We have written them also below the abstract in the main text.

- Please remove Author Contributions from the manuscript text

This has been done.

- We note the phrase "our unpublished data" in the text, which is not allowed as per journal policy. Please either show the data or remove the statement.

We have removed the sentence as it is not essential to mention these data and they do not easily fit in this story.

- Funding information needs to be part of Acknowledgments, Funding section heading should be removed.

This has been corrected.

- The legends of EV should be removed from the figure files and placed after the main figure legends at the end of the manuscript.

This has been corrected.

- Some error messages pop up upon opening DATASET EV files. Please double check.

This has been solved.

- We note the following regarding the Appendix file:
 - o Please remove the two tables from the Appendix. Please submit Table EV2 as a separate file as Dataset EV. Please include its legend in a separate tab of the Excel file and please remember to update all callouts and file type.

We have now included the legends as a separate tab and we have included the Table EV2 as Data set EV3

- o I think Table EV1 in the Appendix file should be a part of Reagents & Tools table.

We have now included the reagents mentioned in Table EV1 in the Reagents and Tool table.

- o Please add page numbers and a Table of Contents where all items and their page numbers are listed.

We have now included a Table of Contents in the supplementary figures.

- As per the movies, please remove their legends from the manuscript. Instead, each movie should be zipped up with its corresponding legend (in a readme file) so that we have 3 zip folder uploaded.

This has been prepared accordingly.

- Please rename the Material and Methods section as Methods.

This has been corrected.

- During our routine figure check, we note a possible cell reuse between Figure 1B and Figure EV1B. Please clarify, also in the legends of both affected panels.

This was indeed a copy paste error (luckily the genotype at least was correct and it comes from the same big experiment which we split in two figures). We are glad your program spotted it. We have now corrected it and put another control embryo. As mentioned in the short cover letter, since one of the figures has changed, we needed to upload all the data again in the Yareta server. This will take about 2-3 days but we could reserve a new DOI which is already in the final version of the manuscript.

- Our production/data editors have asked you to clarify several points in the figure legends - Figure Legends (main + EV):

- o Please define the annotated p values ****/***/**/* as well as provide the exact p-values for the same in the legend of figure EV1 C as appropriate.

This was added.

- o Please note that the exact p values are not provided in the legends of figures 4B, 5A, B; 7B, C, E, F, H, I; EV3 A-D; EV4 B, D; EV5 A, B, D, F. *I note that the current Table EV2 in the Appendix file (which is to be converted into a Dataset EV, please see the relevant point above) is the table of p values. You can also refer to the table in the relevant legends.

We have now referred to the table in the appropriate figure legends.

- o Please indicate the statistical test used for data analysis in the legend of figure EV1 C

We have now added this information.

- Papers published in EMBO Reports include a 'synopsis' and 'bullet points' to further enhance discoverability. Both are displayed on the html version of the paper and are freely accessible to all readers. The synopsis includes a short standfirst summarizing the study in 1 or 2 sentences (max 35 words) that summarize the paper and are provided by the authors and streamlined by the handling editor. I would therefore ask you to include your synopsis blurb and 3-5 bullet points listing the key experimental findings.

We have provided a synopsis.

- In addition, please provide an image for the synopsis. This image should provide a rapid overview of the question addressed in the study but still needs to be kept fairly modest since the image size cannot exceed 550 (width) x 300-600 (height) pixels.

We have provided a scheme with the synopsis.

Thank you again for giving us to consider your manuscript for EMBO Reports, I look forward to your minor revision.

Kind regards,

Martina

--

Martina Rembold, PhD
Senior Scientific Editor
EMBO Reports

=====

Referee #1:

Overall, the authors have addressed the concerns raised in review. It was particularly useful to see the analysis using non-GFP/mNG tagged alleles as this alleviates our only major concern. We also appreciate the adjustments to language regarding conclusions. We have only a few minor comments that the authors may wish to address in revision.

Specific Comments:

Figure 1C, 3G: Reducing PP1 phosphatase activity in embryos should increase the total proportion of phosphorylated PAR-2 and hence reduce PAR-2 membrane binding. However, in the *pkc-3(ne4246)* background, posterior membrane levels of GFP::PAR-2 are higher in *sds-22(RNAi)* or *sds-22(E153A)* than controls (Fig 1C, 3G). Similar results were also seen in *gsp-2(RNAi)* (Fig 1C of Calvi et al., 2022). The authors may wish to comment on this somewhat paradoxical observation. It obviously fits with the idea that *gsp-2(RNAi)* partially rescues polarization, but one might not necessarily expect membrane binding to actually increase when the phosphatase is depleted.

Here it is important to keep into account that in the experiment mentioned SDS-22 is depleted in a kinase partially inactive background (*pkc-3(ts)*) and the levels of posteriorly enriched PAR-2 in the *pkc-3(ts)* strains depleted of SDS-22 are compared with the levels of PAR-2 in *pkc-3(ts)*, where PAR-2 is distributed all around the cortex. We have now better clarified in the text what is the control strain (page 5, line 124-127).

Line 136: show > shows

We have corrected this.

Line 320: To conclude, while our genetic data on PAR-2 cortical localization suggest that SDS-22 is not required to fully activate GSP-1 and/or GSP-2. (I don't understand this statement - see below line 493)

We have now re-phrased this sentence as suggested below by the reviewer:
"To conclude, while our genetic data on PAR-2 cortical localization suggest that GSP-1/-2 activity does not strictly require SDS-22,..." (page 8, line 224-225).

Line 332: C.elegans > C. elegans

This was corrected.

Line 493: Similar to above "We show that SDS-22 is not essential for full activation of both GSP-1 and GSP-2, since depletion of SDS-22 does not abolish polarity"
This wording is odd given that the data show that GSP-1/2 activity/levels is likely reduced when SDS-22 is depleted/mutant. It is just that without SDS-22 they don't reach their full potential. I would rephrase to "GSP-1/2 activity does not strictly require SDS-22, though SDS-22 is required for them to reach their full, wild-type levels." or something like that

We have now rephrased this sentence (and the one mentioned above) according to the suggestion of the reviewer.

"We show that GSP-1/-2 activity does not strictly require SDS-22,..." (page 11, line 356-357).

Line 505: Consistent with GSP-2 reduced levels should be with reduced levels of GSP-2

We have changed this (page 12, line 368).

Dataset EV1 and EV2 - It would be helpful to include a description these files (perhaps we missed it).

We have now included this in an additional tab in the Xcel files as suggested by the editor.

Figure EV4 - Perhaps worth noting Conte et al., Curr. Protoc. Mol. Biol., 2015 which note that feeding by RNAi is less effective following previous injection of dsRNA. Might suggest that the magnitude of gsp-2(partial) is different between EV4A-B and EV4C-D. But the fact that the goal was partial depletion means this is a very minor point.

We have now cited Conte et al., 2015 (page 14, line 444-446).

Note - embedded figures have various problems with fonts that are not seen in the attached figures at the end. Worth making sure there isn't a formatting issue.

We have now removed the embedded figures. As the reviewer mentioned, the PDF files of the figures display properly.

Referee #2:

The authors have dedicated substantial effort in addressing all of the reviewers concerns. The manuscript is more clear and the conclusions better justified. I fully support publication. I noted that Figure 6 and the associated caption lack some p value notations, even though I see them in the source file. Overall I thank the authors for making it a pleasure to evaluate this revised manuscript.

Referee #3:

This paper builds upon previous work from Gotta's lab (10.1083/jcb.202201048), which demonstrated that the PP1 catalytic subunits GSP-1 and GSP-2 are involved in *C. elegans* zygote polarisation. In the current study, the authors show that SDS-22, a known interactor of PP1, regulates PP1 activity in the zygote. Notably, depletion of SDS-22-similar to GSP-2 depletion-rescues polarity defects caused by aPKC inactivation. This suggests that SDS-22 promotes GSP-2's function in polarity establishment. The proposed mechanism is that SDS-22 protects GSP-1 and GSP-2 from proteasomal degradation.

The manuscript has undergone a first round of revisions, during which the authors have addressed most of the reviewers' major concerns. I am satisfied with the additional experiments and revisions, particularly those clarifying SDS-22's role in stabilizing GSP-1 and GSP-2 protein levels, and how this contributes to zygote polarisation. In areas where minor concerns could not be fully addressed, the authors have acknowledged these limitations and provided clarifications in the text.

Overall, the study is carefully conducted, the conclusions are well supported, and the manuscript is clearly written. I find it suitable for publication.

We thank all reviewers for their constructive and accurate comments and for helping us to improve our manuscript.

Rev_Com_number: RC-2025-02880

New_manu_number: EMBOR-2025-61928V2

Corr_author: Gotta

Title: SDS-22 stabilizes GSP-1/-2 PP1 subunits contributing to polarity establishment in *C. elegans* embryos

Monica Gotta
University of Geneva
Cell Physiology and Metabolism
1, rue Michel Servet
Geneva 1211
Switzerland

Dear Dr. Gotta,

I am very pleased to accept your manuscript for publication in the next available issue of EMBO reports. Thank you for your contribution to our journal.

Kind regards,

Rev_Com_number: RC-2025-02880

New_manu_number: EMBOR-2025-61928V3

Corr_author: Gotta

Title: SDS-22 stabilizes GSP-1/-2 PP1 subunits contributing to polarity establishment in *C. elegans* embryos